# Modulation of GluA2–γ5 synaptic complex desensitization, polyamine block and antiepileptic perampanel inhibition by auxiliary subunit cornichon-2

**Shanti Pal Gangwar** [1,4], **Laura Y. Yen**[1,2,4], **Maria V. Yelshanskaya**[1,4], **Aryeh Korman** [3], **Drew R. Jones** [3] & **Alexander I. Sobolevsky** [1]✉

Synaptic complexes of α-amino-3-hydroxy-5-methyl-4-isoxazolepropionic acid (AMPA) receptors (AMPARs) with auxiliary subunits mediate most excitatory neurotransmission and can be targeted to treat neuropsychiatric and neurological disorders, including epilepsy. Here we present cryogenic-electron microscopy structures of rat GluA2 AMPAR complexes with inhibitory mouse γ5 and potentiating human cornichon-2 (CNIH2) auxiliary subunits. CNIH2 appears to destabilize the desensitized state of the complex by reducing the separation of the upper lobes in ligand-binding domain dimers. At the same time, CNIH2 stabilizes binding of polyamine spermidine to the selectivity filter of the closed ion channel. Nevertheless, CNIH2, and to a lesser extent γ5, attenuate polyamine block of the open channel and reduce the potency of the antiepileptic drug perampanel that inhibits the synaptic complex allosterically by binding to sites in the ion channel extracellular collar. These findings illustrate the fine-tuning of synaptic complex structure and function in an auxiliary subunit-dependent manner, which is critical for the study of brain region-specific neurotransmission and design of therapeutics for disease treatment.

Complexes of α-amino-3-hydroxy-5-methyl-4-isoxazolepropionic acid receptors (AMPARs) with auxiliary subunits mediate the fast excitatory neurotransmission in the central nervous system (CNS)[1]. The core of each complex is a Y-shaped tetrameric AMPAR with a three-layer domain organization[2–5]. At the base of the Y is a roughly fourfold symmetrical cation-selective ion channel assembled of four transmembrane domains (TMDs). Each TMD includes three transmembrane helices, M1, M3, M4, and a re-entrant M2 pore loop. Above the channel is a layer of four clamshell-shaped ligand-binding domains (LBDs), each composed of two (S1–2) polypeptide stretches. The top layer is formed by four amino-terminal domains (ATDs). Both LBD and ATD layers have twofold symmetrical dimer-of-dimers organization, with domain swapping between these layers.

Auxiliary subunits are membrane proteins that associate with and regulate AMPAR trafficking, cellular localization, gating kinetics and pharmacology[1,6,7]. Transmembrane AMPAR regulatory proteins (TARPs) and cornichons (CNIHs) are the most abundant auxiliary subunits in the CNS[8–12] and can co-assemble with AMPARs simultaneously[13,14]. TARPs, originally named 'γ-subunits' based on sequence homology to the calcium channel γ1 subunit[15], are classified into type I (γ2, γ3, γ4, γ8) and type II (γ5, γ7) subunits, which assume generally activating and suppressive functions, respectively[6,8–12,16–20]. The CNIH family

[1]Department of Biochemistry and Molecular Biophysics, Columbia University, New York, NY, USA. [2]Cellular and Molecular Physiology and Biophysics Graduate Program, Columbia University Irving Medical Center, New York, NY, USA. [3]Department of Biochemistry and Molecular Pharmacology, NYU Langone Health, New York, NY, USA. [4]These authors contributed equally: Shanti Pal Gangwar, Laura Y. Yen, Maria V. Yelshanskaya. ✉e-mail: as4005@cumc.columbia.edu

consists of CNIH1–4, with only CNIH2 and CNIH3 functioning as AMPAR auxiliary subunits. CNIHs promote receptor trafficking and potentiate gating by increasing glutamate (Glu) potency and slowing the rates of deactivation and desensitization[21,22].

Structures of two type I TARPs, γ2 (or stargazin) and γ8; one type II TARP, γ5; and both CNIH2–3 in complex with AMPARs have been determined previously[23–29]. Despite the abundance of AMPAR synaptic complexes that include both TARPs and CNIHs in the CNS[13,30,31], structures are only available for AMPARs co-assembled with type I TARP γ8 and CNIH2 (refs. 32,33). There are no structures of AMPAR complexes co-assembled with CNIHs and type II TARPs. We recently solved structures of AMPAR in complex with type II TARP γ5 (ref. 29). While the probability of γ5 and CNIH2 to be constituents of the same synaptic complex in the CNS has not been studied, they show similar molecular abundancies in the cerebellum[34]. In particular, they are both present in the Bergmann glia, which typically express Ca²⁺-permeable AMPARs and receive direct input from glutamatergic neurons[17]. They are also present in the hippocampal NG2 glial cells[35,36].

Here we describe the cumulative effect of the potentiating CNIH2 and inhibitory γ5 auxiliary subunits on the structure and function of homotetrameric AMPARs, assembled from GluA2 subunits. CNIH2 slows AMPAR deactivation and desensitization, and this deceleration is attenuated in the presence of γ5, which does not affect the rates of deactivation and desensitization alone. Both γ5 and CNIH2 slow down recovery from desensitization but the effect of γ5 appears to be much stronger. CNIH2 completely reverses the inhibitory effect of γ5 on steady-state currents, indicating a dramatic weakening of desensitization caused by CNIH2. We identify the polyamine spermidine (SPD) binding site in the selectivity filter of the closed-state GluA2–γ5–CNIH2 pore. Both γ5 and CNIH2 attenuate the polyamine block of the open channels and reduce the potency of AMPAR noncompetitive inhibition by perampanel (PMP), although in both cases the effect of CNIH2 appears to be stronger.

## Results

### Structure of the GluA2–γ5–CNIH2 complex

Previously, we solved the structure of AMPAR–γ5 complex by expressing the GluA2–γ5 fusion construct, where the N terminus of γ5 was covalently linked to the C terminus of GluA2 (modified calcium-permeable rat GluA2(Q)$_{flip}$ subunit; Methods) in human embryonic kidney 293S (HEK293S) GnTI⁻ cells in the presence of competitive antagonists ZK200775 (ZK, 2 nM) and kynurenic acid (0.1 mM)[29]. Recently, we noticed that HEK293S GnTI⁻ cells transduced with GluA2–γ5 and grown in the absence of antagonists displayed more intense fluorescence of the C-terminally concatenated green fluorescent protein (GFP). We purified protein from these cells, supplemented it with 0.1 mM ZK and subjected to cryo-EM. Data analysis revealed two distinct populations of particles (Extended Data Fig. 1). One population resulted in the GluA2–γ5$_{ZK}$ structure, identical to the one published previously[29], with two molecules of γ5 around the GluA2 TMD.

The second population produced a three-dimensional (3D) reconstruction with densities for four auxiliary subunits around the GluA2 channel (Fig. 1a,b). An overall resolution of the corresponding cryo-EM map (3.58 Å) was improved for the LBD–TMD region (3.21 Å) by ATD and micelle signal subtraction (Extended Data Fig. 1, Supplementary Fig. 1 and Table 1). The map quality was sufficient to unambiguously identify and build models of four GluA2, two γ5 and two endogenous human CNIH2 subunits (Fig. 1c–e). Separately, we reprocessed the data for GluA2–γ5 structures published previously[29] and collected cryo-EM data for GluA2 alone, and found no endogenous CNIH2 in the resulting 3D reconstructions, suggesting that recruitment of CNIH2 to the GluA2–γ5–CNIH2 complex is related to both the presence of γ5 and absence of antagonists in the expression media.

Despite each GluA2 subunit in GluA2–γ5–CNIH2 being covalently linked to γ5, density was observed for only two γ5 subunits, suggesting

that similar to GluA2–γ5, the other two γ5s were disordered. The ordered γ5 subunits adapt the claudin fold[29] and comprise a bundle of four transmembrane α-helices (TM1–TM4), with an extracellular head domain assembled of TM1–TM2 and TM3–TM4 loops, with a five-stranded β-sheet core (Fig. 1f). TM3 and TM4 of γ5 form a binding interface with M1 of GluA2 subunit A or C and M4 of GluA2 subunit B or D (Fig. 1e), while the β1–β2 loop of γ5 makes contact with the lower lobe D2 of LBD in GluA2 subunits A and C, important for regulation of AMPAR function[29].

Two CNIH2 subunits on the periphery of AMPAR TMD occupy positions between GluA2 protomers A/D and B/C. Each CNIH2 subunit folds into a tetrahelical bundle, common for CNIH2 and CNIH3 (refs. 32,37), with the N and C termini located extracellularly (Fig. 1g). The transmembrane helices TM1 and TM2 of CNIH2 form a binding interface with M1 of GluA2 subunit B or D and M4 of GluA2 subunit A or C (Fig. 1e). Positions of CNIH2 in GluA2–γ5–CNIH2 complex are similar to their positions in the competitive antagonist-bound GluA1/2–γ8–CNIH2 complex[32] (Extended Data Fig. 2a,b). Despite the nearly identical positioning of the transmembrane elements of the auxiliary subunits in these two complexes, the LBD layer undergoes a roughly 9° rotation (Extended Data Fig. 2c). This striking difference is likely the result of distinct interactions of the γ5 and γ8 head domains with the LBDs, represented by contacts of β1–β2 loop in γ5 with the D2 lobes of LBDs in GluA2 subunits A and C versus putative contacts of β4–TM2 loop in γ8 with the D2 lobes of LBDs in GluA2 subunits B and D.

The role of CNIH2 in determining the conformation of the antagonist-bound closed state appears to be minor, as evidenced by only small differences in the ZK-bound structure of GluA2–γ5–CNIH2 compared to the previously solved structure of GluA2–γ5$_{ZK}$ (refs. 29) (Extended Data Fig. 3a–f). Thus, binding of CNIH2 to GluA2–γ5 appears to cause slight splaying of the LBD dimers, clockwise rotation of the ATD layer and counterclockwise displacement of the γ5 subunits and M4 segments (Extended Data Fig. 3g–i). Nevertheless, these changes are minor and do not affect conformations of individual LBD clamshells (Extended Data Fig. 3j–k), LBD dimers (Extended Data Fig. 3l) or the closed ion channel pore (Extended Data Fig. 4). We also investigated whether ZK affected the closed-state conformations of the GluA2–γ5 and GluA2–γ5–CNIH2 complexes by solving structures in the apo state (Supplementary Fig. 2). We found that the ZK-bound and apo-state structures were indistinguishable with nearly identical conformations of the individual LBDs and ion channel (Extended Data Fig. 5), suggesting that both structures represent the same closed state. ZK-bound structures are used as references for further analysis as they produced higher quality structures.

### Functional characterization

We transfected HEK293 cells with GluA2 or GluA2–γ5 fusion individually or cotransfected them with wild-type human CNIH2 and subjected these cells to patch-clamp recordings (Fig. 2 and Extended Data Table 1). For GluA2–γ5 coexpressed with CNIH2, a 2 ms application of 3 mM Glu at −60 mV membrane potential elicited a rapidly activating inward current that quickly ($\tau_{Deact}$ = 4.21 ± 0.42 ms, $n$ = 9) decayed to zero, mainly due to receptor deactivation (Fig. 2a, red trace). The corresponding deactivation rate was nevertheless about twice slower than for the GluA2 receptor alone ($\tau_{Deact}$ = 1.76 ± 0.24 ms, $n$ = 9)[38] or GluA2–γ5 ($\tau_{Deact}$ = 2.14 ± 0.15 ms, $n$ = 8)[29] but roughly 2.5 times faster than for GluA2 coexpressed with CNIH2 ($\tau_{Deact}$ = 10.8 ± 1.8 ms, $n$ = 11) (Fig. 2b). A prolonged, 500 ms Glu application elicited an inward current that decayed in the continuous presence of Glu more slowly ($\tau_{Des}$ = 14.8 ± 1.6 ms, $n$ = 14) than in response to the short 2 ms Glu application, apparently due to receptor desensitization (Fig. 2a, blue trace). Similar to deactivation (Fig. 2b), the desensitization rate for GluA2–γ5 coexpressed with CNIH2 was substantially slower compared to GluA2 alone ($\tau_{Des}$ = 7.70 ± 0.35 ms, $n$ = 35)[38] or GluA2–γ5 ($\tau_{Des}$ = 8.04 ± 0.85 ms, $n$ = 10)[29] but faster than for GluA2 coexpressed with CNIH2 ($\tau_{Des}$ = 28.0 ± 4.5 ms, $n$ = 11) (Fig. 2c).

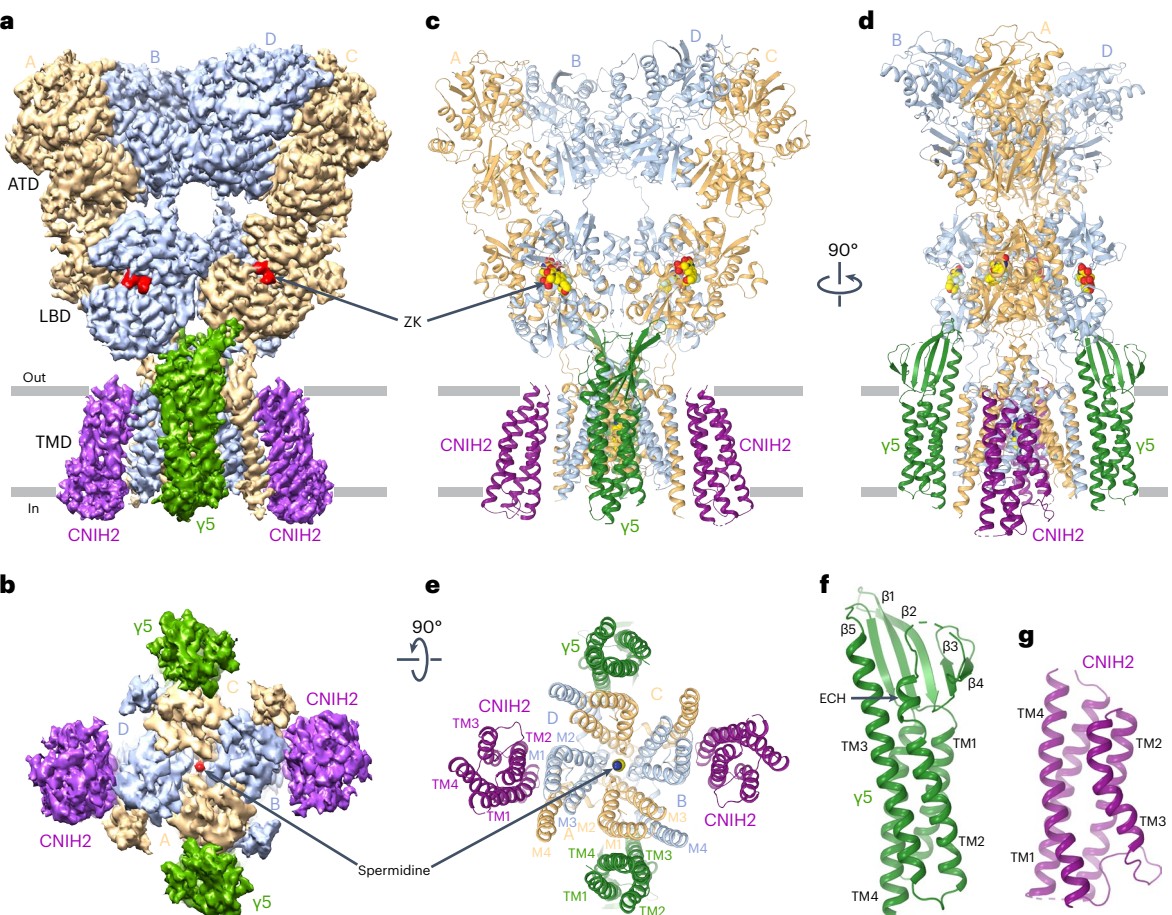

**Fig. 1 | Structure of GluA2–γ5–CNIH2 complex. a,b,** 3D cryo-EM reconstruction of GluA2–γ5–CNIH2$_{ZK-SPD}$ viewed parallel to membrane (**a**) and intracellularly (**b**), with density for GluA2 subunits colored yellow (subunits A and C) and blue (subunits B and D), γ5 in dark green, CNIH2 in purple, antagonist ZK and channel blocker SPD in red. **c–e,** GluA2–γ5–CNIH2$_{ZK-SPD}$ structure viewed parallel to membrane (**c**, broad face; **d**, narrow face) or intracellularly (**e**), with protomers colored similarly to **a** and **b** and molecules of ZK and SPD shown as space-filling models. **f,g,** Structures of auxiliary subunits γ5 and CNIH2, with the secondary structure elements labeled.

Desensitization was nearly eliminated when Glu was applied in the presence of the positive allosteric modulator cyclothiazide (CTZ) (Fig. 2a, green trace). As an estimate of the fraction of nondesensitized channels, we calculated the ratio of steady-state current amplitude in the continuous presence of Glu ($I_{SS}$) and the maximal current amplitude in the presence of CTZ ($I_{Max}$). For GluA2–γ5 coexpressed with CNIH2, $I_{SS}/I_{Max} = 0.060 \pm 0.014$ ($n = 11$). This value was similar to the value for the GluA2 receptor alone ($I_{SS}/I_{Max} = 0.046 \pm 0.01$, $n = 26$)[39] but appeared to be a result of the significant increase in the fraction of nondesensitized channels caused by CNIH2 (for GluA2 plus CNIH2, $I_{SS}/I_{Max} = 0.211 \pm 0.031$, $n = 11$) compensated by the dramatic reduction of the steady-state current induced by γ5 (for GluA2–γ5, $I_{SS}/I_{Max} = 0.0094 \pm 0.0024$, $n = 9$)[29] (Fig. 2d).

We calculated the time constant of recovery from desensitization using a double-pulse protocol (Fig. 2e). The rate of recovery from desensitization for GluA2–γ5 coexpressed with CNIH2 ($\tau_{RecDes} = 36.8 \pm 1.4$ ms, $m = 3.80 \pm 0.62$, $n = 8$) was slower than for GluA2 ($\tau_{RecDes} = 15.3 \pm 1.1$ ms, $m = 4.07 \pm 0.66$, $n = 14$)[38] or GluA2 coexpressed with CNIH2 ($\tau_{RecDes} = 21.7 \pm 1.4$ ms, $m = 2.62 \pm 0.14$, $n = 9$) but similar to the one for GluA2–γ5 ($\tau_{RecDes} = 29.8 \pm 1.9$ ms, $m = 2.81 \pm 0.21$, $n = 6$)[29], confirmed by statistical comparison of $\tau_{RecDes}$ values obtained from fitting the data for individual cells (Fig. 2f and Extended Data Table 1). Therefore, while both γ5 and CNIH2 slowed down the recovery from desensitization, the effect of γ5 appeared to be stronger. Compared to GluA2–γ5, the increase in $\tau_{RecDes}$ observed for GluA2–γ5 coexpressed with CNIH2 was not statistically significant (Fig. 2f). This may happen because similar to the HEK293S GnTI⁻ cells, which were used for structural studies, HEK293 cells used for patch-clamp recordings also express endogenous CNIH2. However, given the statistically significant opposing effects of γ5 and CNIH2 on $I_{SS}/I_{Max}$ for GluA2–γ5 and GluA2 coexpressed with CNIH2 compared to the GluA2 alone (Fig. 2d), the expression of endogenous CNIH2 in the plasma membrane of HEK293 cells transfected with GluA2–γ5 can be considered negligible.

Our functional experiments, therefore, allowed crude separation and analysis of the individual effects of γ5 and CNIH2 auxiliary subunits on AMPAR function, consistent with previous reports[21,22,40].

## Polyamine SPD binding in the selectivity filter

Comparison of the closed-state ZK-bound structures in the presence and absence of CNIH2 revealed a strong cylindrical density in the selectivity filter of the GluA2–γ5–CNIH2 pore, which was not present in GluA2–γ5. Based on its shape and location, this density may represent endogenous polyamines that produce rectification of Ca²⁺-permeable AMPAR currents in physiological conditions[41–43]. We subjected the protein that was used in cryo-EM experiments to mass spectrometry (MS) analysis and identified the presence of polyamine SPD (Extended Data Fig. 6). In agreement with this finding, the density in the selectivity

**Table 1 | Cryo-EM data collection, refinement and validation statistics**

| | GluA2–γ5–CNIH2ZK-SPD FL (EMDB-40741) (PDB 8SS2) | GluA2–γ5–CNIH2ZK-SPD LBD–TMD (EMDB-40742) (PDB 8SS3) | GluA2–γ5–CNIH2SPD LBD–TMD (EMDB-40743) (PDB 8SS4) | GluA2–γ5apo LBD–TMD (EMDB-40744) (PDB 8SS5) | GluA2–γ5–CNIH2ZK-PMP-SPD FL (EMDB-40745) (PDB 8SS6) |
|---|---|---|---|---|---|
| **Data collection and processing** | | | | | |
| Magnification | ×105,000 | ×105,000 | ×81,000 | ×81,000 | ×75,000 |
| Voltage (kV) | 300 | 300 | 300 | 300 | 300 |
| Electron exposure (e−/Å²) | 58 | 58 | 45 | 45 | 50 |
| Defocus range (μm) | −1 to −2 | −1 to −2 | −1 to −2 | −1 to −2 | −1 to −2 |
| Pixel size (Å) | 0.83 | 0.83 | 1.1 | 1.1 | 0.925 |
| Symmetry imposed | C2 | C2 | C2 | C2 | C2 |
| Initial particle images (no.) | 1,438,201 | 1,438,201 | 3,730,352 | 3,730,352 | 2,360,956 |
| Final particle images (no.) | 55,275 | 81,723 | 121,812 | 81,879 | 106,582 |
| Map resolution (Å) | 3.58 | 3.21 | 3.33 | 3.56 | 3.01 |
| FSC threshold | 0.143 | 0.143 | 0.143 | 0.143 | 0.143 |
| Map resolution range (Å) | 1.9/4.2/43.5 | 1.9/3.5/40.3 | 2.3/3.5/10.7 | 2.3/4.3/37.8 | 2.0/3.8/39.8 |
| **Refinement** | | | | | |
| Initial model used (PDB code) | 7RZ5, 7OCE | 7RZ5, 7OCE | 7RZ5, 7OCE | 7RZ5, 7OCE | 7RZ5, 7OCE |
| Model resolution (Å) | 3.58 | 3.21 | 3.33 | 3.56 | 3.01 |
| FSC threshold | 0.143 | 0.143 | 0.143 | 0.143 | 0.143 |
| Map sharpening B factor (Å²) | −101.1 | −102.3 | −110.3 | −119.4 | −138.4 |
| Model composition | | | | | |
| Nonhydrogen atoms | 31,146 | 19,888 | 17,978 | 15,637 | 31,069 |
| Protein residues | 3,802 | 2,280 | 2,278 | 1,998 | 3,794 |
| Ligands | | | | | |
| PMP | – | – | – | – | 4 |
| SPD | 1 | 1 | 1 | – | 1 |
| ZK | 4 | 4 | – | – | 4 |
| B factors (Å²) | | | | | |
| Protein | 90.45 | 72.04 | 67.53 | 78.94 | 87.81 |
| Ligand | 62.19 | 25.38 | 85.35 | 67.93 | 39.86 |
| R.m.s. deviations | | | | | |
| Bond lengths (Å) | 0.012 | 0.011 | 0.009 | 0.006 | 0.007 |
| Bond angles (°) | 1.489 | 1.576 | 1.332 | 1.305 | 1.484 |
| **Validation** | | | | | |
| MolProbity score | 2.08 | 2.06 | 1.63 | 1.77 | 1.76 |
| Clashscore | 7.35 | 11.22 | 3.21 | 3.84 | 4.82 |
| Poor rotamers (%) | 0.80 | 0.52 | 0.36 | 0.24 | 1.05 |
| Ramachandran plot | | | | | |
| Favored (%) | 84.29 | 91.70 | 91.15 | 88.25 | 91.66 |
| Allowed (%) | 15.02 | 7.59 | 8.49 | 10.63 | 7.91 |
| Disallowed (%) | 0.69 | 0.71 | 0.36 | 1.12 | 0.43 |

filter of the GluA2–γ5–CNIH2 pore showed a near-perfect fit of the SPD molecule (Fig. 3a–d), independent of data processed in C1 or C2 symmetry (Extended Data Fig. 7). The SPD binding site is formed by the amide group of Q586 and backbone carbonyls of Q586 and G588, consistent with the prediction based on GluA2–γ2 structures in complex with toxins and toxin-like molecules[25].

We compared voltage dependencies of Glu-induced currents recorded in the presence of CTZ from HEK293 cells transfected with GluA2 or the GluA2–γ5 individually or cotransfected with CNIH2 (Fig. 3e,f). Consistent with previous observations[41–43], the Ca²⁺-permeable GluA2 homotetramers that have Q at the Q/R site showed strong inward rectification, characterized by low values of the current rectification index ($I_{+40\,mV}/I_{−40\,mV}$) calculated as a ratio of current amplitudes at +40 and −40 mV (Fig. 3g). GluA2–γ5-mediated currents showed reduced inward rectification, consistent with the previously observed attenuation of polyamine block of GluA4 receptors by γ5 (ref. 17).

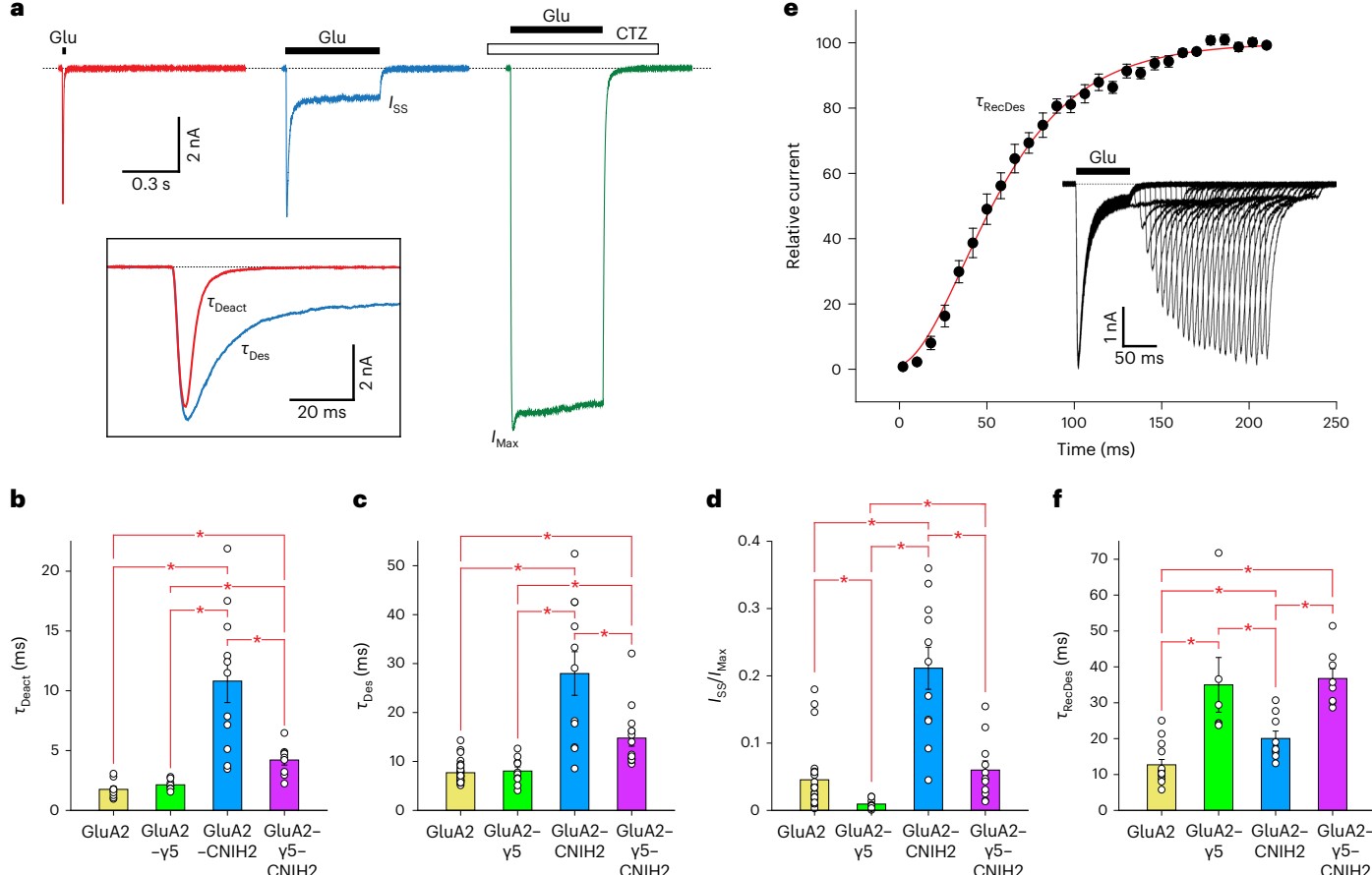

**Fig. 2 | Functional characterization of GluA2 complexes with γ5 and CNIH2. a**, Representative whole-cell currents recorded at −60 mV membrane potential from HEK293 cell coexpressing GluA2–γ5 and CNIH2 in response to 2 ms (red) or 500 ms (blue) application of 3 mM Glu alone or application of Glu in the continuous presence of 30 μM CTZ (green). The inset shows close-up superposition of currents in response to 2 ms and 0.5 s applications of Glu alone. **b–d**, Time constants of deactivation (**b**, $\tau_{Deact}$) and desensitization (**c**, $\tau_{Des}$) and the fraction of nondesensitized channels (**d**, $I_{SS}/I_{Max}$) measured for currents recorded from HEK293 cells transfected with GluA2 (yellow), GluA2–γ5 (green), GluA2 and CNIH2 (blue) and GluA2–γ5 and CNIH2 (purple). Asterisks indicate statistically significant differences (two-sided two-sample $t$-test; the significance is assumed if $P < 0.05$). For $\tau_{Deact}$ (**b**), the number of independent experiments, $n = 9$ for GluA2, $n = 8$ for GluA2–γ5, $n = 11$ for GluA2 and CNIH2, and $n = 9$ for GluA2–γ5 and CNIH2. Probabilities for the two-sided two-sample $t$-test, $P = 0.222$ for GluA2 versus GluA2–γ5, $P = 2.70 \times 10^{-4}$ for GluA2 versus GluA2 and CNIH2, $P = 1.15 \times 10^{-4}$ for GluA2 versus GluA2–γ5 and CNIH2, $P = 7.79 \times 10^{-4}$ for GluA2–γ5 versus GluA2 and CNIH2, $P = 4.83 \times 10^{-4}$ for GluA2–γ5 versus GluA2–γ5 and CNIH2, and $P = 0.00445$ for GluA2 and CNIH2 versus GluA2–γ5 and CNIH2. For $\tau_{Des}$ (**c**), the number of independent experiments, $n = 35$ for GluA2, $n = 10$ for GluA2–γ5, $n = 11$ for GluA2 and CNIH2, and $n = 14$ for GluA2–γ5 and CNIH2. Probabilities for the two-sided two-sample $t$-test, $P = 0.673$ for GluA2 versus GluA2–γ5, $P = 3.28 \times 10^{-10}$ for GluA2 versus GluA2 and CNIH2, $P = 1.91 \times 10^{-7}$ for GluA2 versus GluA2–γ5 and CNIH2,

$P = 4.91 \times 10^{-4}$ for GluA2–γ5 versus GluA2 and CNIH2, $P = 0.0037$ for GluA2–γ5 versus GluA2–γ5 and CNIH2, and $P = 0.0058$ for GluA2 and CNIH2 versus GluA2–γ5 and CNIH2. For $I_{SS}/I_{Max}$ (**d**), the number of independent experiments, $n = 26$ for GluA2, $n = 9$ for GluA2–γ5, $n = 11$ for GluA2 and CNIH2, and $n = 11$ for GluA2–γ5 and CNIH2. Probabilities for the two-sided two-sample $t$-test, $P = 0.0265$ for GluA2 versus GluA2–γ5, $P = 6.76 \times 10^{-8}$ for GluA2 versus GluA2 and CNIH2, $P = 0.385$ for GluA2 versus GluA2–γ5 and CNIH2, $P = 1.72 \times 10^{-5}$ for GluA2–γ5 versus GluA2 and CNIH2, $P = 0.00395$ for GluA2–γ5 versus GluA2–γ5 and CNIH2, and $P = 2.55 \times 10^{-4}$ for GluA2 and CNIH2 versus GluA2–γ5 and CNIH2. Data are mean ± s.e.m. **e**, Mean recovery from desensitization for GluA2–γ5–CNIH2 activated by Glu measured using the two-pulse protocol illustrated in the inset. The red curve through the points is a fit with the Hodgkin–Huxley equation. The number of independent experiments, $n = 8$. Error bars represent s.e.m. **f**, Time constant of recovery from desensitization ($\tau_{RecDes}$). Asterisks indicate statistically significant differences (two-sided two-sample $t$-test; the significance is assumed if $P < 0.05$). The number of independent experiments, $n = 14$ for GluA2, $n = 6$ for GluA2–γ5, $n = 9$ for GluA2 and CNIH2, and $n = 8$ for GluA2–γ5 and CNIH2. Probabilities for the two-sided two-sample $t$-test, $P = 5.53 \times 10^{-4}$ for GluA2 versus GluA2–γ5, $P = 0.00707$ for GluA2 versus GluA2 and CNIH2, $P = 4.38 \times 10^{-8}$ for GluA2 versus GluA2–γ5 and CNIH2, $P = 0.0418$ for GluA2–γ5 versus GluA2 and CNIH2, $P = 0.811$ for GluA2–γ5 versus GluA2–γ5 and CNIH2, and $P = 1.72 \times 10^{-4}$ for GluA2 and CNIH2 versus GluA2–γ5 and CNIH2. Data are mean ± s.e.m.

Even stronger reduction of inward rectification was observed when HEK293 cells were cotransfected with GluA2 and CNIH2, consistent with the CNIH2-induced attenuation of polyamine block reported before[22,44]. Similarly weakened rectification was observed for currents recorded from HEK293 cells cotransfected with GluA2–γ5 and CNIH2, suggesting stronger attenuation of polyamine block by CNIH2 than γ5. This effect of CNIH2 was seemingly opposite to what was expected, given that the presence of CNIH2 was crucial for revealing SPD binding (Fig. 3a–d). However, SPD binding was observed in the closed state,

while the functional effect of CNIH2 is mediated by the open channel conformation.

We superimposed the SPD-bound closed-state GluA2–γ5–CNIH2$_{ZK-SPD}$ structure and 1-naphthyl acetylspermine (NASPM)-bound open-state GluA2–γ2$_{Glu+CTZ+NASPM}$ structure[25] (Fig. 4a,b). The near-perfect overlap of SPD and SPD-like moiety of NASPM (Fig. 4c) strongly supported the idea that binding of polyamines to the selectivity filter is the likely cause of inward rectification of $Ca^{2+}$-permeable AMPAR-mediated currents[41–43]. Comparison of the pore dimensions

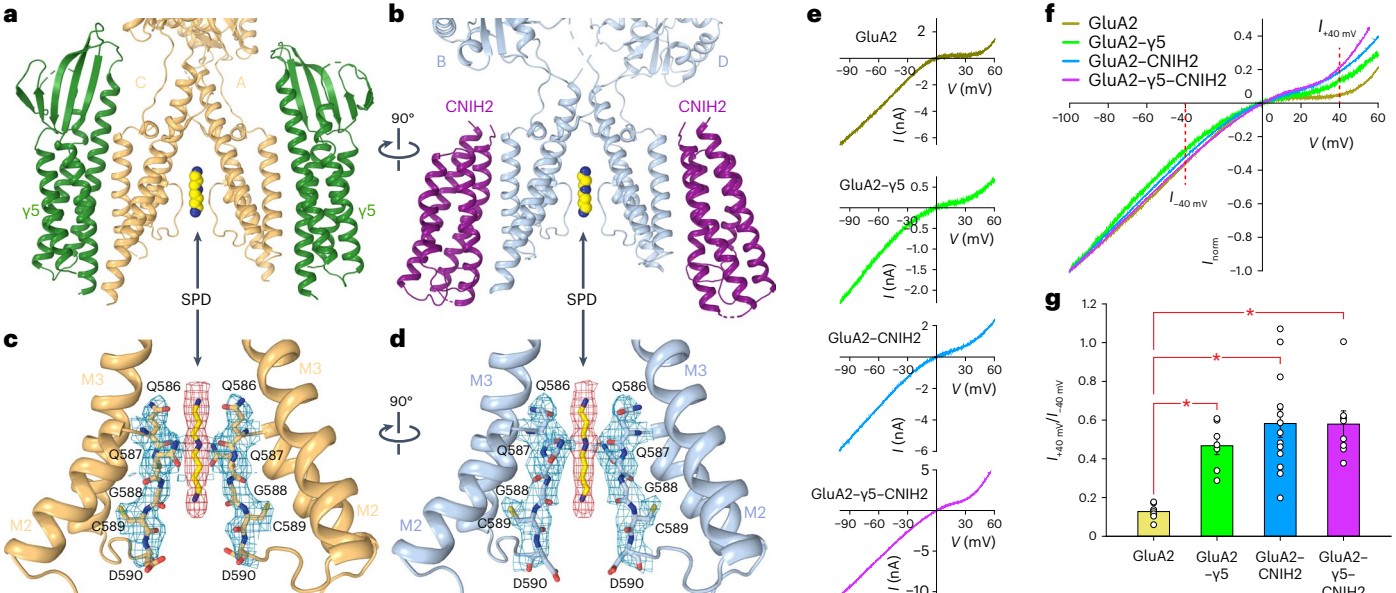

**Fig. 3 | Polyamine spermidine binding site in GluA2–γ5–CNIH2 pore and voltage dependence. a,b**, TMD of GluA2–γ5–CNIH2$_{ZK-SPD}$ viewed parallel to the membrane from two perpendicular directions. Only two of four GluA2 subunits and two of four auxiliary subunits are shown in each panel, with subunits B/D and CNIH2 (**a**) or A/C and γ5 (**b**) omitted for clarity. Molecules of SPD are shown as space-filling models. **c,d**, Close-up views of the SPD binding site, with SPD molecule and residues of the selectivity filter shown in sticks and the corresponding cryo-EM density as red and blue mesh, respectively. Only two of four GluA2 subunits are shown in each panel, with subunits B/D (**c**) or A/C (**d**) omitted for clarity. **e**, Examples of voltage dependencies of whole-cell currents recorded from HEK293 cells transfected with GluA2 (yellow), GluA2–γ5 (green), GluA2 and CNIH2 (blue), and GluA2–γ5 and CNIH2 (purple) in response

to –100 to +60 mV voltage ramp in the continuous presence of 30 μM CTZ. **f**, Superposition of voltage dependencies from **e** normalized to the current amplitude at –100 mV. **g**, Rectification index ($I_{+40\,mV}/I_{-40\,mV}$) calculated as a ratio of current amplitudes at +40 and –40 mV. Asterisks indicate statistically significant differences (two-sided two-sample $t$-test; the significance is assumed if $P < 0.05$). The number of independent experiments, $n = 7$ for GluA2, $n = 7$ for GluA2–γ5, $n = 13$ for GluA2 and CNIH2, and $n = 8$ for GluA2–γ5 and CNIH2. Probabilities for the two-sided two-sample $t$-test, $P = 1.39 \times 10^{-5}$ for GluA2 versus GluA2–γ5, $P = 2.36 \times 10^{-4}$ for GluA2 versus GluA2 and CNIH2, $P = 4.04 \times 10^{-5}$ for GluA2 versus GluA2–γ5 and CNIH2, $P = 0.286$ for GluA2–γ5 versus GluA2 and CNIH2, $P = 0.210$ for GluA2–γ5 versus GluA2–γ5 and CNIH2, and $P = 0.978$ for GluA2 and CNIH2 versus GluA2–γ5 and CNIH2. Data are mean ± s.e.m.

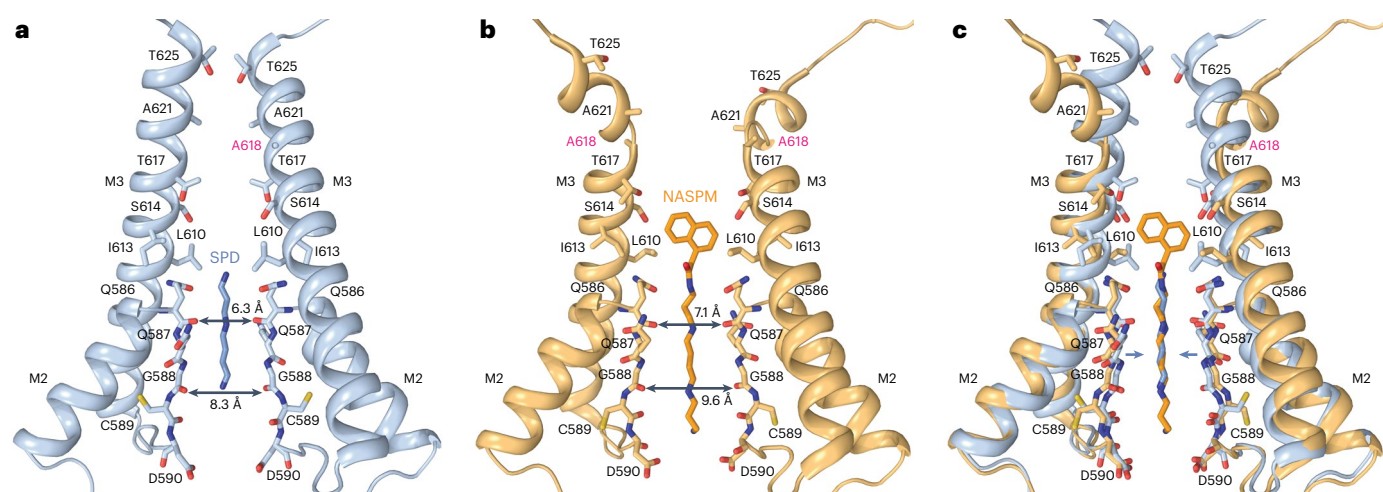

**Fig. 4 | Comparison of SPD and NASPM binding sites. a–c**, Pore-forming domains in the closed-pore structure of GluA2–γ5–CNIH2$_{ZK-SPD}$ (**a**) and open-pore structure of GluA2–γ2$_{Glu+CTZ+NASPM}$ (**b**, PDB ID 6DM1), and their superposition (**c**). Only two of four subunits (B and D) are shown, with the front and back subunits (A and C) omitted for clarity. Residues lining the pore are shown as stick models.

Double arrows in **a** and **b** indicate distances between backbone carbonyls of Q586 and G588 that form narrow constrictions in the selectivity filter. Blue arrows in **c** illustrate narrowing of the selectivity filter in the open-pore compared to closed-pore structures.

suggested that channel opening can cause substantial widening of the selectivity filter. Similar to GluA2–γ2$_{Glu+CTZ+NASPM}$ stronger widening of the selectivity filter in GluA2–γ5–CNIH2 may lead to a substantial weakening of SPD binding, SPD permeation through the pore and reduced current rectification, as observed in our experiments (Fig. 3e–g) and previously[22,44].

## Inhibition by antiepileptic drug PMP
We tested the effect of CNIH2 on noncompetitive inhibition by PMP. Glu-activated currents recorded in the presence of CTZ from HEK293 cells cotransfected with GluA2–γ5 and CNIH2 were inhibited by PMP in a concentration-dependent manner (Fig. 5a). The concentration dependence of PMP inhibition was fitted by the logistic equation with

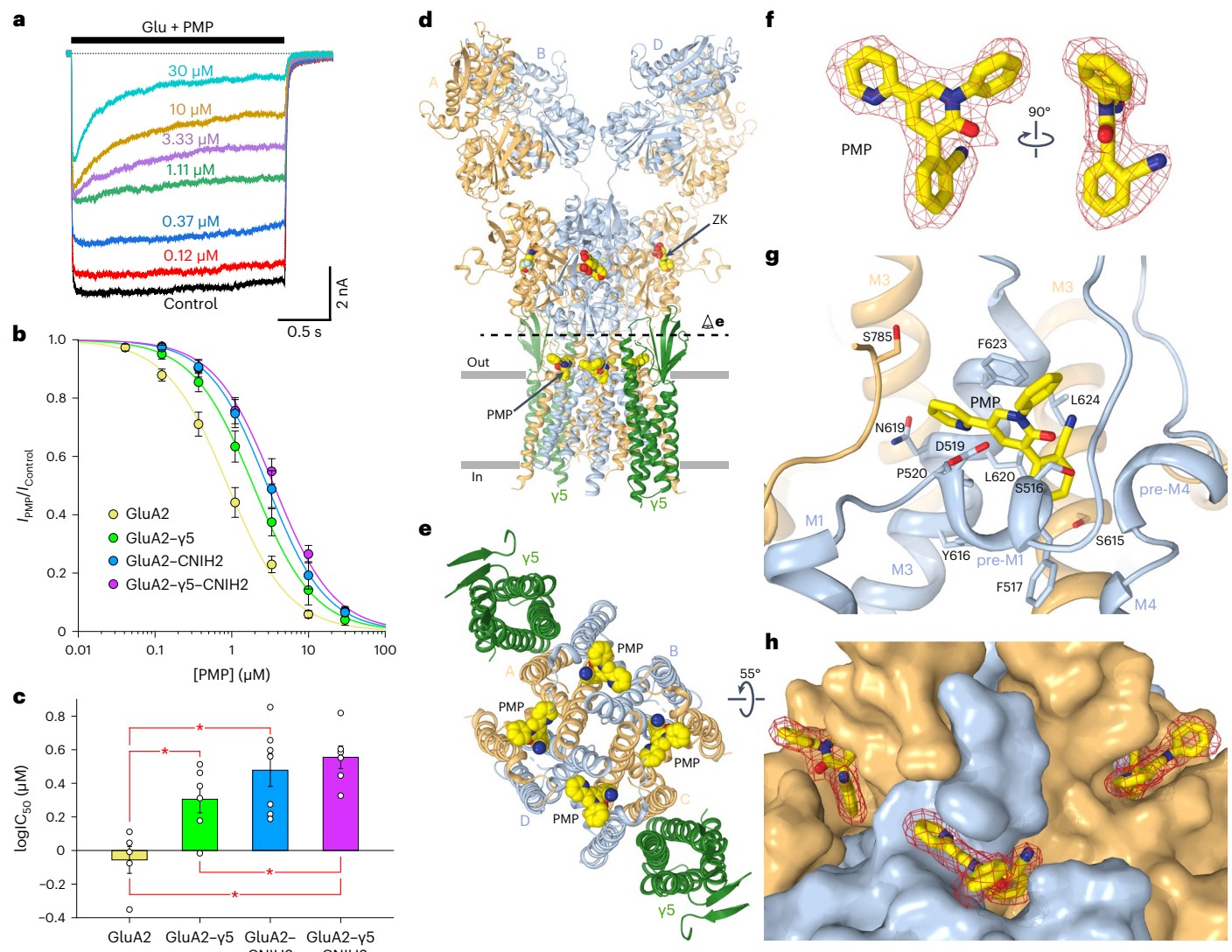

**Fig. 5 | GluA2–γ5_ZK-PMP structure and PMP binding site. a**, Superposition of typical whole-cell currents recorded at −60 mV membrane potential from an HEK293 cell coexpressing GluA2–γ5 and CNIH2 in response to 2 s coapplications of 3 mM Glu and PMP at different concentrations in the continuous presence of 30 μM CTZ. Labels indicate PMP concentrations. Current response to application of Glu in the absence of PMP is labeled as Control. **b**, PMP concentration dependencies for HEK293 cells transfected with GluA2 (yellow), GluA2–γ5 (green), GluA2 and CNIH2 (blue), and GluA2–γ5 and CNIH2 (purple). Curves through the points are logistic equation fits of the normalized current amplitude. The number of independent experiments, $n = 5$ for GluA2, $n = 6$ for GluA2–γ5, $n = 7$ for GluA2 and CNIH2, and $n = 6$ for GluA2–γ5 and CNIH2. Data are mean ± s.e.m. **c**, The $\log IC_{50}$ values calculated for individual cells contributing to the average PMP concentration dependencies in **b**. Asterisks indicate statistically significant differences (two-sided two-sample $t$-test; the significance is assumed if $P < 0.05$). The number of independent experiments, $n = 5$ for GluA2, $n = 6$ for GluA2–γ5,

$n = 7$ for GluA2 and CNIH2, and $n = 6$ for GluA2–γ5 and CNIH2. Probabilities for the two-sided two-sample $t$-test, $P = 0.0117$ for GluA2 versus GluA2–γ5, $P = 0.00245$ for GluA2 versus GluA2 and CNIH2, $P = 2.34 \times 10^{-4}$ for GluA2 versus GluA2–γ5 and CNIH2, $P = 0.204$ for GluA2–γ5 versus GluA2 and CNIH2, $P = 0.0384$ for GluA2–γ5 versus GluA2–γ5 and CNIH2, and $P = 0.537$ for GluA2 and CNIH2 versus GluA2–γ5 and CNIH2. Data are mean ± s.e.m. **d,e**, Structure of GluA2–γ5_ZK-PMP viewed (**d**) parallel to the membrane and (**e**) extracellularly from the level indicated by the dashed line in **d**, with GluA2 subunits colored yellow (subunits A and C) and blue (subunits B and D), γ5 in dark green and molecules of ZK and PMP shown as space-filling models. **f**, Stick model of PMP with cryo-EM density shown as red mesh. **g**, Close-up view of the PMP binding site, with PMP and residues contributing to its binding shown in sticks. **h**, View of PMP binding sites, with the protein shown in surface representation, PMP molecules as stick models and their cryo-EM density as red mesh.

the half-maximal inhibitory concentration ($IC_{50}$) of 3.57 ± 0.25 μM, and the Hill coefficient, $n_{Hill} = 1.11 ± 0.05$ ($n = 6$) (Fig. 5b). Comparison of this concentration dependence with those obtained by recording currents from HEK cells transfected with GluA2 alone ($IC_{50} = 0.89 ± 0.07$ μM, $n_{Hill} = 1.08 ± 0.05$, $n = 5$)[45], GluA2–γ5 alone ($IC_{50} = 1.92 ± 0.07$ μM, $n_{Hill} = 1.09 ± 0.03$, $n = 6$) and cotransfected with GluA2 and CNIH2 ($IC_{50} = 2.92 ± 0.07$ μM, $n_{Hill} = 1.13 ± 0.15$, $n = 7$) suggested that the presence of both γ5 and CNIH2 weakened the potency of PMP inhibition but the effect of CNIH2 appeared to be

stronger. Indeed, a comparison of the $IC_{50}$ values obtained from fitting the concentration dependencies for individual cells (Fig. 5c and Extended Data Table 1) showed that CNIH2 alone caused a stronger increase in $IC_{50}$ (GluA2 versus GluA2 plus CNIH2) than γ5 alone (GluA2 versus GluA2–γ5). While the addition of CNIH2 to GluA2–γ5 (GluA2–γ5 versus GluA2–γ5–CNIH2) caused a significant reduction in PMP potency, reduction of PMP potency in response to addition of γ5 to GluA2–CNIH2 (GluA2–CNIH2 versus GluA2–γ5–CNIH2) did not seem significant (Fig. 5c).

**Table 2 | Cryo-EM data collection, refinement and validation statistics**

| | GluA2–γ5–CNIH2$_{ZK-PMP-SPD}$ LBD–TMD (EMDB-40746) (PDB 8SS7) | GluA2–γ5$_{ZK-PMP}$ FL (EMDB-40747) (PDB 8SS8) | GluA2–γ5$_{ZK-PMP}$ LBD–TMD (EMDB-40748) (PDB 8SS9) | GluA2–γ5–CNIH2$_{Glu-SPD}$ FL (EMDB-40749) (PDB 8SSA) | GluA2–γ5–CNIH2$_{Glu-SPD}$ LBD–TMD (EMDB-40750) (PDB 8SSB) |
|---|---|---|---|---|---|
| **Data collection and processing** | | | | | |
| Magnification | ×75,000 | ×75,000 | ×75,000 | ×105,000 | ×105,000 |
| Voltage (kV) | 300 | 300 | 300 | 300 | 300 |
| Electron exposure (e⁻/Å²) | 50 | 50 | 50 | 58 | 58 |
| Defocus range (µm) | −1 to −2 | −1 to −2 | −1 to −2 | −1 to −2 | −1 to −2 |
| Pixel size (Å) | 0.925 | 0.925 | 0.925 | 0.83 | 0.83 |
| Symmetry imposed | C2 | C2 | C2 | C2 | C2 |
| Initial particle images (no.) | 2,360,956 | 2,360,956 | 2,360,956 | 2,282,408 | 2,282,408 |
| Final particle images (no.) | 126,263 | 126,964 | 117,939 | 58,186 | 48,434 |
| Map resolution (Å) | 2.76 | 2.81 | 2.72 | 3.88 | 3.66 |
| FSC threshold | 0.143 | 0.143 | 0.143 | 0.143 | 0.143 |
| Map resolution range (Å) | 2.6/3.3/34.4 | 2.0/3.3/36.3 | 2.0/3.3/36.0 | 2.1/5.8/42.5 | 2.1/4.8/36.5 |
| **Refinement** | | | | | |
| Initial model used (PDB code) | 7RZ5, 7OCE | 7RZ5, 7OCE | 7RZ5, 7OCE | 7RZ5, 7OCE | 7RZ5, 7OCE |
| Model resolution (Å) | 2.76 | 2.81 | 2.72 | 3.88 | 3.66 |
| FSC threshold | | | | | |
| Map sharpening B factor (Å²) | −132.7 | −135.5 | −131.4 | −101.4 | −97.2 |
| Model composition | | | | | |
| Nonhydrogen atoms | 19,436 | 28,487 | 16,319 | 30,888 | 19,048 |
| Protein residues | 2,266 | 3,514 | 1,986 | 3,840 | 2,304 |
| Ligands | | | | | |
| PMP | 4 | 4 | 4 | – | – |
| SPD | 1 | – | – | 1 | 1 |
| ZK | 4 | 4 | 4 | – | – |
| Glutamate (Glu) | – | – | – | 4 | 4 |
| B factors (Å²) | | | | | |
| Protein | 63.09 | 68.50 | 65.55 | 128.43 | 68.50 |
| Ligand | 41.07 | 98.06 | 65.93 | 66.30 | 98.06 |
| R.m.s. deviations | | | | | |
| Bond lengths (Å) | 0.010 | 0.014 | 0.008 | 0.010 | 0.007 |
| Bond angles (°) | 1.657 | 1.585 | 1.541 | 1.431 | 1.376 |
| **Validation** | | | | | |
| MolProbity score | 1.92 | 1.75 | 1.51 | 1.96 | 2.01 |
| Clashscore | 6.95 | 3.86 | 4.43 | 6.33 | 7.27 |
| Poor rotamers (%) | 1.45 | 1.33 | 0.36 | 1.00 | 0.72 |
| Ramachandran plot | | | | | |
| Favored (%) | 93.80 | 92.19 | 95.80 | 87.63 | 87.37 |
| Allowed (%) | 5.66 | 7.12 | 3.89 | 11.37 | 11.40 |
| Disallowed (%) | 0.54 | 0.69 | 0.31 | 1.00 | 1.24 |

To gain insight into the structural mechanism of PMP inhibition in the presence of γ5 and CNIH2, we supplemented the purified protein with 100 µM ZK and 100 µM PMP and subjected it to cryo-EM. Data processing revealed two populations of particles (Supplementary Fig. 3). The first group produced a 3D reconstruction with density for two γ5 subunits per AMPAR tetramer and yielded a GluA2–γ5$_{ZK-PMP}$ structure

(Fig. 5d,e, Supplementary Figs. 1 and 4 and Tables 1 and 2), grossly similar to the GluA2–γ5$_{ZK}$ structure published previously[29]. However, compared to GluA2–γ5$_{ZK}$, GluA2–γ5$_{ZK-PMP}$ revealed four molecules of PMP bound at the ion channel extracellular collar, at the location predicted by the crystal structure of the PMP-bound receptor alone, GluA2$_{PMP}$ (ref. 45). Because of the much higher resolution of GluA2–γ5$_{ZK-PMP}$ compared to

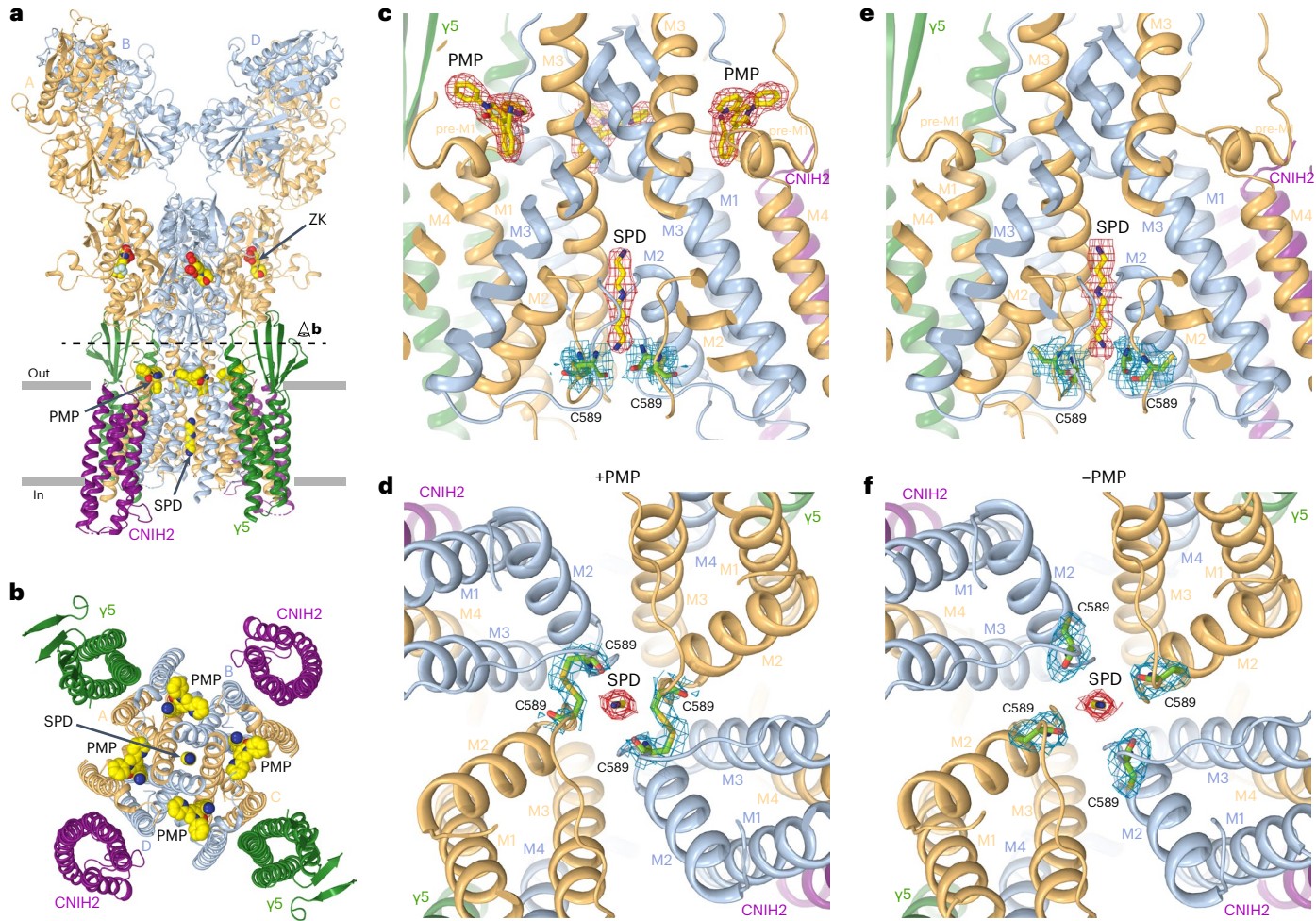

**Fig. 6 | GluA2−γ5−CNIH2_ZK-PMP-SPD structure and asymmetry of the intracellular pore entrance. a,b,** Structure of GluA2−γ5−CNIH2_ZK-PMP-SPD viewed (**a**) parallel to membrane and (**b**) extracellularly from the level indicated by the dashed line in **a**, with GluA2 subunits colored yellow (subunits A and C) and blue (subunits B and D), γ5 in dark green, CNIH2 in purple and molecules of ZK, PMP and SPD shown as space-filling models. **c–f,** Parallel to membrane cutoff (**c,e**) and intracellular (**d,f**) views of the ion channel in the GluA2−γ5−CNIH2_ZK-PMP-SPD (**c** and **d**) and GluA2−γ5−CNIH2_ZK-SPD (**e** and **f**) structures, with the PMP and SPD molecules and cysteines C589 shown in sticks and the corresponding cryo-EM density as red and blue mesh, respectively.

GluA2_PMP (Fig. 5f and Extended Data Fig. 8), the positions and orientations of PMP molecules in the GluA2−γ5_ZK-PMP binding pockets were determined unambiguously. The pockets are mainly hydrophobic, mostly enclosed within individual GluA2 protomers and contributed by residues P512, S516, F517, D519, P520, Y616, N619, L620, F623 and L624 of one subunit and S615 of the neighboring subunit (Fig. 5g). Consistent with the critical role of S516, F517, P520, S615 and F623 in PMP binding, alanine substitutions of these residues caused a substantial weakening of the PMP inhibitory potency[45]. The molecules of PMP, which snugly fit into the extracellular collar binding pockets (Fig. 5h), are likely acting as wedges that prevent conformational changes within this region during AMPAR gating[3,4,24].

Reconstruction from the second population of particles revealed densities for two γ5 and two CNIH2 subunits around the GluA2 TMD, molecules of ZK bound to LBD clamshells, PMP at the extracellular collar and SPD in the channel selectivity filter (Extended Data Fig. 8 and Supplementary Fig. 4). The resulting GluA2−γ5−CNIH2_ZK-SPD-PMP structure (Fig. 6a,b) is similar to GluA2−γ5_ZK-PMP (Fig. 5d,e), except that the latter lacks two CNIH2 subunits and SPD. Comparison of these structures suggested that binding of CNIH2 to GluA2−γ5 complex causes slight splaying of the LBD dimers, clockwise rotation of the ATD layer, movement of γ5 away from the channel core and counterclockwise displacement of the M4 segments (Extended Data Fig. 9a–i). Nevertheless, all these changes are minor, and do not affect conformations of individual LBD clamshells (Extended Data Fig. 9j,k), LBD dimers (Extended Data Fig. 9l) or the closed channel pore (Extended Data Fig. 4).

GluA2−γ5−CNIH2_ZK-SPD-PMP showed apparent asymmetry of the intracellular portion of the channel pore, which has not been observed in AMPAR structures before. Cysteines C589 formed two pairs of disulfide bridges clearly detected in cryo-EM density, illustrating the twofold symmetry of the channel intracellular entrance (Fig. 6c,d). Crosslinking of C589 cysteines is unlikely to have physiological relevance because of reducing conditions inside the cell that prevent disulfide bond formation. Besides, the alanine substitution of C589 did not produce substantial changes in AMPAR function[2]. Crosslinking of C589 cysteines was not observed in the GluA2−γ5_ZK-PMP structure (Fig. 5) or in the absence of PMP (GluA2−γ5−CNIH2_ZK-SPD; Fig. 6e,f), indicating that CNIH2 and PMP together are the two factors that cause strong deviation of the AMPAR inner pore from the fourfold rotational symmetry, likely due to increased protein flexibility in this region. While CNIH2 is in close proximity to the selectivity filter-forming M2 and can influence M2 dynamics directly through a single lipid molecule that separates them, PMP is distal from the inner pore. Being in the ion channel extracellular collar, PMP can influence

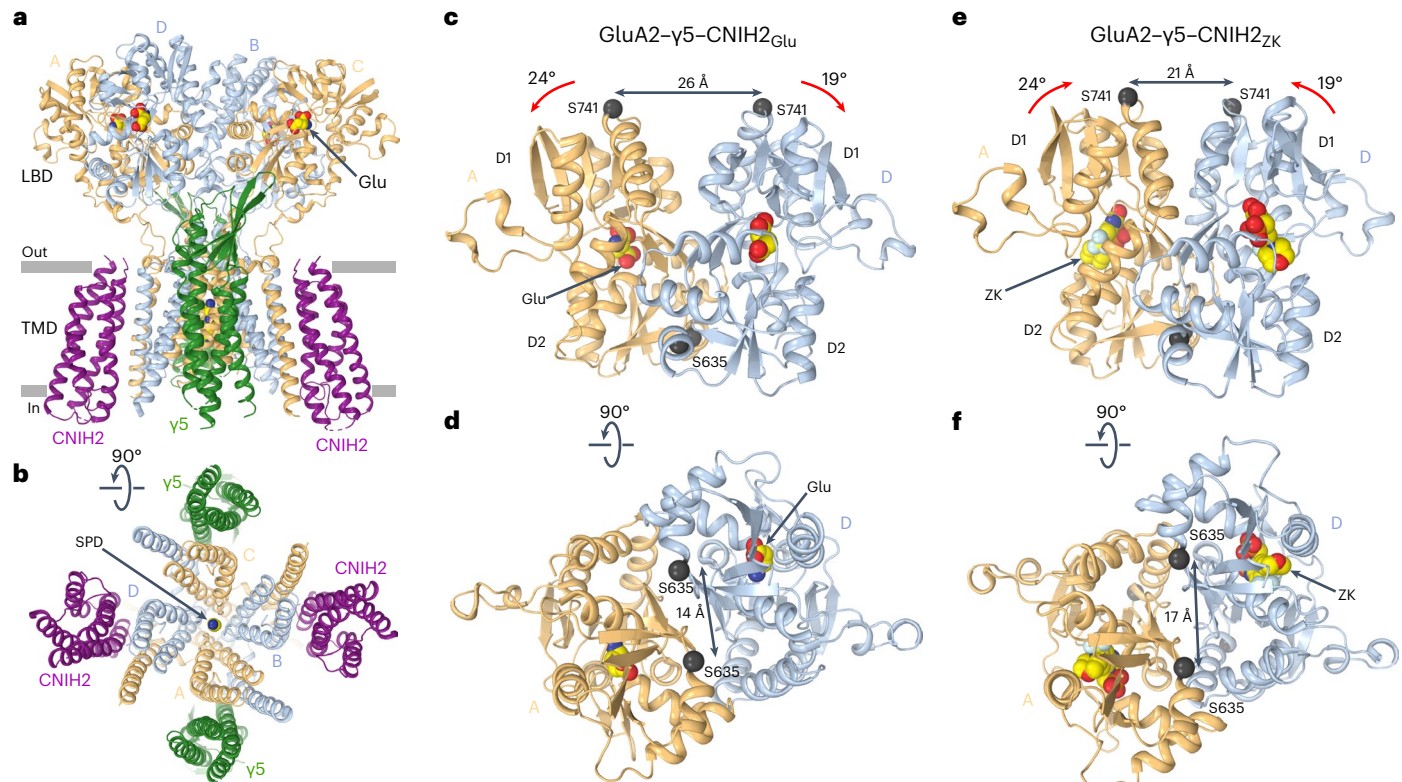

**Fig. 7 | Desensitized-state structure GluA2–γ5–CNIH2_Glu. a,b**, Structure of GluA2–γ5–CNIH2_Glu viewed parallel to the membrane (**a**) and intracellularly (**b**), with GluA2 subunits colored yellow (subunits A and C) and blue (subunits B and D), γ5 in dark green, CNIH2 in purple and molecules of Glu shown as space-filling models. **c–f**, Side (**c,e**) and bottom (**d,f**) views of the LBD dimer in the desensitized-state GluA2-γ5-CNIH2_Glu (**c** and **d**) and closed-state GluA2-γ5-

CNIH2_ZK (**e** and **f**) structures, with the Glu and ZK molecules shown as space-filling models and Cα atoms for residues S635 and S741 as dark spheres. Red arrows illustrate rotation of the upper lobe D1 relative to lower lobe D2 in the desensitized versus closed states. Double arrows indicate the distances between Cα atoms of S635 or S741.

the selectivity filter dynamics allosterically, through the M1, M3 and M4 helices. Accordingly, increased selectivity filter dynamics can cause a reduction in PMP affinity to GluA2–γ5–CNIH2_ZK-SPD-PMP (Fig. 5b,c). Alternatively, because of the strong dependence of PMP inhibition on AMPAR gating[3,4,24], CNIH2 may induce the rightward shift in PMP concentration dependence by influencing gating equilibrium. The last hypothesis is supported by strong effects of CNIH2 on deactivation (Fig. 2b) and desensitization (Fig. 2c,d).

### CNIH2 weakens desensitization by changing LBD dimers
To study the effect of CNIH2 on gating equilibrium in the AMPAR–γ5 complex, we solved the GluA2–γ5–CNIH2 structure in the desensitized state. The purified protein was supplemented with 10 mM Glu and subjected to cryo-EM. Again, two particle populations were observed (Supplementary Fig. 5). The first population resulted in the GluA2–γ5_Glu structure, identical to the one published before[29], while the second population yielded the GluA2–γ5–CNIH2_Glu structure (Fig. 7a,b). Compared to the ZK-bound structures, the individual LBD clamshells in GluA2–γ5–CNIH2_Glu were 24° (subunits A, C) and 19° (subunits B, D) more closed, the LBD dimers showed increased separation of the upper D1 lobes and reduced separation of the D2 lobes (Fig. 7c–f), while the channel showed no gate opening (Extended Data Fig. 4). All these observations are consistent with GluA2–γ5–CNIH2_Glu representing the desensitized state[3,4,29]. To understand how CNIH2 reduces the extent of desensitization and slows the rate of desensitization (Fig. 2), we superposed the desensitized-state GluA2–γ5_Glu and GluA2–γ5–CNIH2_Glu structures (Fig. 8).

GluA2–γ5–CNIH2_Glu appears to have a slightly more compact TMD than GluA2–γ5_Glu, emphasized by small shifts of the γ5 subunits and M4

segments toward the pore center in the presence of CNIH2 (Fig. 8a–k). This TMD constriction is associated with the movement of the β1–β2 loop in the extracellular head domain of γ5 toward the LBDs of subunits A and C (Fig. 8h), which in turn causes an overall rotation of the LBD dimer of dimers (Fig. 8g) and changes conformations of individual LBDs and LBD dimers. Indeed, the individual LBD clamshells are roughly 1° more closed in subunits A and C (Fig. 8l) and roughly 3° more closed in subunits B and D (Fig. 8m). These changes cause a roughly 3 Å reduction in separation of the D1 lobes within LBD dimers (Fig. 8n). Since rupture of the D1–D1 interface is associated with AMPAR desensitization[29,38,46,47], the reduced separation of the D1 lobes appears to be a mechanism underlying weakening of desensitization.

## Discussion
We studied the effects of two different auxiliary subunits, potentiating CNIH2 and inhibitory γ5, on structure and function of homotetrameric AMPARs composed of GluA2 subunits. Structures of GluA2–γ5–CNIH2 complexes were determined by transducing HEK293S GnTI⁻ cells with GluA2–γ5 baculovirus and serendipitously inducing coexpression of endogenous human CNIH2, previously identified in cultured HEK cells by transcriptome analysis[48]. Expression of endogenous CNIH2 appears to be weaker than expression of the engineered GluA2–γ5 fusion construct, as evidenced by the presence of GluA2–γ5 complexes in all collected cryo-EM datasets (Extended Data Fig. 1 and Supplementary Figs. 2, 3 and 5). At the same time, patch-clamp recordings of currents through the plasma membrane of HEK293 cells transfected with GluA2 or GluA2–γ5 fusion individually or together with CNIH2 reported function of GluA2, GluA2–γ5, GluA2–CNIH2 and GluA2–γ5–CNIH2 channels separately, as evidenced by statistically different results obtained for

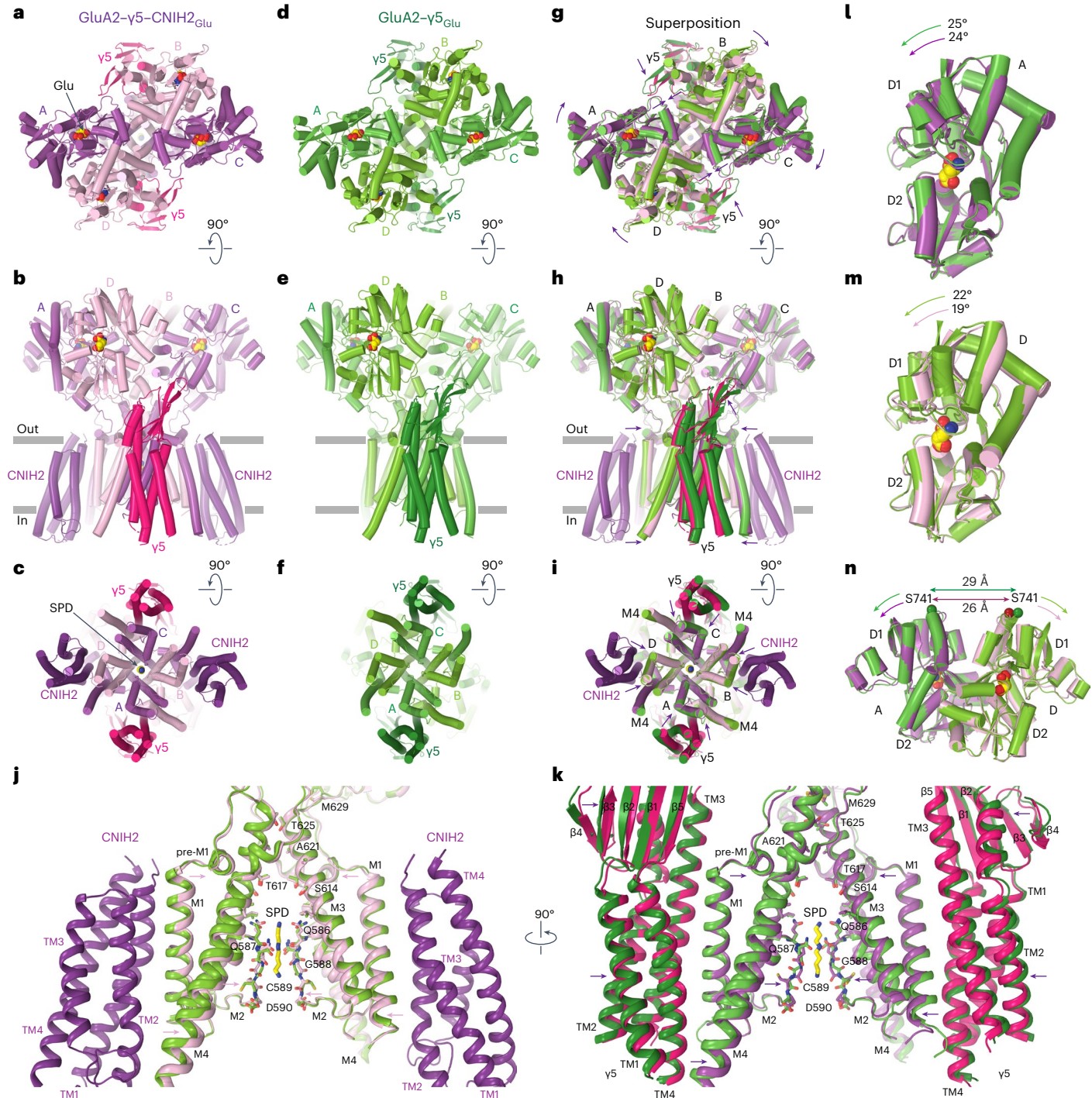

**Fig. 8 | CNIH2 reduces separation of the upper D1 lobes in LBD dimers during desensitization. a–i**, Structures of GluA2–γ5–CNIH2_Glu (**a–c**), GluA2–γ5_Glu (**d–f**) and their superposition based on the GluA2 TMD (**g–i**) viewed extracellularly (**a,d,g**), parallel to the membrane (**b,e,h**) and intracellularly (**c,f,i**), with molecules of Glu and SPD shown as space-filling models. Relative movement of domains upon CNIH2 binding is illustrated with purple arrows. **j,k**, GluA2–γ5–CNIH2_Glu and GluA2–γ5_Glu TMD superposition viewed parallel to the membrane from two perpendicular directions, with GluA2 subunits A and C and γ5 molecules (**j**) and GluA2 subunits B and D and CNIH2 molecules (**k**) omitted for clarity.

The pore-lining residues and the molecule of SPD are shown as sticks. Arrows indicate tightening of the TMD in the presence of CNIH2. **l,m**, D2 lobe-based superposition of subunits A (**l**) and D (**m**) LBDs from GluA2–γ5–CNIH2_Glu (purple) and GluA2–γ5_Glu (green) structures. Arrows show rotation of the upper lobe D1 toward lower lobe D2 relative to the close-state structures GluA2–γ5–CNIH2_ZK and GluA2–γ5_ZK. **n**, D2 lobe-based superposition of subunits A/D LBD dimers from GluA2–γ5–CNIH2_Glu (purple) and GluA2–γ5_Glu (green) structures. Double arrows indicate the distance between Cα atoms of S741.

each type of transfected cells (Figs. 2, 3 and 5). Why did electrophysiological experiments not show signs of endogenous CNIH2 expression? Different cell lines and methods of expression (baculovirus versus transfection with DNA) could explain the discrepancies. Alternatively, most complexes with endogenous CNIH2 may reside in intracellular compartments instead of the plasma membrane. These complexes may still be purified for structural studies but would not contribute to recorded currents. Expression of endogenous CNIH2 has been only observed in

cells expressing the GluA2–γ5 fusion and can be strongly suppressed by addition of AMPAR antagonists to the expression media[29]. What is special about the GluA2–γ5 fusion construct and whether it contains a specific signal for CNIH2 expression remains to be determined.

Compared to the inhibitory TARP γ5, which strengthens desensitization by increasing the fraction of desensitized channels and slowing the rate of recovery from desensitization[29], the potentiating auxiliary subunit CNIH2 weakens desensitization by reducing the fraction of desensitized channels and slowing the rate of desensitization (Fig. 2). When expressed in cells together, γ5 and CNIH2 appear to cancel each other's effects on the maximum number of desensitized channels (Fig. 2d). Nevertheless, CNIH2 appears to have a stronger effect on the rate of desensitization (Fig. 2c), while γ5 on the rate of recovery from desensitization (Fig. 2f). Whereas the deceleration of recovery from desensitization by γ5 was proposed to be due to interaction of the γ5 extracellular domain with AMPAR LBDs[29], CNIH2 appears to weaken desensitization by reducing the separation of the upper lobes in LBD dimers (Fig. 8). How CNIH2, which contacts the AMPAR TMD only, exerts an allosteric effect on the LBDs remains unknown. It may influence AMPAR gating equilibrium, consistent with CNIH2 effect of the rate of deactivation (Fig. 2b). However, uncovering a structural basis of this allosteric action may require solving open-state GluA2–γ5 structures in the presence and absence of CNIH2.

Our structures of GluA2–γ5–CNIH2 have also revealed binding of polyamine SPD in the AMPAR channel selectivity filter (Fig. 3). The polyamine block is a ubiquitous feature of $Ca^{2+}$-permeable AMPARs, characterized by inward current rectification[41–43,49]. Apart from intracellular polyamines, which represent physiological blockers[50–52], AMPAR channels undergo extracellular block by polyamine- or acylpolyamine-containing toxins and their natural or synthetic analogs[53–61]. Given that polyamine tails of these extracellular blockers were shown to interact with the channel selectivity filter, this region was proposed to bind intracellular polyamines as well[25]. The near-perfect overlap of SPD and SPD-like moiety of NASPM (Fig. 4c) and MS data strongly support this idea. A paradox to be resolved is that CNIH2, which appears to help stabilize SPD binding in the closed state (Fig. 3a–d), weakens polyamine block of open channels (Fig. 3e–g). One possible explanation is that channel opening is accompanied by substantial conformational changes of the selectivity filter, as is widening demonstrated by the comparison of GluA2–γ2$_{Glu+CTZ+NASPM}$ and GluA2–γ5–CNIH2$_{ZK-SPD}$ (Fig. 4c), which may lead to lowering of SPD affinity and, as a result, reduced current rectification.

Finally, we found that both γ5 and CNIH2 reduced the PMP inhibitory potency, although, similar to the polyamine block, the effect of CNIH2 appeared to be stronger (Fig. 5a–c). We identified four PMP binding sites in the ion channel extracellular collar (Figs. 5d–h and 6), a hub for regulation of iGluR gating[39,62,63]. These sites were originally identified in the crystal structures of AMPAR alone[45], but the high quality and resolution of our cryo-EM reconstructions provide much more precise positions and orientations of PMP molecules. Reduced potency of PMP inhibition by γ5 and CNIH2 (increase in IC$_{50}$ from roughly 1 μM for the receptor alone to 2–4 μM for the complexes; Fig. 5a–c and Extended Data Table 1) preserved occupancy of all four extracellular collar binding sites. For example, even lower potency (IC$_{50}$ of 58 μM) was proposed to result in only two of four extracellular collar sites occupied by PMP in GluK2 kainate receptors[64]. Given the substantial remodeling of the extracellular collar sites upon channel opening due to kinking of the pore-lining M3 helices[24,65], PMP acts as a wedge that binds to these sites in the closed state and prevents channel opening[3,4,45]. Despite the reduction of PMP potency in the presence of CNIH2 (Fig. 5a–c), the PMP-bound closed-state structures solved in the presence and absence of CNIH2 appeared to be similar (Extended Data Fig. 9). CNIH2 likely reduces the potency of PMP inhibition by influencing AMPAR gating equilibrium, supported by conformational rearrangements in the LBD layer (Extended Data Fig. 9g,h), movement of the

transmembrane helices (Extended Data Fig. 9i) and altered flexibility of the selectivity filter that results in C589 crosslinking (Fig. 6c,d). Better understanding of allosteric regulation of AMPAR noncompetitive inhibition by CNIH2 will help design next-generation drugs targeting AMPAR synaptic complexes.

## Online content

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

## Methods

### Constructs for large-scale protein expression

The fusion construct GluA2–γ5 (ref. 29) was prepared by introducing a GTG linker between GluA2 and γ5, where GluA2 represents a modified rat GluA2$_{flip}$ subunit (UniProt ID P19491, Q586 at the Q/R site), which was previously called GluA2* (ref. 66) and used for making GluA2–γ2 and GluA2–GSG1L fusion constructs[24,38]. γ5 is a C-terminally truncated mouse subunit (Cacng5, UniPort ID Q8VHW4, residues S2-E207), for which DNA was provided by B. Fakler. GluA2–γ5 was introduced into BacMam vector for baculovirus-based protein expression in mammalian cells[67], with the C-terminal thrombin cleavage site (LVPRG), followed by enhanced GPF (eGFP) and streptavidin affinity tag (WSHPQFEK).

### Protein expression and purification

GluA2–γ5 bacmid and baculovirus were made using standard methods described previously[29,67]. The P2 virus stock produced in Sf9 cells (Gibco, 12659017) was added to HEK293S GnTI⁻ cells (ATCC, CRL-3022) incubated in Freestyle media at 37 °C and 5% $CO_2$. Twelve hours post-transduction, the cells were supplemented with 10 mM sodium butyrate and the temperature was changed to 30 °C. The cells were collected 72 h after transduction by centrifugation (5,500$g$, 10 min), washed with PBS (pH 8.0) and pelleted again (5,500$g$, 15 min). The cell pellet was resuspended in ice-cold lysis buffer, containing 150 mM NaCl, 20 mM Tris pH 8.0, 1 mM β-mercaptoethanol (βME), 0.8 μM aprotinin, 2 μg ml⁻¹ leupeptin, 2 μM pepstatin A and 1 mM phenylmethylsulfonyl fluoride and 2 μM ZK. Cells were subsequently lysed using a Misonix Sonicator under constant stirring on ice. The lysate was centrifuged (9,900$g$, 15 min) to remove unbroken cells and cell debris, and the supernatant was subjected to ultracentrifugation (186,000$g$, 40 min) to pellet the cell membranes. The membrane pellet was mechanically homogenized and solubilized at 4 °C in the buffer containing 150 mM NaCl, 20 mM Tris-HCl pH 8.0, 1 mM βME, 0.1% cholesteryl hemisuccinate Tris salt (CHS) and 1% n-dodecyl-β-D-maltoside (Anatrace, 4211929). After 2–3 h of solubilization, insoluble material was removed by ultracentrifugation (186,000$g$, 40 min), the supernatant was added to streptavidin-linked resin (2 ml resin per 1 l of the initial cell culture) and the mixture was rotated for 10–14 h at 4 °C. The protein-bound resin was washed with 25 ml of the buffer containing 150 mM NaCl, 20 mM Tris-HCl pH 8.0, 0.005% CHS and 0.05% digitonin and 1 mM βME, and the protein was eluted using the same buffer supplemented with 2.5 mM D-desthiobiotin. To remove eGFP and the streptavidin affinity tag, the eluted protein was subjected to thrombin digestion (1:100 w/w) for 1.5 h at 22 °C. The digest reaction was injected into Superose 6 10/30 GL size-exclusion chromatography column (GE Healthcare) equilibrated with the buffer containing 150 mM NaCl, 20 mM Tris-HCl pH 8.0, 0.005% CHS and 0.05% digitonin and 1 mM βME. The peak fractions corresponding to the tetrameric complex were pooled, concentrated to roughly 4–5 mg ml⁻¹ and used for cryo-EM sample preparation. All the steps, unless otherwise noted, were performed at 4 °C.

### Cryo-EM sample preparation and data collection

UltrAuFoil R 0.6/1.0 Au 300 mesh gold grids were used for cryo-EM sample preparation. The grids were plasma treated with Pelco easiGlow cleaning system (Ted Pella, 25 s, 15 mA) immediately before sample application to make their surfaces hydrophilic. Purified protein was supplemented with 100 μM ZK, 10 mM Glu or 100 μM PMP and incubated for 30 min on ice before making the grids. An FEI Vitrobot Mark IV (Thermo Fischer Scientific) was used to plunge-freeze the grids after application of 3 μl protein solution at 4 °C, 100% humidity, with a blot time of 3 s, wait time of 30 s and blot force of 2. The grids were imaged using 300-kV Titan Krios microscope equipped with a Falcon 4 or Gatan K3 direct electron detector with postcolumn GIF Quantum energy filter with the slit width set to 20 eV. We collected 7,701 images (0.83 Å per pixel) in the presence of ZK alone, 18,161 images (1.10 Å per pixel) in the absence of ligands (apo state), 16,808 images (0.946 Å per pixel) in the presence of ZK and PMP and 9,943 images (0.83 Å per pixel) in the presence of Glu in counting mode across the defocus range of −0.8 to −2.0 μm.

### Image processing

GluA2–γ5–CNIH2$_{ZK}$ and GluA2–γ5–CNIH2$_{Glu}$ processing was done in Relion and cryoSPARC[68]. Initially, frame alignment was performed using MotionCor2 (ref. 69) and contrast transfer function (CTF) estimation was performed using GCTF on nondose-weighted micrographs, while subsequent data processing was done on dose-weighted micrographs. Initially, roughly 1,000 particles were picked manually for Topaz training and autopicking using Topaz in Relion[70]. The particles were then extracted as binned and exported to cryoSPARC for cleanup by two-dimensional (2D) classification and heterogeneous refinement. Finally, the clean particles were re-extracted in Relion and subjected to Bayesian polishing and CTF refinement. The consensus refinement yielded a map with apparent density for the receptor and auxiliary subunits. Finally, ATD and micelle signal subtraction improved the density of the LBD–TMD region.

Processing of data for GluA2–γ5–CNIH2$_{SPD}$ and GluA2–γ5$_{apo}$ collected in apo condition (no added ligand) was done in cryoSPARC v.4.0. Patch motion correction and CTF estimation were performed, after which roughly 10,000 particles were picked using reference-free Gaussian autopicker to generate 2D classes that were then used as templates to autopick 3,730,352 particles. Particles were extracted with 2 × 2 times binning and subjected to multiple rounds of 2D classification, yielding 203,710 particles that were re-extracted unbinned. Further 3D classification without alignment produced two main classes of particles, resulting in GluA2–γ5$_{apo}$ (81,879 particles) and GluA2–γ5–CNIH2$_{SPD}$ (121,812 particles) structures. These particles were independently subjected to local refinements focusing on the LBD–TMD region of GluA2–γ5$_{apo}$ and GluA2–γ5–CNIH2$_{SPD}$ (Supplementary Fig. 2).

Frame alignment for the data collected in the presence of PMP was performed using on-the-fly MotionCor2, while CTF estimation was performed using patch CTF in cryoSPARC. The remaining processing steps were carried out using cryoSPARC v.4.0 (Supplementary Fig. 3). Initially, roughly 500 micrographs were randomly selected for blob picking to generate 2D classes that were used in autopicking. A total of 2,360,956 particles were picked and extracted as binned and subjected to 2D classification. A set of particles matching the shape of AMPAR was selected for ab initio 3D reconstruction. The binned particles were cleaned up rigorously by 2D classification and heterogenous refinement. The clean particles were re-extracted as unbinned and classified. At this step, we separated the pool of particles without CNIH2 density in the TMD from the one that contained both CNIH2 and γ5. Both sets of particles were then processed independently. After several rounds of particle cleanup and nonuniform refinement, we obtained maps of GluA2–γ5$_{ZK-PMP}$ and GluA2–γ5–CNIH2$_{ZK-PMP}$. ATD and micelle signal subtraction drastically improved the density for the LBD–TMD regions. The resolution of the maps is listed in Tables 1 and 2.

### Model building and refinement

For all structures, molecular models of GluA2, γ5 and CNIH2 subunits were built in COOT[71] using cryo-EM density as well as structures GluA2–γ5$_{ZK}$ (Protein Data Bank (PDB) ID 7RZ5) and GluA1/A2–γ8–CNIH2 (PDB ID 7OCE) as guides. The resulting models were real space refined in Phenix[72] and visualized in Chimera[73] or Pymol[74]. Domain rotations were determined using the DynDom server (http://dyndom.cmp.uea.ac.uk/dyndom/).

### Patch-clamp recordings

DNA encoding GluA2–γ5 fusion (described above) and human CNIH2 were introduced into a pIRES plasmid for expression in eukaryotic cells that were engineered to produce GFP via a downstream internal

ribosome entry site[66]. HEK293 cells (ATCC, catalog no. CRL-1573) grown on glass coverslips in 35 mm dishes were transiently transfected with 1–5 μg of the plasmid DNA using Lipofectamine 2000 Reagent (Invitrogen). Recordings were made 24 to 96 h after transfection at room temperature. Currents from whole cells, typically held at a −60 mV potential, were recorded using Axopatch 200B amplifier (Molecular Devices, LLC), filtered at 5 kHz, and digitized at 10 kHz using low-noise data acquisition system Digidata 1440A and pCLAMP software (Molecular Devices, LLC). The external solution contained (in mM): 140 NaCl, 2.4 KCl, 4 CaCl₂, 4 MgCl₂, 10 HEPES pH 7.3 and 10 glucose; 7 mM NaCl was added to the extracellular activating solution containing 3 mM Glu to improve visualization of the border between two solutions coming out of a two-barrel theta glass pipette, which allowed its more precise positional adjustment for faster solution exchange. The internal solution contained (in mM): 150 CsF, 10 NaCl, 10 EGTA and 20 HEPES pH 7.3. Rapid solution exchange was achieved by mounting the two-barrel theta glass pipette on a piezoelectric translator. Typical 10–90% rise times were 200–300 μs, as measured from junction potentials at the open tip of the patch pipette after recordings. Data analysis was performed using Origin v.9.1.0 software (OriginLab Corp.). Recovery from desensitization recorded using the two-pulse protocol was fitted with the Hodgkin–Huxley equation[75]: $I = (I_{max}^{1/m} - (I_{max}^{1/m} - 1) \times \exp(-t/\tau_{RecDes}))^m$, where $I$ is the peak current at a given interpulse interval, $t$, $I_{max}$ is the peak current at long interpulse intervals, $\tau_{RecDes}$ is the recovery time constant and $m$ is an index that corresponds to the number of kinetically equivalent rate-determining transitions that contribute to the recovery time course.

## MS

### Extraction of metabolites from concentrated purified protein.

Before extraction, samples were moved from −80 °C storage to wet ice and thawed. Extraction buffer consisting of 1% formic acid (Millipore Sigma) and 4.5 μM metabolomics amino acid mix standard (Cambridge Isotope Laboratories, Inc.), was prepared and chilled on dry ice. A cocktail of polyamine standards containing 10 μM Spermine, 10 μM SPD, 10 μM N-acetylspermine and 10 μM Putrescine was prepared in advance of the extraction (Sigma Aldrich). The samples and cocktail were extracted by mixing 10 μl of sample with 10 μl of extraction buffer in 2 ml liquid chromatography (LC) vials containing a 250 μl glass insert. All vials were then incubated at 60 °C in a heat-block for 30 min, followed by addition of 80 μl of warm (60 °C) 100% methanol to the individual samples. The glass inserts were then transferred to microfuge carrier tubes and centrifuged at 3,000g for 20 min at 4 °C. Supernatant (20 μl) was then transferred to a new LC vial containing a new glass insert for LC–MS analysis and the remaining sample was placed in −80 °C for long-term storage.

### LC–MS/MS.

Samples were subjected to an LC–MS analysis to detect and quantify polyamine targets and other metabolites and features. The LC–MS parameters were adapted from a previously described method[76]. The LC column was a WatersTM BEH-Phenyl (2.1 × 150 mm, 1.7 μm) coupled to a Dionex Ultimate 3000TM system and the column oven temperature was set to 25 °C for the gradient elution. A flow rate of 200 μl min⁻¹ was used with the following buffers; A) 0.1% formic acid in water, and B) 0.1% formic acid in acetonitrile. The gradient profile was as follows; 0–35%B (0–10 min), 35–75%B (10–15 min), 75-99%B (15–15.25 min), 99–99%B (15.25–16.5 min), 99–0%B (16.5–16.75 min) and 0–0%B (16.75–20 min). Injection volume was set to 2 μl for all analyses (20 min total run time per injection). MS analyses were carried out by coupling the LC system to a Thermo Q Exactive HFTM mass spectrometer operating in heated electrospray ionization mode. Method duration was 20 min with a polarity switching data-dependent top five method for both positive and negative modes. Spray voltage for both positive and negative modes was 3.5 kV and capillary temperature was set to 320 °C with a sheath gas rate of 35, auxiliary gas of 10 and

maximum spray current of 100 μA. The full MS scan for both polarities used 120,000 resolution with an AGC target of $3 \times 10^6$ and a maximum injection time of 100 ms, and the scan range was from 95–1,000 m/z. Tandem MS spectra for both positive and negative mode used a resolution of 15,000, AGC target of $1 \times 10^5$, maximum injection time of 50 ms, isolation window of 0.4 m/z, isolation offset of 0.1 m/z, fixed first mass of 50 m/z and three-way multiplexed normalized collision energies of 10, 35 and 80. The minimum AGC target was $1 \times 10^4$ with an intensity threshold of $2 \times 10^5$. All data were acquired in profile mode.

### Identification and relative quantification of metabolites.

The resulting ThermoTM RAW files were converted to SQLite format using an in-house python script to enable downstream peak detection and quantification. The available tandem MS (MS/MS) spectra were first searched against the NIST17MS/MS (ref. 77), METLIN[78] to putatively identify candidate metabolites. The polyamine targets were automatically identified by the analysis but were manually checked against the authentic standard controls to verify the observed retention time, polarity and ion type. The peak heights for each putative metabolite hit were extracted from the sqlite3 files based on the metabolite retention time ranges and accurate masses in the above merged metabolite list. Metabolite peaks were extracted based on the theoretical m/z of the expected ion type for example, [M + H]+, with a 15 ppm tolerance and a ±0.2 min peak apex retention time tolerance within an initial retention time search window of ±0.5 min across the study samples for each batch. The resulting data matrix of metabolite intensities for all samples and blank controls was processed with an in-house python script and final peak detection was calculated based on a signal to noise ratio (S/N) of three times compared to blank controls, with a floor of 10,000 arbitrary units. For samples where the peak intensity was lower than the blank threshold, metabolites were annotated as not detected and were imputed with either the blank threshold intensity for any statistical comparisons to enable an estimate of the fold change as applicable, or zeros for median metabolite intensity calculation within a sample. For all group-wise comparisons, t-tests were performed with the Python SciPy (v.1.5.4)[79] library to test differences and statistics generated for downstream analyses. GraphPad Prism 9 (v.9.4.1) was used for all volcano plots. Extracted ion chromatograms were processed manually using Thermo Qual BrowserTM (v.4.0.27.19) and re-plotted in Graphpad.

### Statistics and reproducibility

Statistical analysis (Figs. 2b–d, 3g and 5c) was performed using Origin v.9.1.0 (OriginLab). Statistical significance was calculated using two-sample t-test, with the significance assumed if $P < 0.05$. In all figure legends, n represents the number of independent biological replicates. All quantitative data were presented as mean ± s.e.m.

### Reporting summary

Further information on research design is available in the Nature Portfolio Reporting Summary linked to this article.

## Data availability

The cryo-EM density maps have been deposited to the Electron Microscopy Data Bank (EMDB) under the accession codes EMD-40741 (full-length GluA2–γ5–CNIH2_ZK-SPD), EMD-40742 (LBD-TMD for GluA2–γ5–CNIH2_ZK-SPD), EMD-40743 (LBD-TMD for GluA2–γ5–CNIH2_SPD), EMD-40744 (LBD-TMD for GluA2–γ5_apo), EMD-40745 (full-length GluA2–γ5–CNIH2_ZK-PMP-SPD), EMD-40746 (LBD-TMD for GluA2–γ5–CNIH2_ZK-PMP-SPD), EMD-40747 (full-length GluA2–γ5_ZK-PMP), EMD-40748 (LBD-TMD for GluA2–γ5_ZK-PMP), EMD-40749 (full-length GluA2–γ5–CNIH2_Glu-SPD) and EMD-40750 (LBD-TMD for GluA2–γ5–CNIH2_Glu-SPD). The atomic coordinates have been deposited to the PDB under the accession codes 8SS2 (full-length GluA2–γ5–CNIH2_ZK-SPD), 8SS3 (LBD-TMD for GluA2–γ5–CNIH2_ZK-SPD), 8SS4 (LBD-TMD

for GluA2–γ5–CNIH2$_{SPD}$), 8SS5 (LBD–TMD for GluA2–γ5$_{apo}$), 8SS6 (full-length GluA2–γ5–CNIH2$_{ZK-PMP-SPD}$), 8SS7 (LBD–TMD for GluA2–γ5–CNIH2$_{ZK-PMP-SPD}$), 8SS8 (full-length GluA2–γ5$_{ZK-PMP}$), 8SS9 (LBD–TMD for GluA2–γ5$_{ZK-PMP}$), 8SSA (full-length GluA2–γ5–CNIH2$_{Glu-SPD}$) and 8SSB (LBD–TMD for GluA2–γ5–CNIH2$_{Glu-SPD}$). The atomic coordinates under the accession codes 6DM1, 7OCE, 5WEO and 7R5Z were used for model building and structural comparisons. Source data are provided with this paper.

## Code availability

The in-house python scripts used to analyze MS data are available at https://github.com/NYUMetabolomics/plz.

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

## Acknowledgements

We thank R. Grassucci and Z. Zheng (Columbia University Cryo-Electron Microscopy Center) and P. Mitchell (Stanford Linear Accelerator Center/National Accelerator Laboratory) for help with microscope operation and data collection. Some of this work was performed at the Columbia University Cryo-Electron Microscopy Center. Some of this work was performed at the Stanford-SLAC Cryo-EM Center (S2C2), which is supported by the National Institutes of Health Common Fund Transformative High-Resolution Cryo-Electron Microscopy program (grant no. U24 GM129541). A.I.S. was supported by the National Institutes of Health (grant nos. NS083660, NS107253, AR078814 and CA206573).

## Author contributions

M.V.Y. made constructs for cryo-EM and electrophysiology and carried out protein expression. S.P.G. performed protein purification. S.P.G. and L.Y.Y. prepared cryo-EM samples and carried out cryo-EM data processing. S.P.G. and A.I.S. built molecular models. M.V.Y. performed patch-clamp recordings and electrophysiological data analysis. A.K. and D.R.J. performed MS experiments and analysis. S.P.G., M.V.Y., L.Y.Y. and A.I.S. wrote the manuscript.

## Competing interests

The authors declare no competing interests.

## Additional information

**Extended data** is available for this paper at https://doi.org/10.1038/s41594-023-01080-x.

**Correspondence and requests for materials** should be addressed to Alexander I. Sobolevsky.

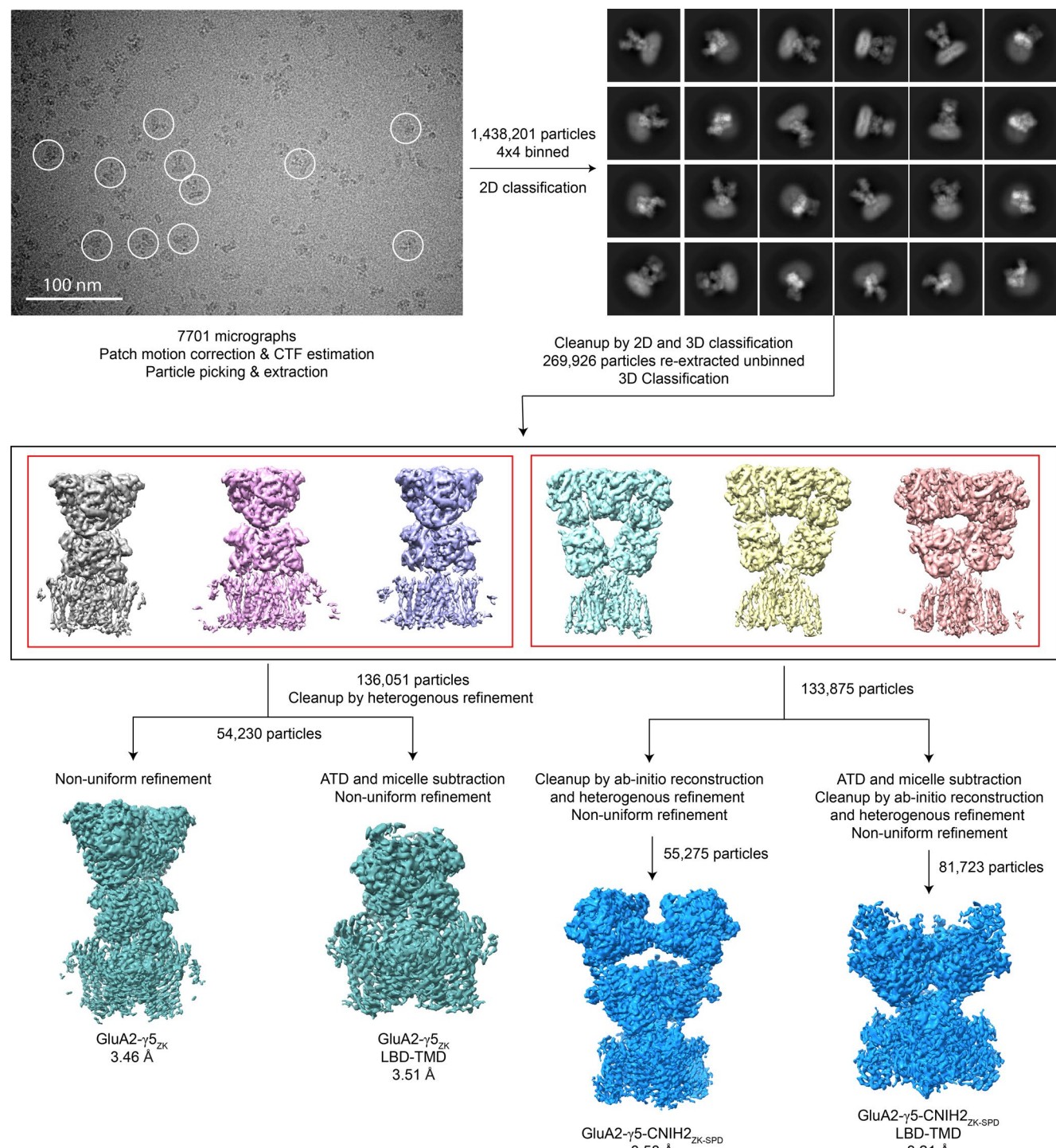

**Extended Data Fig. 1 | Cryo-EM data processing workflow for GluA2–γ5 and GluA2–γ5–CNIH2 in the presence of the competitive antagonist ZK200775.** A representative micrograph for the protein purified from HEK293S GnTI⁻ cells transduced with the GluA2–γ5 virus, supplemented with 0.1 mM ZK and subjected to single-particle cryo-EM shows example particles circled in white.

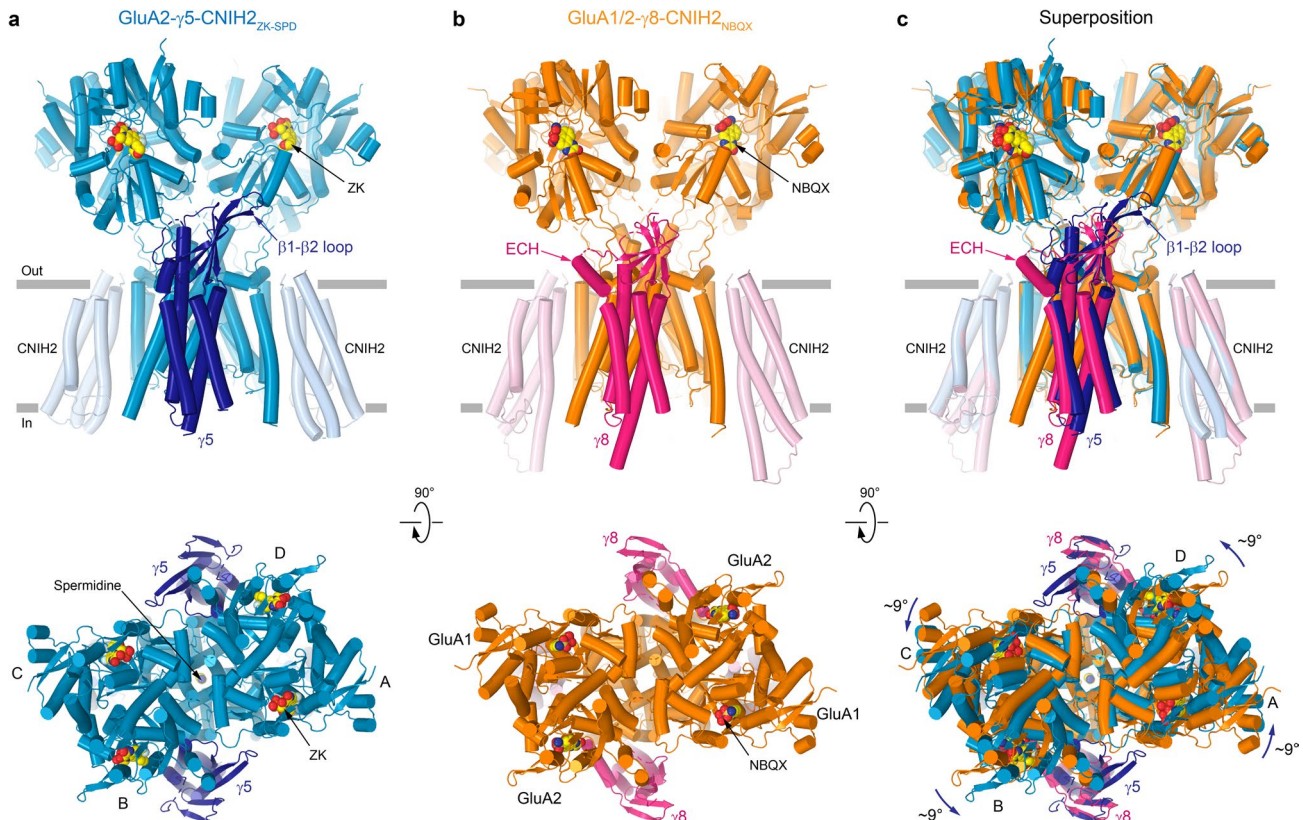

**Extended Data Fig. 2 | Comparison of AMPAR-CNIH2 complexes with γ5 and γ8.** a-c, Structures of GluA2–γ5–CNIH2$_{ZK-SPD}$ (a), GluA1/2–γ8–CNIH2$_{NBQX}$ (b; PDB ID: 7OCE), and their superposition (c) viewed parallel to membrane (top) or extracellularly (bottom), with molecules of ZK, SPD and NBQX shown as space-filling models and relative movement of the LBD layer illustrated by blue arrows. The ATD layer is not shown for clarity.

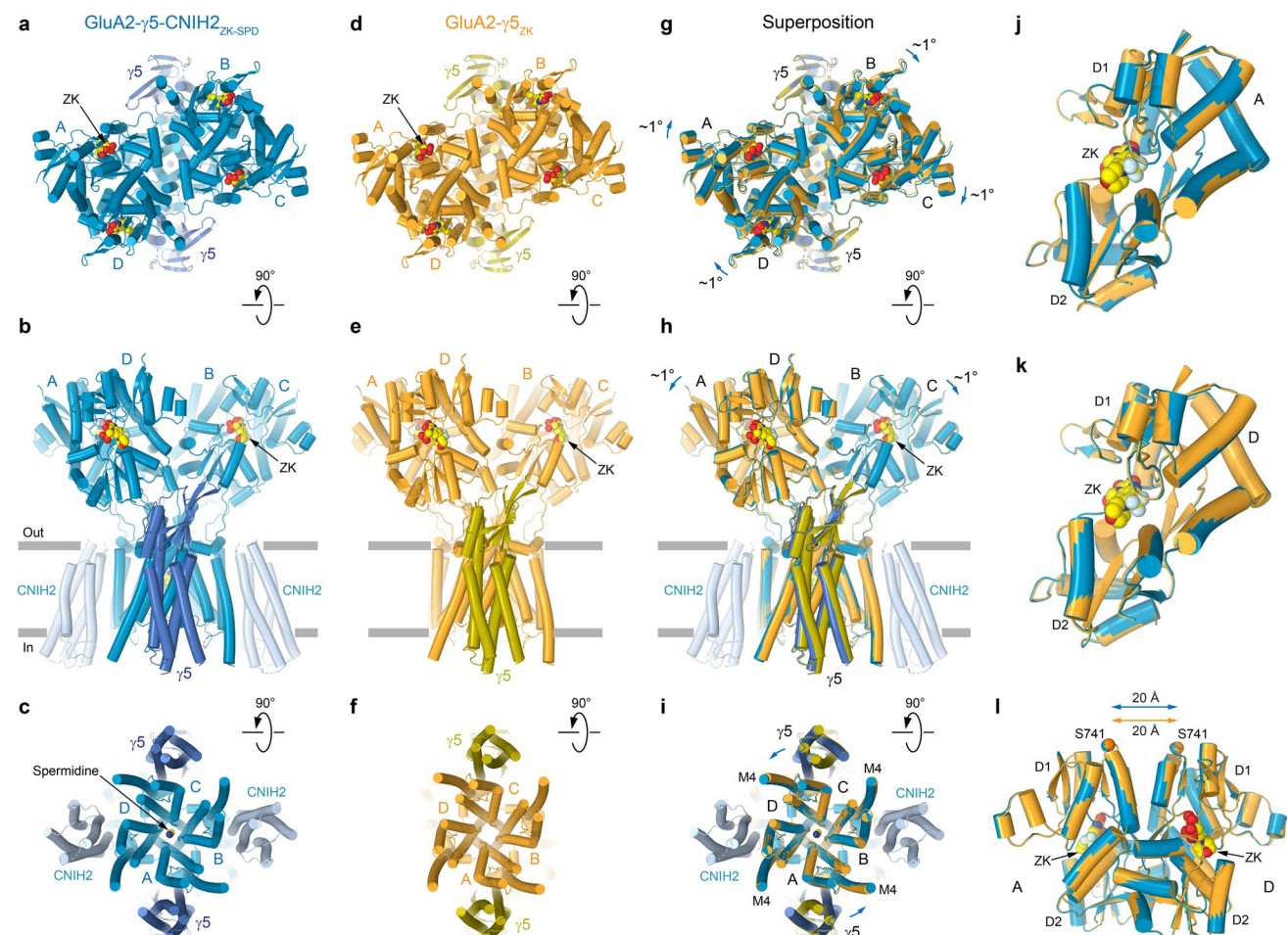

**Extended Data Fig. 3 | Comparison of closed-state structures GluA2-γ5-CNIH2$_{ZK-SPD}$ and GluA2-γ5$_{ZK}$.** a-i, Structures of GluA2-γ5-CNIH2$_{ZK-SPD}$ (a-c), GluA2-γ5$_{ZK}$ (d-f) and their superposition based on the GluA2 TMD (g-i) viewed extracellularly (a,d,g), parallel to the membrane (b,e,h), and intracellularly (c,f,i), with molecules of ZK and SPD shown as space-filling models. Blue arrows illustrate relative movement of domains upon CNIH2 binding. j-k, D2 lobe-based superposition of subunits A (j) and D (k) LBDs from GluA2-γ5-CNIH2$_{ZK-SPD}$ (blue) and GluA2-γ5$_{ZK}$ (yellow) structures. l, D2 lobe-based superposition of subunits A/D LBD dimers from GluA2-γ5-CNIH2$_{ZK}$ (blue) and GluA2-γ5$_{ZK}$ (yellow) structures. Double arrows indicate the distance between Cα atoms of S741.

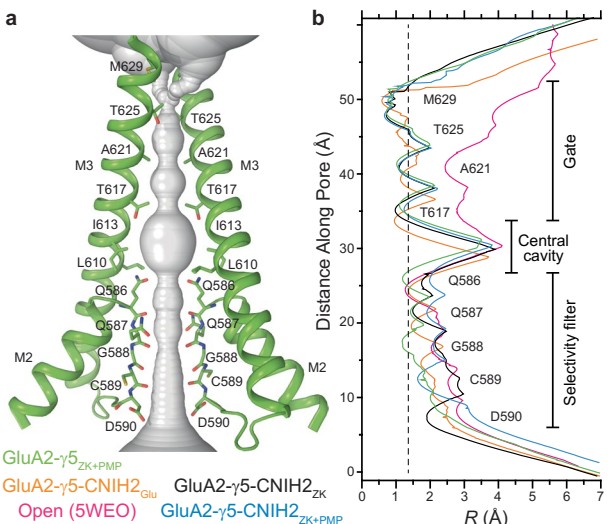

**Extended Data Fig. 4 | Closed conformation of the ion channel pore.** a, Pore-forming domains M2 and M3 in GluA2-γ5$_{ZK-PMP}$ with the residues lining pore shown as sticks. Only two (A and C) of four subunits are shown, with the front and back subunits (B and D) omitted for clarity. The pore profile is shown as a space-filling model (grey). b, Pore radius for the closed-state structures GluA2-γ5$_{ZK-PMP}$ (green), GluA2-γ5-CNIH2$_{ZK-SPD}$ (black), and GluA2-γ5-CNIH2$_{ZK-PMP-SPD}$ (blue), desensitized-state structure GluA2-γ5-CNIH2$_{Glu}$ (orange), and open-state structure GluA2-γ2$_{Glu+CTZ}$ (pink, PDB ID: 5WEO) calculated using HOLE. The vertical dashed line denotes the radius of a water molecule, 1.4 Å.

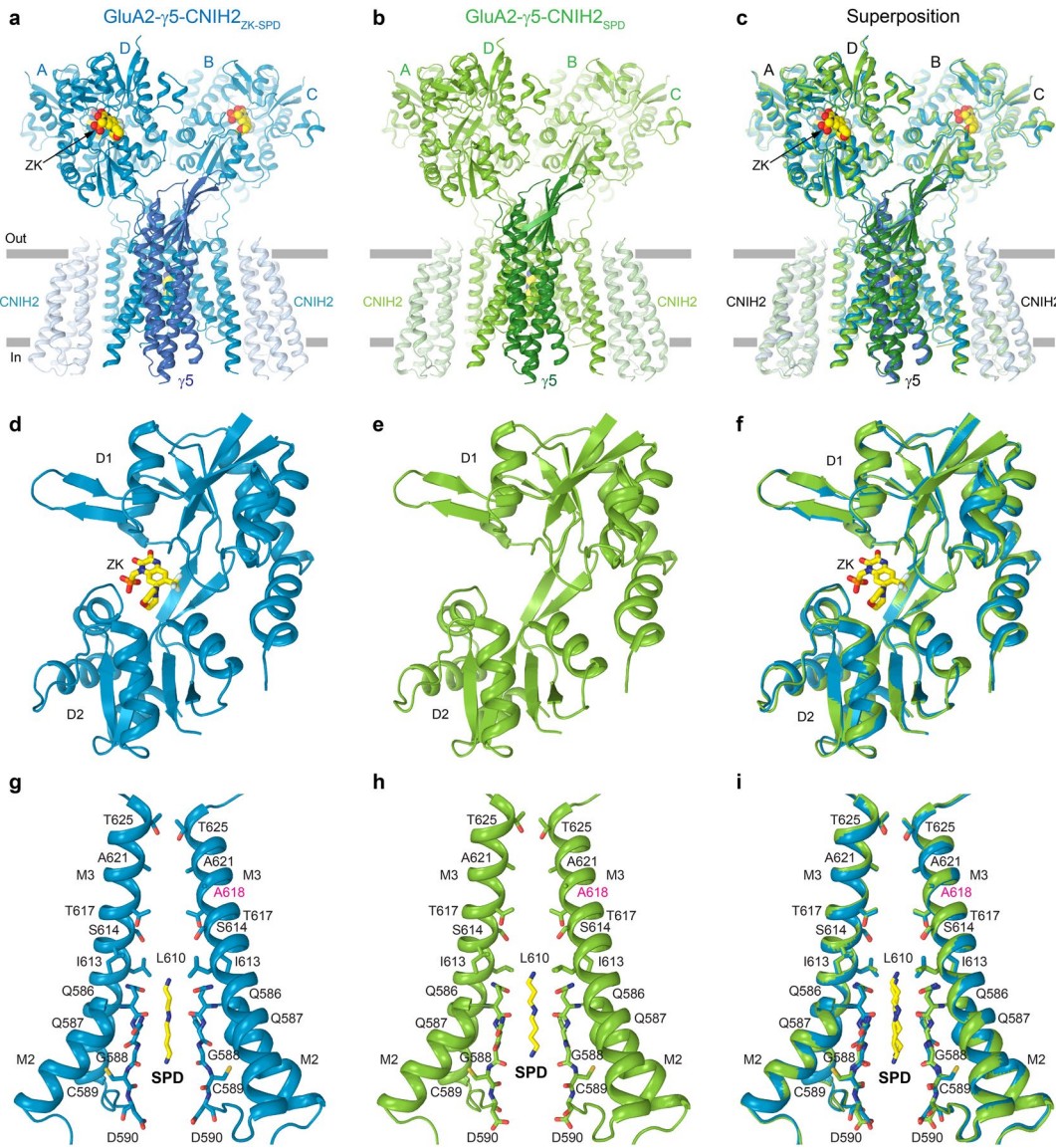

**Extended Data Fig. 5 | Comparison of ZK200775-bound and apo-state structures of GluA2-γ5-CNIH2.** a-c, LBD-TMD from GluA2-γ5-CNIH2$_{ZK-SPD}$ (a), GluA2-γ5-CNIH2$_{SPD}$ (b), and their superposition (c) viewed parallel to the membrane. d-f, LBD clamshell from GluA2-γ5-CNIH2$_{ZK-SPD}$ (d), GluA2-γ5-CNIH2$_{SPD}$ (e), and their superposition (f). g-i, Pore-forming domains in GluA2-γ5-CNIH2$_{ZK-SPD}$ (g), GluA2-γ5-CNIH2$_{SPD}$ (h), and their superposition (i). Only two of four subunits are shown, with the front and back subunits omitted for clarity. Molecules of ZK and SPD are shown as space-filling (a-c) or stick (d-i) models.

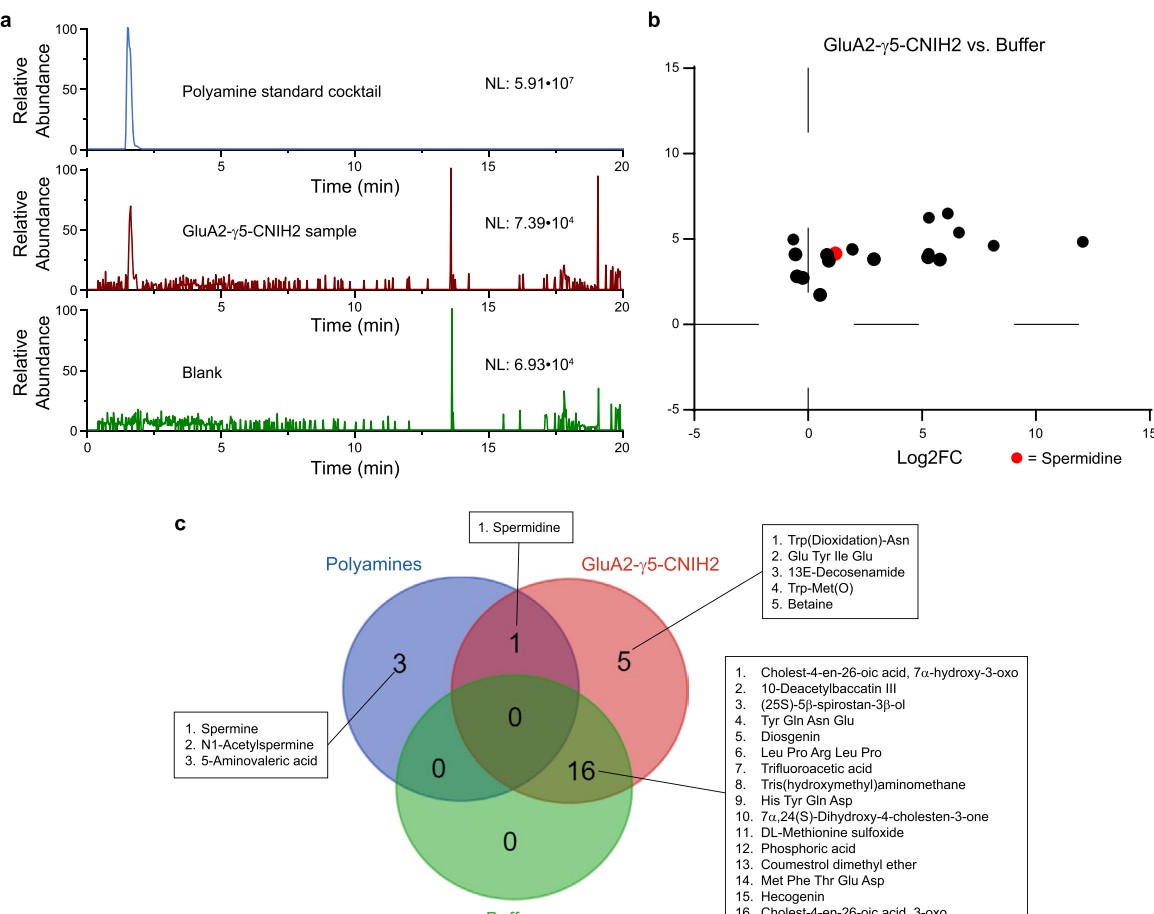

**Extended Data Fig. 6 | Mass spectrometry analysis.** a, Extracted ion chromatograms for SPD in samples m/z = 146.1653, positive mode. Top: authentic polyamine standard cocktail (containing 10 µM spermine, 10 µM SPD, 10 µM putrescine, and 10 µM N-acetyl-spermine). Middle: GluA2-γ5-CNIH2 sample, where SPD is above the detection limit (3X S/N minimum). Bottom: blank negative control processed side-by-side with samples and standards. b, Volcano plot depicting significantly upregulated and downregulated metabolites. SPD (red) is among the significantly upregulated metabolites. c, Venn diagram of overlapping significant metabolites between each of the three groups: polyamines, GluA2-γ5-CNIH2 sample, and buffer. SPD is the only metabolite found in both the GluA2-γ5-CNIH2 sample and the polyamine standard cocktail.

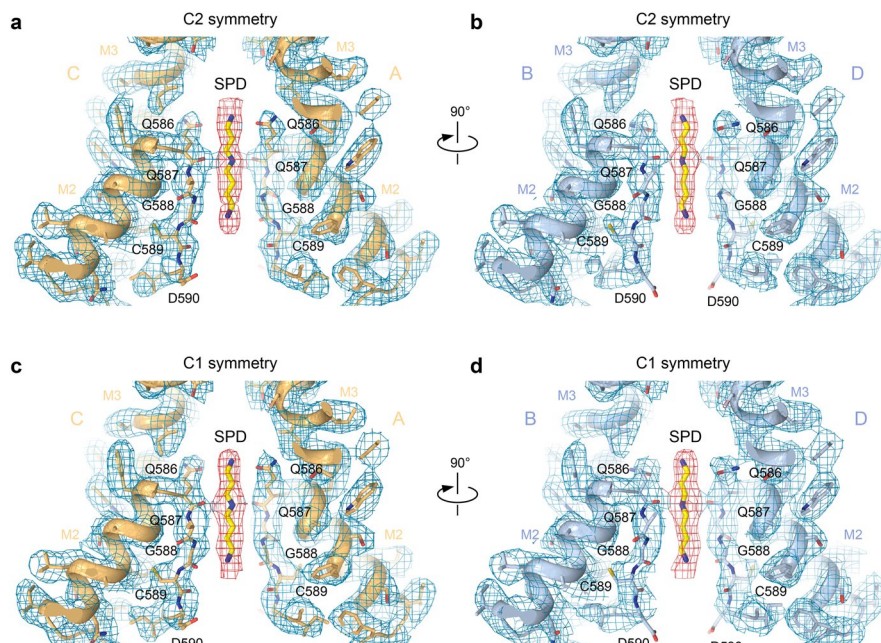

**Extended Data Fig. 7 | Comparison of spermidine density in cryo-EM reconstructions with C2 and C1 symmetry.** a-d, Pore region in GluA2-γ5-CNIH2$_{ZK-SPD}$, with density for cryo-EM reconstructions in the C2 (a-b) and C1 (c-d) symmetry shown as blue mesh for protein and red mesh for SPD, and pore-lining residues shown as stick models. Only subunits A and C are shown in panels a and c, with the front and back subunits (B and D) omitted for clarity. Vice versa, only subunits B and D are shown in panels b and d, with the front and back subunits (A and C) omitted for clarity.

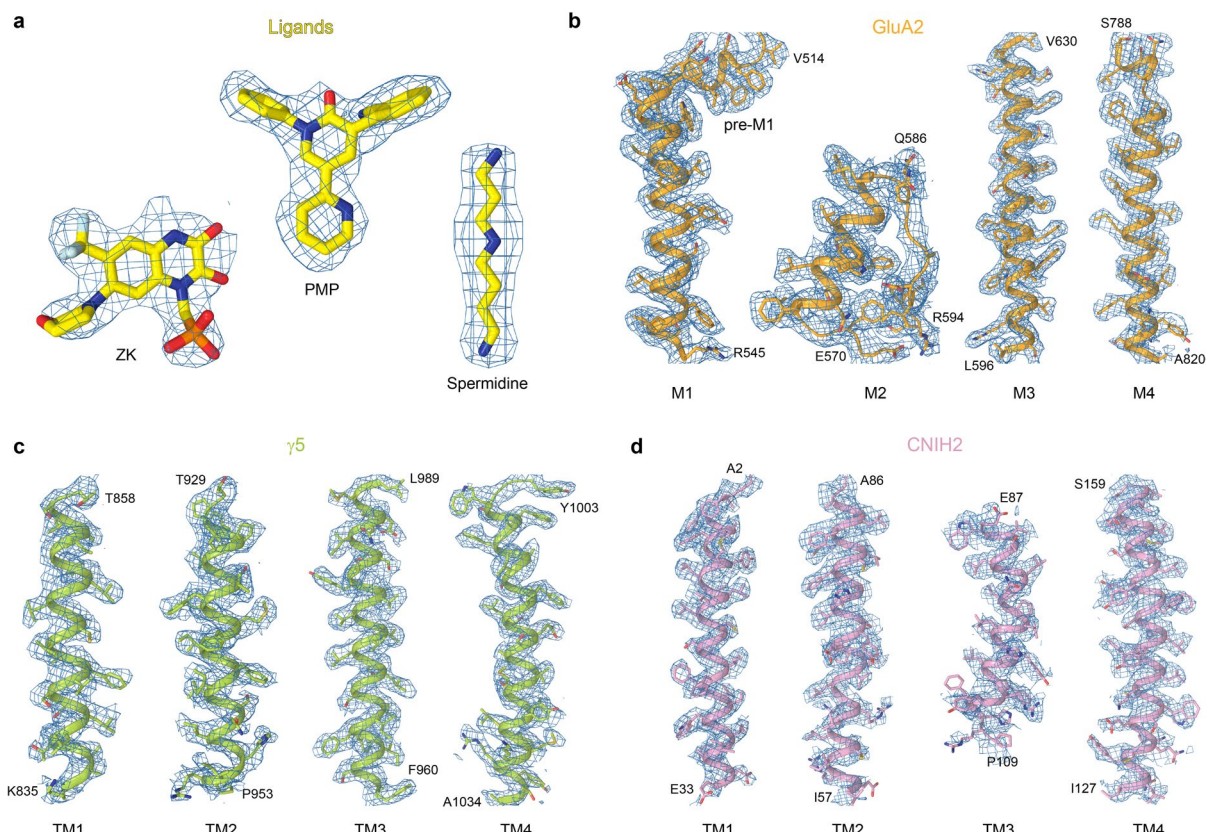

**Extended Data Fig. 8 | Cryo-EM density in GluA2-γ5-CNIH2$_{ZK-PMP-SPD}$.** a, ZK, PMP, and spermidine molecules are shown in sticks with the corresponding cryo-EM density as a blue mesh. b-d, Fragments of cryo-EM density (blue mesh) for the TMD segments of GluA2 (b), γ5 (c), and CNIH2 (d).

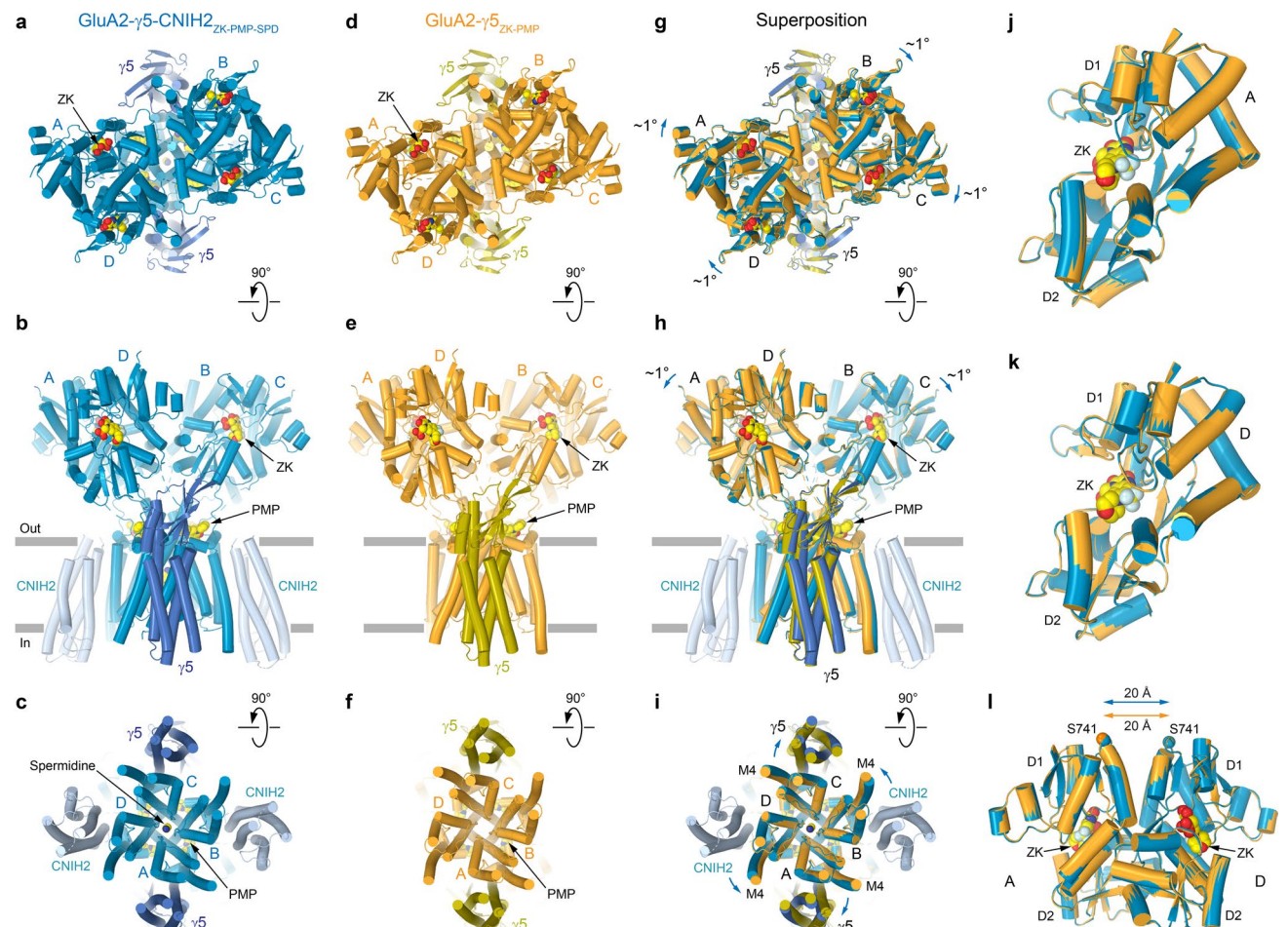

**Extended Data Fig. 9 | Comparison of PMP-bound closed-state structures in the presence and absence of CNIH2.** a-i, Structures of GluA2-γ5-CNIH2$_{ZK-PMP-SPD}$ (a-c), GluA2-γ5$_{ZK-PMP}$ (d-f) and their superposition based on the GluA2 TMD (g-i) viewed extracellularly (a,d,g), parallel to the membrane (b,e,h), and intracellularly (c,f,i), with molecules of ZK, SPD and PMP shown as space-filling models. Blue arrows illustrate relative movement of domains upon CNIH2 binding. j-k, D2 lobe-based superposition of subunits A (j) and D (k) LBDs from GluA2-γ5-CNIH2$_{ZK-PMP-SPD}$ (blue) and GluA2-γ5$_{ZK-PMP}$ (yellow) structures. l, D2 lobe-based superposition of subunits A/D LBD dimers from GluA2-γ5-CNIH2$_{ZK-PMP-SPD}$ (blue) and GluA2-γ5$_{ZK-PMP}$ (yellow) structures. Double arrows indicate the distance between Cα atoms of S741.

**Extended Data Table 1 | Functional characteristics**

| Expression constructs | GluA2 | GluA2-γ5 | GluA2 + CNIH2 | GluA2-γ5 + CNIH2 |
|---|---|---|---|---|
| $\tau_{\text{Deact}}$ (ms) | 1.76 ± 0.24 (n = 9) | 2.14 ± 0.15 (n = 8) | 10.8 ± 1.8 (n = 11) | 4.21 ± 0.42 (n = 9) |
| $\tau_{\text{Des}}$ (ms) | 7.70 ± 0.35 (n = 35) | 8.04 ± 0.85 (n = 10) | 28.0 ± 4.5 (n = 11) | 14.8 ± 1.6 (n = 14) |
| $I_{\text{SS}}/I_{\text{Max}}$ | 0.0455 ± 0.0090 (n = 26) | 0.0094 ± 0.0024 (n = 9) | 0.211 ± 0.031 (n = 11) | 0.0599 ± 0.0136 (n = 11) |
| $\tau_{\text{RecDes}}$ (ms) | 12.7 ± 1.5 (n = 14) | 35.0 ± 7.6 (n = 6) | 20.0 ± 2.0 (n = 9) | 36.7 ± 2.7 (n = 8) |
| $m$ | 5.74 ± 0.92 (n = 14) | 2.67 ± 0.34 (n = 6) | 3.00 ± 0.25 (n = 9) | 3.00 ± 0.32 (n = 8) |
| $I_{+40\text{mV}}/I_{-40\text{mV}}$ | 0.127 ± 0.015 (n = 7) | 0.468 ± 0.046 (n = 7) | 0.582 ± 0.072 (n = 13) | 0.579 ± 0.068 (n = 8) |
| $IC_{50}$ for PMP (μM) | 0.933 ± 0.143 (n = 5) | 2.18 ± 0.35 (n = 6) | 3.47 ± 0.76 (n = 7) | 3.82 ± 0.62 (n = 6) |
| $n_{\text{Hill}}$ | 1.11 ± 0.06 (n = 5) | 1.13 ± 0.11 (n = 6) | 1.24 ± 0.06 (n = 7) | 1.12 ± 0.10 (n = 6) |

The values of the deactivation time constant ($\tau_{\text{Deact}}$), the desensitization time constant ($\tau_{\text{Des}}$), the ratio of the steady-state current in the continuous presence of Glu and the maximal current amplitudes in the presence of CTZ ($I_{\text{SS}}/I_{\text{Max}}$), the recovery from desensitization time constant ($\tau_{\text{RecDes}}$), the index corresponding to the number of kinetically equivalent rate-determining transitions that contribute to the recovery from desensitization time course ($m$), the rectification index measured as the ratio of the current amplitudes at +40 mV and –40 mV in the presence of CTZ ($I_{+40\text{mV}}/I_{-40\text{mV}}$), the half-maximal inhibitory concentration of perampanel ($IC_{50}$), and the Hill coefficient ($n_{\text{Hill}}$) are presented as means ± SEMs.

# Reporting Summary

## Statistics

For all statistical analyses, confirm that the following items are present in the figure legend, table legend, main text, or Methods section.

| n/a | Confirmed | |
|---|---|---|
| ☐ | ☒ | The exact sample size (*n*) for each experimental group/condition, given as a discrete number and unit of measurement |
| ☐ | ☒ | A statement on whether measurements were taken from distinct samples or whether the same sample was measured repeatedly |
| ☐ | ☒ | The statistical test(s) used AND whether they are one- or two-sided *Only common tests should be described solely by name; describe more complex techniques in the Methods section.* |
| ☒ | ☐ | A description of all covariates tested |
| ☒ | ☐ | A description of any assumptions or corrections, such as tests of normality and adjustment for multiple comparisons |
| ☐ | ☒ | A full description of the statistical parameters including central tendency (e.g. means) or other basic estimates (e.g. regression coefficient) AND variation (e.g. standard deviation) or associated estimates of uncertainty (e.g. confidence intervals) |
| ☐ | ☒ | For null hypothesis testing, the test statistic (e.g. *F*, *t*, *r*) with confidence intervals, effect sizes, degrees of freedom and *P* value noted *Give P values as exact values whenever suitable.* |
| ☒ | ☐ | For Bayesian analysis, information on the choice of priors and Markov chain Monte Carlo settings |
| ☒ | ☐ | For hierarchical and complex designs, identification of the appropriate level for tests and full reporting of outcomes |
| ☒ | ☐ | Estimates of effect sizes (e.g. Cohen's *d*, Pearson's *r*), indicating how they were calculated |

*Our web collection on statistics for biologists contains articles on many of the points above.*

## Software and code

Policy information about availability of computer code

| Data collection | Leginon 3.5, SerialEM 4.0, EPU 2 |
|---|---|
| Data analysis | RELION 4.0, MotionCor2, CTFFIND4.1, cryoSPARC 4.2.0, cryoSPARC 4.2.1, UCSF Chimera 1.16, UCSF ChimeraX 1.3, COOT 0.9.8.1, PHENIX 1.18, pyMOL 2.5.2, HOLE 2.1, Origin 9.1.0, , Python SciPy 1.5.4, GraphPad Prism 9.4.1, Thermo Qual BrowserTM 4.0.27.19, pCLAMP 10.2, Clampfit10.3, DynDom 1.5 |

For manuscripts utilizing custom algorithms or software that are central to the research but not yet described in published literature, software must be made available to editors and reviewers. We strongly encourage code deposition in a community repository (e.g. GitHub). See the Nature Portfolio guidelines for submitting code & software for further information.

## Data

Policy information about availability of data

All manuscripts must include a data availability statement. This statement should provide the following information, where applicable:
- Accession codes, unique identifiers, or web links for publicly available datasets
- A description of any restrictions on data availability
- For clinical datasets or third party data, please ensure that the statement adheres to our policy

The cryo-EM density maps have been deposited to the Electron Microscopy Data Bank (EMDB) under the accession codes EMD-40741 [https://www.ebi.ac.uk/emdb/EMD-40741] (full-length GluA2-y5-CNIH2ZK-SPD), EMD-40742 [https://www.ebi.ac.uk/emdb/EMD-40742] (LBD-TMD of GluA2-y5-CNIH2ZK-SPD), EMD-40743

[https://www.ebi.ac.uk/emdb/EMD-40743] (LBD-TMD for GluA2-y5-CNIH2SPD), EMD-40744 [https://www.ebi.ac.uk/emdb/EMD-40744] (LBD-TMD for GluA2-y5apo), EMD-40745 [https://www.ebi.ac.uk/emdb/EMD-40745] (full-length GluA2-y5-CNIH2ZK-PMP-SPD), EMD-40746 [https://www.ebi.ac.uk/emdb/EMD-40746] (LBD-TMD for GluA2-y5-CNIH2ZK-PMP-SPD), EMD-40747 [https://www.ebi.ac.uk/emdb/EMD-40747] (full-length GluA2-y5ZK-PMP), EMD-40748 [https://www.ebi.ac.uk/emdb/EMD-40748] (LBD-TMD for GluA2-y5ZK-PMP), EMD-40749 [https://www.ebi.ac.uk/emdb/EMD-40749] (full-length GluA2-y5-CNIH2Glu-SPD), and EMD-40750 [https://www.ebi.ac.uk/emdb/EMD-40750] (LBD-TMD for GluA2-y5-CNIH2Glu-SPD). The atomic coordinates have been deposited to the Protein Data Bank (PDB) under the accession codes 8SS2 [https://doi.org/10.2210/pdb8SS2/pdb] (full-length GluA2-y5-CNIH2ZK-SPD), 8SS3 [https://doi.org/10.2210/pdb8SS3/pdb] (LBD-TMD of GluA2-y5-CNIH2ZK-SPD), 8SS4 [https://doi.org/10.2210/pdb8SS4/pdb] (LBD-TMD for GluA2-y5-CNIH2SPD), 8SS5 [https://doi.org/10.2210/pdb8SS5/pdb] (LBD-TMD for GluA2-y5apo), 8SS6 [https://doi.org/10.2210/pdb8SS6/pdb] (full-length GluA2-y5-CNIH2ZK-PMP-SPD), 8SS7 [https://doi.org/10.2210/pdb8SS7/pdb] (LBD-TMD for GluA2-y5-CNIH2ZK-PMP-SPD), 8SS8 [https://doi.org/10.2210/pdb8SS8/pdb] (full-length GluA2-y5ZK-PMP), 8SS9 [https://doi.org/10.2210/pdb8SS9/pdb] (LBD-TMD for GluA2-y5ZK-PMP), 8SSA [https://doi.org/10.2210/pdb8SSA/pdb] (full-length GluA2-y5-CNIH2Glu-SPD), and 8SSB [https://doi.org/10.2210/pdb8SSB/pdb] (LBD-TMD for GluA2-y5-CNIH2Glu-SPD). The atomic coordinates under the accession codes 6DM1 [https://doi.org/10.2210/pdb6DM1/pdb], 7OCE [https://doi.org/10.2210/pdb7OCE/pdb], 5WEO [https://doi.org/10.2210/pdb5WEO/pdb], and 7R5Z [https://doi.org/10.2210/pdb7R5Z/pdb] were used for model building and structural comparisons. Source data are provided with this paper.

## Human research participants

Policy information about studies involving human research participants and Sex and Gender in Research.

| Reporting on sex and gender | N/A |
| --- | --- |
| Population characteristics | N/A |
| Recruitment | N/A |
| Ethics oversight | N/A |

Note that full information on the approval of the study protocol must also be provided in the manuscript.

## Field-specific reporting

Please select the one below that is the best fit for your research. If you are not sure, read the appropriate sections before making your selection.

☒ Life sciences      ☐ Behavioural & social sciences      ☐ Ecological, evolutionary & environmental sciences

For a reference copy of the document with all sections, see nature.com/documents/nr-reporting-summary-flat.pdf

## Life sciences study design

All studies must disclose on these points even when the disclosure is negative.

| Sample size | Amount of cryo-EM data collected was limited by time allocation at the microscopes. For electrophysiological experiments, we performed all measurements five times or more. The reported sample size is based on published research by us and others, and is sufficient to obtains reproducible and reliable data in HEK cells using AMPA receptors. |
| --- | --- |
| Data exclusions | No data has been excluded. |
| Replication | No replication attempts have failed. Cryo-EM data collections were performed during continuous two-day data collection sessions and were consistent from the beginning to the end. A replication of the cryo-EM data collection was therefore not necessary or economically justifiable. In electrophysiological experiments, we made at least five independent replicates for each construct. |
| Randomization | Samples were not randomized; it is not technically or practically feasible to do so for cryo-EM or patch-clamp studies. Covariant control is not economically viable in cryo-EM data collections. Covariant control was alos not possible in electrophysiological experiments due to the need to transfect with predetermind cDNAs and optimize protein expression for individual constructs. |
| Blinding | Researchers were not blinded; it is not technically or practically feasible to do so for cryo-EM or electrophysiological experiments. It is not economically viable to blind cryo-EM collections. For electrophysiological experiments, researchers conducting the studies were also in charge of cell as well as protein expression optimization for individual constructs in order to achieve recordings or transfected cells in these studies. These circumstances made blinding not possible. |

## Reporting for specific materials, systems and methods

We require information from authors about some types of materials, experimental systems and methods used in many studies. Here, indicate whether each material, system or method listed is relevant to your study. If you are not sure if a list item applies to your research, read the appropriate section before selecting a response.

## Materials & experimental systems

| n/a | Involved in the study |
|---|---|
| ☒ | ☐ Antibodies |
| ☐ | ☒ Eukaryotic cell lines |
| ☒ | ☐ Palaeontology and archaeology |
| ☒ | ☐ Animals and other organisms |
| ☒ | ☐ Clinical data |
| ☒ | ☐ Dual use research of concern |

## Methods

| n/a | Involved in the study |
|---|---|
| ☒ | ☐ ChIP-seq |
| ☒ | ☐ Flow cytometry |
| ☒ | ☐ MRI-based neuroimaging |

## Eukaryotic cell lines

Policy information about cell lines and Sex and Gender in Research

| | |
|---|---|
| Cell line source(s) | HEK293S GnTI-, ATCC, Cat#CRL-3022<br>Sf9, Gibco, Cat#12659017<br>HEK 293, ATCC, Cat#CRL-1573 |
| Authentication | None of the cell lines used have been authenticated. |
| Mycoplasma contamination | The cell lines used have been tested for mycoplasma contamination by the providers (negative results) but have not been retested in the lab. |
| Commonly misidentified lines<br>(See ICLAC register) | No commonly misidentified lines were used in this study. |

