## [Peer Review File · Nature Structural & Molecular Biology]

Peer Review Information

Manuscript Title: Modulation of GluA2- γ 5 synaptic complex desensitization, polyamine block and antiepileptic perampanel inhibition by auxiliary subunit cornichon-2

Corresponding author name(s): Alexander Sobolevsky

Reviewer Comments & Decisions:

Decision Letter, initial version:

Message: 6th Apr 2023

Dear Sasha,

Thank you again for submitting your manuscript "Modulation of GluA2- γ 5 synaptic complex desensitization, polyamine block and antiepileptic perampanel inhibition by auxiliary subunit cornichon-2". I apologize for the delay in responding, which resulted from the difficulty in obtaining suitable referee reports. Nevertheless, we now have comments (below) from the 3 reviewers who evaluated your paper. In light of those reports, we remain interested in your study and would like to see your response to the comments of the referees, in the form of a revised manuscript.

You will see that while the reviewers appreciate the results, they have some concerns and suggestions which will need to be addressed in a revision.

Specifically, reviewers express reservations about the physiological significance of the GluA2/ γ 5/CNIH2 and bring up instances where functional and structural data are not in agreement. Reviewer #1 suggests exploring additive effects of the auxiliary subunits by adding functional data on GluA2+CNIH2 alone. In line with reviewer's #2 comments we would expect the statistical analysis to be revisited as requested. Moreover, we agree with reviewer's #3 suggestion that addition of MD simulations to further support the notions of dynamics as well as the mechanism of spermidine binding would strengthen the manuscript. Analysis of structural differences between GluA2 and A4 leading to a difference in response to γ 5 binding on polyamine block would be encouraged as well. Further to this, discussion on the mechanism of GluA2-TARP5-CNLH2 binding spermidine and comparison to previously reported results will be needed. Finally, reviewer #3 suggests that cryo-EM analysis of complex with PMP but without ZK would provide a more interpretable results.

Please be sure to address/respond to all concerns of the referees in full in a point-by-point

response and highlight all changes in the revised manuscript text file.

Please let me know if you would like to discuss the revision plan over the phone.

We appreciate the requested revisions are extensive. We thus expect to see your revised manuscript within 6 months. If you cannot send it within this time, please let us know. We will be happy to consider your revision as long as nothing similar has been accepted for publication at NSMB or published elsewhere. Should your manuscript be substantially delayed without notifying us in advance and your article is eventually published, the received date would be that of the revised, not the original, version.

Reporting Summary:

When submitting the revised version of your manuscript, please pay close attention to our [href="https://www.nature.com/nature-portfolio/editorial-policies/image-integrity">Digital Image Integrity Guidelines. and to the following points below:](https://www.nature.com/nature-portfolio/editorial-policies/image-integrity)

Please note that all key data shown in the main figures as cropped gels or blots should be presented in uncropped form, with molecular weight markers. These data can be aggregated into a single supplementary figure. While these data can be displayed in a relatively informal style, they must refer back to the relevant figures. These data should be submitted with the last revision, prior to acceptance, but you may want to start putting it together at this point.

SOURCE DATA: we urge authors to provide, in tabular form, the data underlying the graphical representations used in figures. This is to further increase transparency in data reporting, as detailed in this editorial

(<http://www.nature.com/nsmb/journal/v22/n10/full/nsmb.3110.html>). Spreadsheets can be submitted in excel format. Only one (1) file per figure is permitted; thus, for multi-paneled figures, the source data for each panel should be clearly labeled in the Excel file; alternately the data can be provided as multiple, clearly labeled sheets in an Excel file. When submitting files, the title field should indicate which figure the source data pertains to. We encourage our authors to provide source data at the revision stage, so that they are part of the peer-review process.

We require deposition of coordinates (and, in the case of crystal structures, structure factors) into the Protein Data Bank with the designation of immediate release upon publication (HPUB). Electron microscopy-derived density maps and coordinate data must be deposited in EMDB and released upon publication. Deposition and immediate release of NMR chemical shift assignments are highly encouraged. Deposition of deep sequencing and microarray data is mandatory, and the datasets must be released prior to or upon publication. To avoid delays in publication, dataset accession numbers must be supplied with the final accepted manuscript and appropriate release dates must be indicated at the galley proof stage. Please find the complete NRG policies on data availability at <http://www.nature.com/authors/policies/availability.html>.

[redacted]

Sincerely,
Kat

Katarzyna Ciazynska
(she/her)
Associate Editor
Nature Structural & Molecular Biology
<https://orcid.org/0000-0002-9899-2428>

Referee expertise:

Referee #1: AMPA receptors, pharmacology

Referee #2: neuropharmacology, receptors

Referee #3: cryo EM, AMPA receptors

Reviewers' Comments:

Reviewer #1:

Remarks to the Author:

The study of Gangwar et al. uses cryo-EM method to resolve structure of a complex consisting of AMPA receptor (AMPA) A2 subunit, fused to $\gamma 5$ auxiliary subunit and co-expressed with CNIH2 subunit.

The complex is solved in the presence of competitive antagonist ZK capturing the AMPAR in a closed, inactive state (with and without perampanel), and also in the presence of glutamate, capturing the AMPAR in a desensitized state.

The authors used whole-cell patch-clamp technique to investigate functional properties of A2_ $\gamma 5$ + CNIH2 complex and mass spectrometry to identify a density within the ion channel pore present in some of their cryo-EM data. Using functional and mass spectrometry data to support and enhance structural data is certainly a strength of the study.

The data are of very high quality and for most part presented clearly (see below a note on the mention of "dynamics" in line 238). This is the first structure of an AMPAR in complex with type II TARP ($\gamma 5$) and CNIH2 auxiliary subunits and the resolution of perampanel binding site is increased compared to the previous publication (Yelshanskaya, *Neuron*, 2016). The data, hence, contribute to our growing understanding of AMPAR complexes which mediate excitatory neurotransmission in the CNS and provide further insights into how these complexes might be regulated by clinically approved drug perampanel, which is of interest to a wider neuroscientific community.

My main comment would be that I am not sure what is the relationship between the functional and structural data presented in the manuscript. Both sets of data are of high quality, but as the authors nicely show in Extended Figure 3, the presence of CNIH2 in A2_ $\gamma 5$ + ZK complex has very little structural impact. However, functional data in Figure 1 indicate that CNIH2 is functionally dominant over $\gamma 5$. This is despite the fact that $\gamma 5$ makes contacts with A2 LBDs, unlike CNIH2. Is there anything in the structural data that could explain this? Perhaps CNIH2 interacts more extensively with BD subunits of the A2 tetramer, compared to $\gamma 5$ and BD subunits might play a bigger role in AMPAR gating?

Related to this, it might be useful to have A2+CNIH2 functional data for comparison. If it

is not possible to perform the recordings, than at least to share already published values from, e.g. Coombs et al., Journal of Neuroscience, 2012. Having both, A2_y5 and A2+CNIH2 functional data will make it easier to see whether the effects of the 2 types of auxiliary subunits are additive or intermediate and might help interpret structure-function relationship.

There is also an unexpected result with spermidine in terms of structure-function relationship, but in this case, the authors did a good job by making this clear and discussing possible explanations.

The observation of C589 disulphide bond formation in the presence of CNIH2 and PMP is interesting and not observed before. Given that C589 is located intracellularly, how likely is this bond to form in vivo? If I am not mistaken, C589 was mutated into Ala in some previous A2 constructs for X-ray crystallography (e.g. 3kg2) and the mutation is functionally silent. If so, this should be mentioned. Nevertheless, the observation does suggest conformational changes in this region linked to CNIH2 and PMP. In line 238, the authors suggest dynamics of this region (and explain it further), but it is unclear what is meant by this dynamics. Is this region dynamic in general, allowing crosslinking in the presence of CNIH2 and PMP or is the dynamics somehow imparted by CNIH2 and PMP? If it is imparted by CNIH2 and PMP, it would be helpful to see, in a panel for example, what is meant by this dynamics, what specific movements of the protein are caused by CNIH2 and/or PMP binding.

Summary and Outlook section could be expanded into more thorough Discussion including some discussion points suggested here.

Minor suggestions:

- 1) It is true AMPARs form many different complexes with auxiliary subunits in vivo, but majority, at least in hippocampus, seem to contain $\gamma 8$ and CNIH2 (Yu, Nature, 2021). Perhaps there is some evidence of $\gamma 5$ and CNIH2 co-expression? In any case, it would be good to include a sentence about $\gamma 5$ expression, if known, just to put the complexes in a bit more physiological context.
- 2) Figure 1 – please include statistics and show individual data points.
- 3) p. 5, line 100 – more slowly than what?
- 4) p. 5, line 108 – add to which condition/construct the numbers refer to
- 5) In cryo-EM samples of A2_y5+CNIH2 with ZK and PMP, about 50% of the final particles do not have CNIH2 – it might be worth commenting on this in the Discussion. Could it be that PMP causes CNIH2 dissociation? Structures do not seem to suggest this as the presence of PMP does not cause large enough conformational changes, but perhaps I am missing something. The recordings in Fig. 4a also don't seem to suggest that (no reduction in Iss at 30 μ M PMP), but perhaps it's a very slow process.
- 6) Structure of the desensitized complex A2_y5+CNIH2+Glu indicates structural mechanism by which CNIH2 weakens AMPAR desensitization (reduced separation of D1 lobes in LBD dimers). This explains slower desensitization and higher Iss, but might be at odds with slower recovery from desensitization in the presence of CNIH2 (Fig. 1F). Perhaps this is not the only desensitized conformation of A2_y5+CNIH2?

Reviewer #2:

Remarks to the Author:

This study describes cryo-EM structures of homotetrameric GluA2 AMPA receptors associated with TARP gamma5 and CNIH2 or with gamma5 alone. The structures provide mechanistic insights to explain the functional effects of receptor association with both a potentiating auxiliary subunit CNIH2 and an inhibiting subunit gamma5. Observations from the new structures suggest that CNIH2 destabilize the desensitized state of the receptor by reducing separation of the upper lobes in the LBD dimers. Furthermore, it is observed that CNIH2 stabilizes spermidine binding in the closed ion channel, and it is suggested that CNIH2 facilitates release of spermidine from the open channel. Effects of receptor association with gamma5 and CNIH2 on inhibition by the antiepileptic drug perampanel are also investigated and it is argued that these auxiliary subunits cause small reductions in perampanel potency. While the mechanism behind this observation was not resolved in the study, it is likely that CNIH2 simply reduces perampanel potency by influencing gating equilibrium. The observations from the new structures are nicely supported by high-quality functional data, data presentation and interpretation are reasonable, and there are no questions related to the rigor and robustness of the results and conclusions.

This is an important study that provides new structures of AMPA receptors in complex with auxiliary subunits. The GluA2-gamma5-CNIH2 structures in this study will add context and additional information to appreciate the structural mechanisms that govern the function of physiologically relevant AMPA receptors in synapses. The quality of the structures is a strength of the study, while it is considered a weakness that more physiologically relevant heterotetrameric AMPA receptors, such as for example GluA1/2, were not investigated. While the structures will certainly advance our understanding of AMPA receptor structure-function, the manuscript is a bit descriptive and the investigations of spermidine block and perampanel binding do not appear to add much in terms of raising the impact of the study. Overall, this is a nicely executed study, but there are, however, some specific points that could be addressed to improve the manuscript and strengthen the study.

- Would CNIH2 be expected to modify rectification in native GluA2-lacking, Ca²⁺-permeable AMPA receptors. Please elaborate on this. The rectification data shown in panels d-g of Extended Figure 5 are not quantitative. Are the results observed on multiple cells or just one cell? Extended Figure 5d-g might also be helpful to include in the main manuscript.
- The differences in perampanel potencies are rather small and statistical tests are not performed to suggest they are significant.
- The authors should elaborate on the brain regions and/or neuronal cell types in which AMPA receptors, gamma5, and CNIH2 might be co-expressed and if any direct evidence exist to suggest that gamma5 and CNIH2 can be found together in native AMPA receptor complexes.
- Introduction, line 77: It would be more accurate to describe CNIH2 as a potentiating auxiliary subunit. "Activating" subunit might suggest that this protein can activate the receptor?
- The "Summary and Outlook" section is mostly summary without outlook. Please revise to include more perspective on how the structures and findings of the study advance our understanding of AMPA receptor structure-function.

- In the methods section, it would be helpful with a description of how domain rotations and LBD closure were determined when comparing the structures.
- Lines 559-560: How will adding 7 mM NaCl to the extracellular solution increase the rate of solution exchange?
- The fitted m values from the Hodgkin-Huxley equation should be provided.

Reviewer #3:

Remarks to the Author:

A. Summary of the key results

Gangwar, Yen et al report the complex between a model AMPAR (unedited GluA2), a type-2 TARP (TARP5) and CNLH2. They analyze the structure in the presence of competitive (ZK) +- non-competitive (PMP) antagonists and also in the presence of L-Glutamate, revealing 1) overall structural differences induced by CNLH2 binding 2) specific spermidine binding to CNLH2 complexes and lack of polyamine inhibition at positive potentials 3) the overall features of the desensitized states and 3) the binding of antagonist PMP.

B. Originality and significance: if not novel, please include references

Other AMPAR complexes including TARPs and CNLH2 have been previously reported (Zhang et al Nature 2021, Zhao et al 2019, Yu et al 2021), but here the focus is on the influence of CNLH2 on type-2-TARP containing AMPARs. A justification of the physiological relevance of such ternary complex is needed: in the cerebellum, in cells expressing AMPAR, CNLH2 and type-2 TARPs (specifically, TARP7), CNLH2 exclusively resides within cells, and not in the surface. Proteomic analysis (Schwenk et al, 2012) reveals a negative correlation between type-2 TARPs (specially TARP5) and GluA2 or CNLH2 subunits, suggesting that both auxiliary proteins would not coexist in the same complex. Therefore, any evidence supporting the physiological relevance of the ternary complex is needed.

On the other hand this is the first time that a CNLH2-containing AMPAR complex is determined in a desensitized state. Structures also reveal binding of spermidine in a CNLH2-dependent manner; this unique observation is striking, as their functional analysis suggest that such complex is not inhibited by spermidine.

C. Data & methodology: validity of approach, quality of data, quality of presentation

Authors combine cryo-EM data with electrophysiology characterization. Cryo-EM maps are of excellent quality. It is not clear why they choose to analyze PMP binding in combination with another antagonist (see below in specific comments)

D. Appropriate use of statistics and treatment of uncertainties

Electrophysiology data includes N values and appropriate statistical tests. Showing each data point in charts as well as significance when comparing conditions may be needed. For I/V graphs, adding the analysis of rectification is needed to really be able to compare the conditions.

E. Conclusions: robustness, validity, reliability F. Suggested improvements: experiments, data for possible revision G. References: appropriate credit to previous work?

See below for specific comments.

H. Clarity and context:

Abstract is clear, the introduction needs some clarification about the physiological relevance of the complex, a summary figure would be helpful (see below).

Specific comments are listed below:

1. The authors report that TARP5 strengthens polyamine block in unedited GluA2, contrary to what happens in GluA4-TARP5, and surprisingly, CNLH2 completely removes polyamine block and displays I/V curves similar to calcium-impermeable AMPARs. Extended figure 5 does not show clear differences between GluA2 and GluA2-G5: a comparison of the rectification index is needed as well as a discussion about GluA2/A4 differences that may explain the opposite effects observed with both AMPAR subunits.
2. Authors are not very convincing justifying why the only complex which is not blocked by polyamines (GluA2-TARP5-CNLH2) is actually binding spermidine in all the analyzed states (desensitized, ZK and ZK+PNP bound structures). In line 169 and line 180, authors use a very similar sentence to justify two different things: "Given the similarity of the selectivity filter in the in the closed state GluA2- γ 5-CNIH2 and open-state GluA2- γ 2... binding of polyamines to this site is the likely cause of inward rectification of Ca²⁺-permeable AMPAR currents" and "Given the similarity of the selectivity filter structure in the closed state GluA2- γ 5 and open-state GluA2- γ 2 (Extended Data Fig. 4), our results suggest that the selectivity filter in the open state of CNIH2-bound AMPAR adapts a polyamine-unbound conformation, different from the open state in the absence of CNIH2". First, most AMPAR structures are very similar in the pore region, if the authors mean that the region is similarly accessible in open and closed structures, why is this binding specifically observed in the CNIH2 complex, which is not blocked by polyamines at positive potentials? Authors previously reported how polyamines were blocking Ca-Permeable AMPARs (Twomey et al, 2018): a comparison with these structures rather than a comparison of the pore dimensions with 5WEO would be more adequate to understand spermidine inhibition. How does the binding of NASPM and spermidine compare? Why would you expect that the binding site currently observed is the same that is blocking currents at positive potentials in the absence of CNLH2? In Twomey et al, 2018, they reported that activation was a requirement for channel block by polyamines: if the observed spermidine binding site is the blocking site at positive potentials, and here it is detected in closed receptors, how do you reconcile all the data?
3. In sentence 180, again they analyze the similarities in the pore dimensions between closed GluA2-TARP5 and open GluA2-TARP2 to conclude that the selectivity filter in the open state of CNIH2-bound AMPAR adapts a polyamine-unbound conformation. This observation must be true as there is a linear I/V response upon CNLH2 binding, but the structural data are not supporting it. Binding of spermidine to CNLH2 containing AMPARs and lack of inhibition at positive potentials is the most novel aspect in the manuscript, but needs additional support to understand how it is achieved. A comparison of the electrostatics that could point at differences at the pore between complexes, a structure of the open-state ternary complex, or molecular dynamics simulations may provide some information about why the structures actually show the opposite to what is expected. Besides, other cryo-EM maps (EMD12805, EMD12806) with CNIH2 seem to have elongated densities and display linear I/V relationship (although they are calcium-impermeable). These densities are not that clear in cryo-EM maps lacking CNLH2, and maybe this is pointing towards a more general CNLH2-dependent mechanism.
4. The purified protein shows binding of spermidine in MS data, despite only a subset of the pure protein seems to be forming a complex with cornichon (according to the classification workflow in ext.figure 6). Have you analyzed spermidine binding to the complex lacking CNLH2 by MS? Do the densities for spermidine are still clear when refining data in C1 symmetry?

5. Previous reports in GluA1/A2/TARP8 revealed that the interaction between Ile569 and Val220 in TARP8 is relevant for polyamine inhibition (Herguedas et al, 2022), and mutations in Ile569 reduce polyamine inhibition. Do you see any differences in the region?

6. The authors show that both TARP5 alone and TARP5+ CNLH2 weaken PMP inhibition and proceed to solve the structure of PMP complexes. Structures are determined in the presence of another antagonist and it seems that any modulatory difference by PMP is masked by ZK-binding. For example, extended figure 3 and extended figure 8 are almost identical, as well as the differences highlighted in lines 225-230, which seem mainly due to CNLH2 binding rather than PMP binding. Apart from showing a detailed analysis of the PMP binding site and a novel disulfide in the pore, the structures do not provide information about how PMP is exerting a different modulation on GluA2, GluA2-TARP5 or GluA2-TARP5-CNLH2 complexes. The disulfide bridge is very interesting, but it is not clear why the authors point at an increased protein dynamics in the region to justify the decrease in PMP inhibition upon CNLH2 binding? Do you have any evidence of dynamics from the cryo-EM data? Have you analyzed the excluded particles during cryo-EM analysis that could support this idea? The authors say that "PMP can influence the selectivity filter dynamics allosterically, through the M1, M3 and M4 helices" but they should specify which residues in M1, M3 or M4 helices are allosterically coupling the PMP binding site and the pore, as well as how CNLH2 may influence such PMP inhibition. In general, this section is a bit weak, as obtaining the structure of the complex with two antagonists is not ideal; the same authors recently revealed PMP binding to Kainate Receptors (Cell Reports, 2023), and in that case they used a receptor bound to an allosteric modulator. In other AMPAR structures (GluA1/2/TARP8 open state), striking differences are observed in the binding sites of PMP, with only two of the sites accessible. Such differences are only clear in the open state structure, and a similar mechanism may be happening here. Therefore, an apo structure with PMP or CTZ-L-Glu bound structure may illuminate the PMP inhibition mechanism and the differences between complexes. Alternatively, MD simulations may support the proposed increased dynamics.

7. Authors report the presence of a lipid bound in the vicinity of the pore; is it only present in CNLH2-bound structures? Does the head of the lipid interact at the proximity of the extracellular region (and maybe closer to the PMP binding region) or the cytosolic region? Do you see any state-dependent or complex-dependent change in the conformation of the lipid? If the lipid may have any functional implication a figure should be presented.

8. For desensitization data authors report a slower recovery of desensitization in GluA2-TARP5-CNLH2 complexes in comparison with GluA2-TARP5 and GluA2 alone, but you suggest that there is a weaker desensitization: why a reduced separation in lower lobes would lead to a much slower recovery from the desensitized state? Also, while the observations are clear, how is cornichon specifically causing this? Apart from the "slightly more compact TMD" the authors do not analyze the details of how cornichon may modulate desensitization.

Minor comments:

1. Please, describe precisely how the heteromeric complex is obtained. The CNLH2 construct seems to contain GFP and strep tag, as well as the GluA2-TARP5 complex. If the same tag has been used for both constructs, how do you ensure that the "ternary" complex is obtained?

2. Please, show 2Ds and processing workflow of the three datasets. In Ext. Figure 6 it seems that some classes lack auxiliary proteins (gray class), is this the case? Do you observe different stoichiometries in CNLH2 binding apart from 2-TARP5-2-CNLH2? In the ZK and L-Glu datasets, do you also observe particles lacking cornichon or is this

heterogeneity only present in the PMP bound dataset?

3. Extended data Fig 6: correct "full-length".

4. Figure 5, label PMP a bit bigger.

5. Figure 7 is not really helping much to see the differences in desensitized states of CNLH2-lacking and CNLH2-containing complexes: maybe a zoom into the main regions that are different between complexes will be more adequate.

6. Extended Data Figure 4: add a green label for GluA2-g5-ZK-PMP.

Final comment:

The manuscript shows some really interesting observations (both ephys and structural data), however functional and structural observations are not fitting well, probably some MD would help a lot to reconcile all the data, as well as analyzing it in the context of other AMPAR structural biology work (including their own publications such as the Twomey Neuron paper in 2018).

Author Rebuttal to Initial comments

We thank Reviewers for their excellent suggestions that have led to a significant improvement of this manuscript. We have made changes in the manuscript with details outlined in our responses below.

Reviewer #1:

Remarks to the Author:

The study of Gangwar et al. uses cryo-EM method to resolve structure of a complex consisting of AMPA receptor (AMPA) A2 subunit, fused to y5 auxiliary subunit and co-expressed with CNIH2 subunit.

The complex is solved in the presence of competitive antagonist ZK capturing the AMPAR in a closed, inactive state (with and without perampanel), and also in the presence of glutamate, capturing the AMPAR in a desensitized state.

The authors used whole-cell patch-clamp technique to investigate functional properties of A2_y5 + CNIH2 complex and mass spectrometry to identify a density within the ion channel pore present in some of their cryo-EM data. Using functional and mass spectrometry data to support and enhance structural data is certainly a strength of the study.

The data are of very high quality and for most part presented clearly (see below a note on the mention of "dynamics" in line 238). This is the first structure of an AMPAR in complex with type II TARP (y5) and CNIH2 auxiliary subunits and the resolution of perampanel binding site is increased compared to the previous publication (Yelshanskaya, Neuron, 2016). The data, hence, contribute to our growing understanding of AMPAR complexes which mediate excitatory neurotransmission in the CNS and provide further insights into how these complexes might be regulated by clinically approved drug perampanel, which is of interest to a wider neuroscientific community.

We thank Reviewer #1 for the generous assessment of our work.

My main comment would be that I am not sure what is the relationship between the functional and structural data presented in the manuscript. Both sets of data are of high quality, but as the authors nicely show in Extended Figure 3, the presence of CNIH2 in A2_y5 + ZK complex has very little structural impact. However, functional data in Figure 1 indicate that CNIH2 is functionally dominant over y5. This is despite the fact that y5 makes contacts with A2 LBDs, unlike CNIH2. Is there anything in the structural data that could explain this? Perhaps CNIH2 interacts more extensively with BD subunits of the A2 tetramer, compared to y5 and BD subunits might play a bigger role in AMPAR gating?

We have now added functional data for GluA2-CNIH2 complex, which allows to better separate individual effects of y5 and CNIH2 on the receptor and compare them to the influence of both (Figures 2, 3 and 5). We have demonstrated movement of the transmembrane helices, changes in the pore profile and rotation of individual LBD upon CNIH2 incorporation into the complex (Extended Data Figs. 4, 5 and 13). While these are relatively small changes that characterize closed states, they might be part of changes in gating equilibrium caused by CNIH2. More importantly, we demonstrate binding of SPD to the closed-channel selectivity filter that occurs only in the presence of CNIH2 (Figures 3), changes in symmetrical organization of the intracellular pore entrance upon additional binding of PMP (Figures 6) and reduced separation of the upper lobes in ligand-binding domain dimers that accompanies destabilization of the desensitized state by CNIH2 (Figure 8). We have also added a Discussion that describes all these structural changes in the context of the corresponding functional observations (lines 337-414).

Related to this, it might be useful to have A2+CNIH2 functional data for comparison. If it is not possible to perform the recordings, than at least to share already published values from, e.g. Coombs et al., Journal of Neuroscience, 2012. Having both, A2_y5 and A2+CNIH2 functional data will make it easier to see whether the effects of the 2 types of auxiliary subunits are additive or intermediate and might help interpret structure-function relationship.

We are very thankful to Reviewer #1 for this suggestion. We have collected data for GluA2-CNIH2, which now allows us to make a much more direct connection between our structural and functional observations. The corresponding changes have been made in Figures 2, 3 and 5 and throughout the text of the manuscript.

There is also an unexpected result with spermidine in terms of structure-function relationship, but in this case, the authors did a good job by making this clear and discussing possible explanations.

The observation of C589 disulphide bond formation in the presence of CNIH2 and PMP is interesting and not observed before. Given that C589 is located intracellularly, how likely is this bond to form in vivo? If I am not mistaken, C589 was mutated into Ala in some previous A2 constructs for X-ray crystallography (e.g. 3kg2) and the mutation is functionally silent. If so, this should be mentioned.

We agree with Reviewer #1 that this disulfide bond is unlikely to be present in vivo due to the reducing condition inside the cell. The possibility of this disulfide bond formation in our structural conditions, however, is an indication of flexible character of the selectivity filter region. We have now mentioned that the C589A mutation is functionally silent and references the original paper reporting the 3KG2 structure (lines 297-303).

Nevertheless, the observation does suggest conformational changes in this region linked to CNIH2 and PMP. In line 238, the authors suggest dynamics of this region (and explain it further), but it is unclear what is meant by this dynamics. Is this region dynamic in general, allowing crosslinking in the presence of CNIH2 and PMP or is the dynamics somehow imparted by CNIH2 and PMP? If it is imparted by CNIH2 and PMP, it would be helpful to see, in a panel for example, what is meant by this dynamics, what specific movements of the protein are caused by CNIH2 and/or PMP binding.

We meant increased flexibility of the selectivity filter region that allows disulfide bond formation between cysteines C589. We have now changed the text to make our point clear (lines 297-303). To illustrate conformational changes caused by addition of CNIH2, we have made Extended Data Figure 13, which compares GluA2-y5-CNIH2_{ZK-PMP-SPD} and GluA2-y5_{ZK-PMP} structures. In the absence of PMP, the conformational changes introduced by addition of CNIH2 are illustrated in Extended Data Figure 4. The effect of PMP addition is illustrated in Figure 6 by side-by-side comparison of GluA2-y5-CNIH2_{ZK-PMP-SPD} and GluA2-y5-CNIH2_{ZK-SPD} structures.

Summary and Outlook section could be expanded into more thorough Discussion including some discussion points suggested here.

Minor suggestions:

1) It is true AMPARs form many different complexes with auxiliary subunits in vivo, but majority, at least in hippocampus, seem to contain y8 and CNIH2 (Yu, Nature, 2021). Perhaps there is some evidence of y5 and CNIH2 co-expression? In any case, it would be good to include a sentence about y5 expression, if known, just to put the complexes in a bit more physiological context.

Although y5 and CNIH2 are likely not the predominant constituents of synaptic complexes, they show similar molecular abundances in the cerebellum (Schwenk, 2014). In particular, they are

present in Bergmann glia, that typically express Ca-permeable AMPARs and receive direct synaptic input from glutamatergic neurons (Soto et al., 2009). They are also present in the hippocampal NG2 glial cells (Cahoy et al., 2008; Hardt et al., 2021). The corresponding information has been added to the text (lines 76-80).

2) Figure 1 – please include statistics and show individual data points.

As suggested, we showed individual points and performed statistical comparisons (see the updated Figure 2).

3) p. 5, line 100 – more slowly than what?

Prolonged Glu application elicited an inward current that decayed in the continuous presence of Glu more slowly than in response to the short 2-ms Glu application. The corresponding change has been introduced into the text (lines 168-169).

4) p. 5, line 108 – add to which condition/construct the numbers refer to

Done (lines 177-182).

5) In cryo-EM samples of A2_y5+CNIH2 with ZK and PMP, about 50% of the final particles do not have CNIH2 – it might be worth commenting on this in the Discussion. Could it be that PMP causes CNIH2 dissociation? Structures do not seem to suggest this as the presence of PMP does not cause large enough conformational changes, but perhaps I am missing something. The recordings in Fig. 4a also don't seem to suggest that (no reduction in I_{ss} at 30 uM PMP), but perhaps it's a very slow process.

Structures of GluA2-y5-CNIH2 complexes were determined by transducing HEK 293S GnTI⁻ cells with GluA2-y5 baculovirus and serendipitously inducing co-expression of endogenous human CNIH2, previously identified in cultured HEK cells by transcriptome analysis. Expression of endogenous CNIH2 appears to be weaker than expression of the engineered GluA2-y5 fusion construct, as evidenced by the presence of GluA2-y5 complexes in all collected cryo-EM datasets (Extended Data Figs. 1, 6, 10, and 14). Whether application of ligands can cause dissociation of auxiliary subunits can in principle be assessed based on proportions of GluA2-y5-CNIH2 and GluA2-y5 particles. However, this is an unreliable method of estimation because apart from the proportion of the corresponding complexes, these numbers also depend on computational parameters used during processing, which are hard to take into account. We would therefore refrain from making unsupported claims. One sure thing is that we see comparable populations of GluA2-y5-CNIH2 and GluA2-y5 particles in each of four collected datasets. The corresponding information has been added to the text (lines 97-103, 113-119, 337-358).

6) Structure of the desensitized complex A2_{y5}+CNIH2+Glu indicates structural mechanism by which CNIH2 weakens AMPAR desensitization (reduced separation of D1 lobes in LBD dimers). This explains slower desensitization and higher I_{ss}, but might be at odds with slower recovery from desensitization in the presence of CNIH2 (Fig. 1F). Perhaps this is not the only desensitized conformation of A2_{y5}+CNIH2?

We performed statistical comparison of the time constants of recovery from desensitization (Figure 2f), which clearly shows that the time constants of recovery from desensitization for GluA2-y5 and GluA2-y5-CNIH2 are not statistically different. Accordingly, CNIH2 does not appear to affect recovery of GluA2-y5 from desensitization.

Reviewer #2:

Remarks to the Author:

This study describes cryo-EM structures of homotetrameric GluA2 AMPA receptors associated with TARP gamma5 and CNIH2 or with gamma5 alone. The structures provide mechanistic insights to explain the functional effects of receptor association with both a potentiating auxiliary subunit CNIH2 and an inhibiting subunit gamma5. Observations from the new structures suggest that CNIH2 destabilize the desensitized state of the receptor by reducing separation of the upper lobes in the LBD dimers. Furthermore, it is observed that CNIH2 stabilizes spermidine binding in the closed ion channel, and it is suggested that CNIH2 facilitates release of spermidine from the open channel. Effects of receptor association with gamma5 and CNIH2 on inhibition by the antiepileptic drug perampanel are also investigated and it is argued that these auxiliary subunits cause small reductions in perampanel potency. While the mechanism behind this observation was not resolved in the study, it is likely that CNIH2 simply reduces perampanel potency by influencing gating equilibrium. The observations from the new structures are nicely supported by high-quality functional data, data presentation and interpretation are reasonable, and there are no questions related to the rigor and robustness of the results and conclusions.

This is an important study that provides new structures of AMPA receptors in complex with auxiliary subunits. The GluA2-gamma5-CNIH2 structures in this study will add context and additional information to appreciate the structural mechanisms that govern the function of physiologically relevant AMPA receptors in synapses. The quality of the structures is a strength of the study, while it is considered a weakness that more physiologically relevant heterotetrameric AMPA receptors, such as for example GluA1/2, were not investigated. While the structures will certainly advance our understanding of AMPA receptor structure-function, the manuscript is a bit descriptive and the investigations of spermidine block and perampanel binding do not appear to add much in terms of raising the impact of the study. Overall, this is a nicely executed study, but there are, however, some specific points that could be addressed to improve the manuscript and strengthen the study.

We thank Reviewer #2 for the generous comments on the high quality of our structural and functional data.

- Would CNIH2 be expected to modify rectification in native GluA2-lacking, Ca²⁺-permeable AMPA receptors. Please elaborate on this. The rectification data shown in panels d-g of Extended Figure 5 are not quantitative. Are the results observed on multiple cells or just one cell? Extended Figure 5d-g might also be helpful to include in the main manuscript.

We greatly appreciate this comment of Reviewer #2. The original curves in the Extended Data Figure 5 were indeed single examples. We have now performed numerous recordings of voltage-dependencies and provide their statistical comparison. The typical examples of IV curves as well as the rectification index for GluA2, GluA2-y5, GluA2-CNIH2 and GluA2-y5-CNIH2 are now included in the main Figure 3. The corresponding information has been added to the text (lines 219-231).

- The differences in peramp panel potencies are rather small and statistical tests are not performed to suggest they are significant.

We have performed statistical comparison of the IC₅₀ values measured for individual cells contributing to the average PMP concentration dependencies illustrated in Figure 5b and shown that the IC₅₀ values are statistically different for PMP concentration dependencies measured from HEK 293 cells transfected with GluA2 versus GluA2-y5, GluA2 versus GluA2 and CNIH2, GluA2 versus GluA2-y5 and CNIH2, and GluA2-y5 versus GluA2-y5 and CNIH2 (see new panel c in Figure 5).

- The authors should elaborate on the brain regions and/or neuronal cell types in which AMPA receptors, gamma5, and CNIH2 might be co-expressed and if any direct evidence exist to suggest that gamma5 and CNIH2 can be found together in native AMPA receptor complexes.

Although y5 and CNIH2 are likely not the predominant constituents of synaptic complexes, they show similar molecular abundances in the cerebellum (Schwenk, 2014). In particular, they are present in Bergmann glia, that typically express Ca-permeable AMPARs and receive direct synaptic input from glutamatergic neurons (Soto et al., 2009). They are also present in the hippocampal NG2 glial cells (Cahoy et al., 2008; Hardt et al., 2021). The corresponding information has been added to the text (lines 76-80).

- Introduction, line 77: It would be more accurate to describe CNIH2 as a potentiating auxiliary subunit. “Activating” subunit might suggest that this protein can activate the receptor?

We agree with Reviewer #2 and replaced “activating” with “potentiating”.

- The “Summary and Outlook” section is mostly summary without outlook. Please revise to

include more perspective on how the structures and findings of the study advance our understanding of AMPA receptor structure-function.

We have revised this entire section and converted “Summary and Outlook” into the Introduction (lines 337-414).

- In the methods section, it would be helpful with a description of how domain rotations and LBD closure were determined when comparing the structures.

To determine domain rotations, we used the DynDom server (<http://dyndom.cmp.uea.ac.uk/dyndom/>). The corresponding information has been added to the Methods section. (lines 505-506).

- Lines 559-560: How will adding 7 mM NaCl to the extracellular solution increase the rate of solution exchange?

We apologize for the confusing statement. The addition of 7 mM NaCl improved visualization of the border between two solutions coming out of the two-barrel theta pipette and allowed its more precise positional adjustment for faster solution exchange. The text of the Methods section has been altered accordingly (lines 517-522).

- The fitted m values from the Hodgkin-Huxley equation should be provided.

The m values for the Hodgkin-Huxley fits have now been added to the manuscript (Extended Data Table 2, lines 185-189).

Reviewer #3:

Remarks to the Author:

A. Summary of the key results

Gangwar, Yen et al report the complex between a model AMPAR (unedited GluA2), a type-2 TARP (TARP5) and CNLH2. They analyze the structure in the presence of competitive (ZK) +/- non-competitive (PMP) antagonists and also in the presence of L-Glutamate, revealing 1) overall structural differences induced by CNLH2 binding 2) specific spermidine binding to CNLH2 complexes and lack of polyamine inhibition at positive potentials 3) the overall features of the desensitized states and 3) the binding of antagonist PMP.

B. Originality and significance: if not novel, please include references

Other AMPAR complexes including TARPs and CNLH2 have been previously reported (Zhang et al Nature 2021, Zhao et al 2019, Yu et al 2021), but here the focus is on the influence of CNLH2 on type-2-TARP containing AMPARs. A justification of the physiological relevance of such ternary complex is needed: in the cerebellum, in cells expressing AMPAR, CNLH2 and type-2 TARPs (specifically, TARP7), CNLH2 exclusively resides within cells, and not in the surface. Proteomic analysis (Schwenk et al, 2012) reveals a negative correlation between type-2

TARPs (specially TARP5) and GluA2 or CNLH2 subunits, suggesting that both auxiliary proteins would not coexist in the same complex. Therefore, any evidence supporting the physiological relevance of the ternary complex is needed.

Although $\gamma 5$ and CNLH2 are likely not the predominant constituents of synaptic complexes, they show similar molecular abundances in the cerebellum (Schwenk, 2014). In particular, they are present in Bergmann glia, that typically express Ca-permeable AMPARs and receive direct synaptic input from glutamatergic neurons (Soto et al., 2009). They are also present in the hippocampal NG2 glial cells (Cahoy et al., 2008; Hardt et al., 2021). The corresponding information has been added to the text (lines 76-80).

On the other hand this is the first time that a CNLH2-containing AMPAR complex is determined in a desensitized state. Structures also reveal binding of spermidine in a CNLH2-dependent manner; this unique observation is striking, as their functional analysis suggest that such complex is not inhibited by spermidine.

C.Data & methodology: validity of approach, quality of data, quality of presentation
Authors combine cryo-EM data with electrophysiology characterization. Cryo-EM maps are of excellent quality. It is not clear why they choose to analyze PMP binding in combination with another antagonist (see below in specific comments)

We chose to solve the closed-state PMP-bound structure in the presence of the competitive antagonist ZK because (1) all structures solved in complex with ZK were identical to the corresponding apo state structures and (2) the protein bound to the competitive antagonist is in general more stable than the apo protein (for this reason, many more structures and at higher resolution have been solved for the ZK-bound closed state compared to the apo closed state). For example, the first full-length structure of GluA2 was published in 2009, PDB ID: 3KG2, while the first true apo state structure only in 2016, PDB ID: 5L1B).

D.Appropriate use of statistics and treatment of uncertainties

Electrophysiology data includes N values and appropriate statistic tests. Showing each data point in charts as well as significance when comparing conditions may be needed. For I/V graphs, adding the analysis of rectification is needed to really be able to compare the conditions.

We very much appreciate this comment of Reviewer #3. We have now performed statistical comparison of all electrophysiological parameters (Figures 2, 3, and 5, and the corresponding text). For the IV curves, we calculated the rectification index and also performed statistical comparisons (Figure 3f-g).

E.Conclusions: robustness, validity, reliability F. Suggested improvements: experiments, data for possible revision G. References: appropriate credit to previous work?

See below for specific comments.

H. Clarity and context:

Abstract is clear, the introduction needs some clarification about the physiological relevance of the complex, a summary figure would be helpful (see below).

Specific comments are listed below:

1. The authors report that TARP5 strengthens polyamine block in unedited GluA2, contrary to what happens in GluA4-TARP5, and surprisingly, CNLH2 completely removes polyamine block and displays I/V curves similar to calcium-impermeable AMPARs. Extended figure 5 does not show clear differences between GluA2 and GluA2-G5: a comparison of the rectification index is needed as well as a discussion about GluA2/A4 differences that may explain the opposite effects observed with both AMPAR subunits.

We have calculated the rectification index for the IV curves and performed the corresponding statistical comparisons (Figure 3f-g). The previously used in Extended Data Figure 5 IV curve was an outlier and now we have collected extensive statistics and used typical IV curves in the new Figure 3e-g. We have now also mentioned in the text that GluA2- γ 5-mediated currents showed reduced inward rectification compared to GluA2, consistent with the previously observed attenuation of polyamine block of GluA4 receptors by the γ 5 subunit (lines 225-226).

2. Authors are not very convincing justifying why the only complex which is not blocked by polyamines (GluA2-TARP5-CNLH2) is actually binding spermidine in all the analyzed states (desensitized, ZK and ZK+PNP bound structures). In line 169 and line 180, authors use a very similar sentence to justify two different things: “Given the similarity of the selectivity filter in the in the closed state GluA2- γ 5-CNIH2 and open-state GluA2- γ 2... binding of polyamines to this site is the likely cause of inward rectification of Ca²⁺-permeable AMPAR currents” and “Given the similarity of the selectivity filter structure in the closed state GluA2- γ 5 and open-state GluA2- γ 2 (Extended Data Fig. 4), our results suggest that the selectivity filter in the open state of CNIH2-bound AMPAR adapts a polyamine-unbound conformation, different from the open state in the absence of CNIH2”. First, most AMPAR structures are very similar in the pore region, if the authors mean that the region is similarly accessible in open and closed structures, why is this binding specifically observed in the CNIH2 complex, which is not blocked by polyamines at positive potentials? Authors previously reported how polyamines were blocking Ca-Permeable AMPARs (Twomey et al, 2018): a comparison with these structures rather than a comparison of the pore dimensions with 5WEO would be more adequate to understand spermidine inhibition. How does the binding of NASPM and spermidine compare? Why would you expect that the binding site currently observed is the same that is blocking currents at positive potentials in the absence of CNLH2? In Twomey et al, 2018, they reported that activation was a requirement for channel block by polyamines: if the observed spermidine binding site is the blocking site at positive potentials, and here it is detected in closed receptors, how do you reconcile all the data?

We have added Figure 4 that compares the NASPM-bound open-state structure of GluA2- γ 2 and the SPD-bound closed-state structure of GluA2- γ 5-CNIH2. As clearly illustrated by this

comparison, the selectivity filter of GluA2-y5-CNIH2 may undergo substantial conformational changes upon channel opening. For example, it may undergo a significant widening, alike the widening demonstrated by the comparison of GluA2-y2^{Glu+CTZ+NASPM} and GluA2-y5-CNIH2^{ZK-SPD} (Figure 4c), which may lead to lowering of the SPD affinity and, as a result, reduced current rectification. The corresponding discussion has been added to the text of the manuscript (lines 235-244, 374-388).

3. In sentence 180, again they analyze the similarities in the pore dimensions between closed GluA2-TARP5 and open GluA2-TARP2 to conclude that the selectivity filter in the open state of CNIH2-bound AMPAR adapts a polyamine-unbound conformation. This observation must be true as there is a linear I/V response upon CNLH2 binding, but the structural data are not supporting it. Binding of spermidine to CNLH2 containing AMPARs and lack of inhibition at positive potentials is the most novel aspect in the manuscript, but needs additional support to understand how it is achieved. A comparison of the electrostatics that could point at differences at the pore between complexes, a structure of the open-state ternary complex, or molecular dynamics simulations may provide some information about why the structures actually show the opposite to what is expected. Besides, other cryo-EM maps (EMD12805, EMD12806) with CNIH2 seem to have elongated densities and display linear I/V relationship (although they are calcium-impermeable). These densities are not that clear in cryo-EM maps lacking CNLH2, and maybe this is pointing towards a more general CNLH2-dependent mechanism.

We have now commented on the paradox that CNIH2, which appears to help stabilize SPD binding in the closed state (Figure 3a-d), weakens polyamine block of open channels (Figure 3e-g). One possible explanation is that channel opening is accompanied by substantial conformational changes of the selectivity filter, alike widening demonstrated by the comparison of GluA2-y2^{Glu+CTZ+NASPM} and GluA2-y5-CNIH2^{ZK-SPD} (Figure 4c), which may lead to lowering of SPD affinity and, as a result, reduced current rectification. Open-state structures of GluA2-y5 in the presence and absence of CNIH2 may shed some additional light on this question but unfortunately, we have not been able to solve them yet. The corresponding information has been added to the text (235-244, 374-388). We have also inspected the EMD12805 and EMD12806 cryo-EM maps and found densities in the pore that are much weaker than SPD densities in GluA2-y5-CNIH2 structures. While we agree that there may be a general CNIH2-dependent mechanism, this question requires additional experiments to be carried out with AMPAR-y8 complexes.

4. The purified protein shows binding of spermidine in MS data, despite only a subset of the pure protein seems to be forming a complex with cornichon (according to the classification workflow in ext.figure 6). Have you analyzed spermidine binding to the complex lacking CNLH2 by MS? Do the densities for spermidine are still clear when refining data in C1 symmetry?

Structures of GluA2-y5-CNIH2 complexes were determined by transducing HEK 293S GnTI⁻ cells with GluA2-y5 baculovirus and serendipitously inducing co-expression of endogenous human CNIH2, previously identified in cultured HEK cells by transcriptome analysis. Expression of endogenous CNIH2 appears to be weaker than expression of the engineered GluA2-y5 fusion construct, as evidenced by the presence of GluA2-y5 complexes in all collected cryo-EM datasets (Extended Data Figs. 1, 6, 10, and 14). Obviously, we did not physically separate CNIH2-bound from CNIH2-unbound fraction (for the structures we did this computationally) and the entire sample went into the MS analysis. Since we see the density for spermidine in only CNIH2-bound structure, the MS analysis likely detects spermidine in CNIH2-bound protein only. We have refined the spermidine-bound structure in the C1 symmetry and illustrated in the new Extended Data Figure 9 that the density for SPD looks as clear as in the C2 symmetry reconstruction.

5. Previous reports in GluA1/A2/TARP8 revealed that the interaction between Ile569 and Val220 in TARP8 is relevant for polyamine inhibition (Herguedas et al, 2022), and mutations in Ile569 reduce polyamine inhibition. Do you see any differences in the region?

There is no such interaction in GluA2-y5 or GluA2-y5-CNIH2 structures. More generally, we do not see significant changes of the interface between TM4 of y5 and M1 of GluA2 (see Extended Data Figures 4 and 13). However, it is possible that such differences will be revealed in the future by open-state structures of GluA2-y5 or GluA2-y5-CNIH2.

6. The authors show that both TARP5 alone and TARP5+ CNLH2 weaken PMP inhibition and proceed to solve the structure of PMP complexes. Structures are determined in the presence of another antagonist and it seems that any modulatory difference by PMP is masked by ZK-binding. For example, extended figure 3 and extended figure 8 are almost identical, as well as the differences highlighted in lines 225-230, which seem mainly due to CNLH2 binding rather than PMP binding. Apart from showing a detailed analysis of the PMP binding site and a novel disulfide in the pore, the structures do not provide information about how PMP is exerting a different modulation on GluA2, GluA2-TARP5 or GluA2-TARP5-CNLH2 complexes. The disulfide bridge is very interesting, but it is not clear why the authors point at an increased protein dynamics in the region to justify the decrease in PMP inhibition upon CNLH2 binding? Do you have any evidence of dynamics from the cryo-EM data? Have you analyzed the excluded particles during cryo-EM analysis that could support this idea? The authors say that “PMP can influence the selectivity filter dynamics allosterically, through the M1, M3 and M4 helices” but they should specify which residues in M1, M3 or M4 helices are allosterically coupling the PMP binding site and the pore, as well as how CNLH2 may influence such PMP inhibition. In general, this section is a bit weak, as obtaining the structure of the complex with two antagonists is not ideal; the same authors recently revealed PMP binding to Kainate Receptors (Cell Reports, 2023), and in that case they used a receptor bound to an allosteric modulator. In other AMPAR structures (GluA1/2/TARP8 open state), striking differences are observed in the binding sites of

PMP, with only two of the sites accessible. Such differences are only clear in the open state structure, and a similar mechanism may be happening here. Therefore, an apo structure with PMP or CTZ-L-Glu bound structure may illuminate the PMP inhibition mechanism and the differences between complexes. Alternatively, MD simulations may support the proposed increased dynamics.

While the mechanism behind reduction in PMP potency is not resolved in our study, it is likely that CNIH2 simply reduces PMP potency by influencing gating equilibrium (see the introductory comments of Reviewer #2 whom we absolutely agree with). We realize that solving the open state structures may help understanding this mechanism better (not guaranteed however because the mechanism might be hidden in transitional states) but our attempts to do this for GluA2-y5/GluA2-y5-CNIH2 complexes have not been yet successful. Given that MD simulations can only assess a microsecond time range dynamic at best, the millisecond time scale gating transitions in AMPA receptor remain simply unapproachable for this technique. Accordingly, MD simulations unlikely would be helpful to explain attenuation of PMP inhibition upon CNIH2 binding. We proposed increased protein dynamics in the selectivity filter region but simply meant increased protein flexibility, based on the observation that the crosslink between C589 cysteines forms in GluA2-y5-CNIH2 but not in GluA2-y5 or any previously published GluA2 structures. We have now corrected the text of the manuscript accordingly. With respect to ZK binding, it has never been shown to induce significant changes in AMPAR structures compared to the apo conditions. Nevertheless, to illustrate this, we solved structures of GluA2-y5 and GluA2-y5-CNIH2 in the apo condition (Extended Data Fig. 6) and confirmed that they are indistinguishable from the ZK-bound structures. We illustrate this in the new Extended Data Fig. 7. The corresponding information has been added to the text (lines 149-156).

7. Authors report the presence of a lipid bound in the vicinity of the pore; is it only present in CNLH2-bound structures? Does the head of the lipid interact at the proximity of the extracellular region (and maybe closer to the PMP binding region) or the cytosolic region? Do you see any state-dependent or complex-dependent change in the conformation of the lipid? If the lipid may have any functional implication a figure should be presented.

We see lipids in all our structures. However, they are much less visible than protein and their heads are resolved very poorly. Therefore, we would like to refrain from overinterpretation of their densities as it definitely can lead to wrong conclusions. To better resolve lipids around the TMD of the AMPAR complex, it might require better resolution or reconstitution into nanodiscs, something that we have not achieved yet.

8. For desensitization data authors report a slower recovery of desensitization in GluA2-TARP5-CNLH2 complexes in comparison with GluA2-TARP5 and GluA2 alone, but you suggest that there is a weaker desensitization: why a reduced separation in lower lobes would lead to a much slower recovery from the desensitized state? Also, while the observations are clear, how is

cornichon specifically causing this? Apart from the “slightly more compact TMD” the authors do not analyze the details of how cornichon may modulate desensitization.

What Reviewer #3 refers to “much slower recovery from the desensitized state” is actually only a 19% increase (from 29.8 ms to 36.8 ms) in the time constant of recovery from desensitization, the effect much weaker than the (33%) increase in the time constant of desensitization (from 9.3 ms to 13.9 ms) and, importantly, the increased (7.5 times or by 647%) fraction of non-desensitized receptors (reflected in the increase in steady-state current from 0.0091 to 0.068), apparently meaning weakening of desensitization. In addition, we performed statistical comparison of the time constants of recovery from desensitization by fitting kinetics for individual cells (Figure 2f), which clearly shows that the time constants of recovery from desensitization for GluA2- γ 5 and GluA2- γ 5-CNIH2 are not statistically different. Accordingly, CNIH2 does not appear to affect recovery of GluA2- γ 5 from desensitization. Why a reduced separation in upper (not lower) lobes leads to a slightly slower recovery from desensitization is a great question. Similar to the effect of CNIH2 on PMP inhibition, it could be due to CNIH2 influencing the AMPAR gating equilibrium. We have now added the entire section to Discussion that proposes possible answers (lines 359-373). To illustrate changes in the TMD caused by CNIH2 during desensitization, we have also added additional panels j and k to Figure 8.

Minor comments:

1. Please, describe precisely how the heteromeric complex is obtained. The CNIH2 construct seems to contain GFP and strep tag, as well as the GluA2-TARP5 complex. If the same tag has been used for both constructs, how do you ensure that the “ternary” complex is obtained?

Structures of GluA2- γ 5-CNIH2 complexes were determined by transducing HEK 293S GnTI⁻ cells with GluA2- γ 5 baculovirus and serendipitously inducing co-expression of endogenous human CNIH2, previously identified in cultured HEK cells by transcriptome analysis. We purified protein from these cells using a strep affinity tag included in the GluA2- γ 5 construct. Expression of endogenous CNIH2 appears to be weaker than expression of the engineered GluA2- γ 5 fusion construct, as evidenced by the presence of GluA2- γ 5 complexes in all collected cryo-EM datasets (Extended Data Figs. 1, 6, 10, and 14). The corresponding information has been added to the text (lines 97-105, 114-119, 339-358).

2. Please, show 2Ds and processing workflow of the three datasets. In Ext. Figure 6 it seems that some classes lack auxiliary proteins (gray class), is this the case? Do you observe different stoichiometries in CNIH2 binding apart from 2-TARP5-2-CNIH2? In the ZK and L-Glu datasets, do you also observe particles lacking cornichon or is this heterogeneity only present in the PMP bound dataset?

The processing workflows for all our datasets, including examples of micrographs and 2D classes, have now been added to the manuscript (new Extended Data Figures 1, 6, 10 and 14). Yes, some 2D classes miss CNIH2 auxiliary proteins as there are classes, which have these

subunits bound and those that do not. We have not resolved stoichiometries other than the ones that we present in the manuscript.

3. Extended data Fig 6: correct “full-length”.

Thank you so much for noticing! The typo has been fixed.

4. Figure 5, label PMP a bit bigger.

Done.

5. Figure 7 is not really helping much to see the differences in desensitized states of CNLH2-lacking and CNLH2-containing complexes: maybe a zoom into the main regions that are different between complexes will be more adequate.

Two additional panels (j and k) have been added to this figure to better reveal the differences (see the new Figure 8).

6. Extended Data Figure 4: add a green label for GluA2-g5-ZK-PMP.

The corresponding label has been moved from the top of the figure to the bottom to make it more apparent (see the new Extended Data Figure 5).

Final comment:

The manuscript shows some really interesting observations (both ephys and structural data), however functional and structural observations are not fitting well, probably some MD would help a lot to reconcile all the data, as well as analyzing it in the context of other AMPAR structural biology work (including their own publications such as the Twomey Neuron paper in 2018).

As mentioned in the Reviewer #2 introductory statement, the observed effects of CNIH2 are likely due to this subunit influencing gating equilibrium. Given that MD simulations can only assess a microsecond time range dynamic at best, millisecond time scale gating transitions in AMPA receptor remain unapproachable for this technique. Accordingly, MD simulations are unlikely to help. Instead, we collected much more complete functional data and included statistical comparisons (Figures 2, 3 and 5) that now make better connection to our structural data, as we outline in our new Discussion section (lines 338-414). We have also added a new Figure 4, that compares the NASPM-bound open-state structure of GluA2-y2 from Twomey et al. (Neuron 2018) and the SPD-bound closed-state structure of GluA2-y5-CNIH2. The corresponding discussion has been added to the text (lines 235-244).

Decision Letter, first revision:

Message: Our ref: NSMB-A47227B

17th Jun 2023

Dear Dr. Sobolevsky,

Thank you for submitting your revised manuscript "Modulation of GluA2-γ5 synaptic complex desensitization, polyamine block and antiepileptic perampanel inhibition by auxiliary subunit cornichon-2" (NSMB-A47227B). It has now been seen by the original referees and their comments are below. The reviewers find that the paper has improved in revision, and therefore we'll be happy in principle to publish it in Nature Structural & Molecular Biology, pending minor revisions to satisfy the referees' final requests and to comply with our editorial and formatting guidelines. Specifically, we would like to ask you to discuss the evidence for co-expression of the two auxiliary subunits in more detail.

Sincerely,
Kat

Katarzyna Ciazynska
(she/her)
Associate Editor
Nature Structural & Molecular Biology
<https://orcid.org/0000-0002-9899-2428>

Reviewer #1 (Remarks to the Author):

I thank the authors for taking all of the comments on board, in particular for obtaining additional recordings of A2, A2_γ5, A2+CNIH2 and A2_γ5+CNIH2, IV curves, PMP dose response for A2_γ5_CNIH2 and apo structures of A2_γ5+CNIH2 with SPD and A2_γ5 with SPD.

Fig. 2b-d showcases very nicely the intermediate effects of co-expressing γ5 and CNIH2 and the stronger impact of γ5 on recovery from desensitization.

Discussion is also much stronger now.

These additional data, together with other changes greatly improve the manuscript and result in much clearer structure-function relationship of the complexes.

My only remaining comments are the following:

1) The authors do provide more info and references (refs 17, 35, 36) for the existence of A2+y5+CNH2 complexes in vivo, but the evidence is still patchy.

2) Line 196-197: "the expression of endogenous CNH2 in HEK 293 cells transfected with GluA2-y5 can be considered negligible" and Discussion lines 350-358
The statement in lines 196-197 does not really agree with the cryo-EM data where there is about 50% of A2_y5+CNH2 particles from HEK cells overexpressing A2_y5, but not CNH2 (before clean-up by heterogeneous refinement, Ext Data Fig 1). However, it might be that in cryo-EM samples, all (surface as well as intracellular) complexes are visualized, whereas in patch clamp recordings, only surface complexes are accessible. Overexpression of CNH2 might lead to more surface complexes containing CNH2, but it is very likely the authors still have a mixture of complexes at the plasma membrane and slight underestimation of y5 effects in patch clamp recordings might be possible.

Reviewer #2 (Remarks to the Author):

The authors have addressed all my points in the revised manuscript, which is strengthened by additional data and analyses. The only minor comment would be that IC50 values shown in Figure 5c are lognormally distributed and they should therefore be plotted on a log scale and the statistical analysis should be performed on logIC50 values.

Reviewer #3 (Remarks to the Author):

Gangwar, Yen, Yelshanskaya have made a great effort to address most of the comments made by reviewers, adding both structural data and electrophysiological recordings, particularly the addition of the recordings for the GluA2_CNLH2 complex.

The new information about how the ternary complex was obtained is crucial to reproduce the data, but also really intriguing, as it seems to be induced only by the transduction with A2_TARP5 baculovirus and only without ZK:

- Please, provide the details of the BACMAM A2_TARP5 construct. This will be essential to reproduce the observation.
- Transient transfection of the GluA2_TARP5 BacMam plasmid in HEK GNTI- also leads to the expression of CNLH2?

Author Rebuttal, first revision:

We thank Reviewers for additional suggestions. We have made changes in the manuscript accordingly with details outlined in our responses below.

Reviewer #1:

Remarks to the Author:

I thank the authors for taking all of the comments on board, in particular for obtaining additional

recordings of A2, A2_y5, A2+CNIH2 and A2_y5+CNIH2, IV curves, PMP dose response for A2_y5_CNIH2 and apo structures of A2_y5+CNIH2 with SPD and A2_y5 with SPD. Fig. 2b-d showcases very nicely the intermediate effects of co-expressing y5 and CNIH2 and the stronger impact of y5 on recovery from desensitization. Discussion is also much stronger now. These additional data, together with other changes greatly improve the manuscript and result in much clearer structure-function relationship of the complexes.

We thank Reviewer #1 for the kind words about our work.

My only remaining comments are the following:

1) The authors do provide more info and references (refs 17, 35, 36) for the existence of A2+y5+CNIH2 complexes in vivo, but the evidence is still patchy.

We agree with Reviewer #1 and have now changed the sentence “While TARP y5 and CNIH2 are not the predominant constituents of synaptic complexes in the brain, they show similar molecular abundancies in the cerebellum” to “While the probability of TARP y5 and CNIH2 to be constituents of the same synaptic complexes in the brain has not been studied, they show similar molecular abundancies in the cerebellum”.

2) Line 196-197: “the expression of endogenous CNIH2 in HEK 293 cells transfected with GluA2-γ5 can be considered negligible” and Discussion lines 350-358
The statement in lines 196-197 does not really agree with the cryo-EM data where there is about 50% of A2_y5+CNIH2 particles from HEK cells overexpressing A2_y5, but not CNIH2 (before clean-up by heterogeneous refinement, Ext Data Fig 1). However, it might be that in cryo-EM samples, all (surface as well as intracellular) complexes are visualized, whereas in patch clamp recordings, only surface complexes are accessible. Overexpression of CNIH2 might lead to more surface complexes containing CNIH2, but it is very likely the authors still have a mixture of complexes at the plasma membrane and slight underestimation of y5 effects in patch clamp recordings might be possible.

This is a very good point. On lines 182-183, we have now stated that the expression of endogenous CNIH2 in the plasma membrane of HEK 293 cells transfected with GluA2-y5 can be considered negligible. In the Discussion, we have now mentioned the alternative that most complexes with endogenous CNIH2 may reside in intracellular compartments instead of the plasma membrane. These complexes would still be purified for structural studies but would not contribute to recorded currents (lines 326-329).

Reviewer #2:

Remarks to the Author:

The authors have addressed all my points in the revised manuscript, which is strengthened by additional data and analyses. The only minor comment would be that IC50 values shown in Figure 5c are lognormally distributed and they should therefore be plotted on a log scale and the statistical analysis should be performed on logIC50 values.

As requested, we replaced the Figure 5c plot of the IC50 values with a plot with LogIC50 values. Statistical analysis has also been redone for the LogIC50 values. The pattern of statistical differences has not changed.

Reviewer #3:

Remarks to the Author:

Gangwar, Yen, Yelshanskaya have made a great effort to address most of the comments made by reviewers, adding both structural data and electrophysiological recordings, particularly the addition of the recordings for the GluA2_CNLH2 complex.

We are grateful to Reviewer #3 for the kind assessment of our efforts.

The new information about how the ternary complex was obtained is crucial to reproduce the data, but also really intriguing, as it seems to be induced only by the transduction with A2_TARP5 baculovirus and only without ZK:

- Please, provide the details of the BACMAM A2_TARP5 construct. This will be essential to reproduce the observation.

Detailed information about GluA2-y5 construct has been added to Methods section (lines 720-727).

- Transient transfection of the GluA2_TARP5 BacMam plasmid in HEK GNTI- also leads to the expression of CNLH2?

We have not tested transient transfection of HEK 293 GNTI- cells with GluA2-y5 BacMam plasmid because it would be too expensive for us on large scale. On a small scale, we would not have enough material to answer this question.

Final Decision Letter:

Message 26th Jul 2023

:

Dear Dr. Sobolevsky,

We are now happy to accept your revised paper "Modulation of GluA2-γ5 synaptic complex desensitization, polyamine block and antiepileptic perampanel inhibition by auxiliary subunit cornichon-2" for publication as an Article in Nature Structural & Molecular Biology.

Your paper will be published online soon after we receive proof corrections and will appear in print in the next available issue. You can find out your date of online publication by contacting the production team shortly after sending your proof corrections. Content is published online weekly on Mondays and Thursdays, and the embargo is set at 16:00 London time (GMT)/11:00 am US Eastern time (EST) on the day of publication. Now is the time to inform your Public Relations or Press Office about your paper, as they might be interested in promoting its publication. This will allow them time to prepare an accurate and satisfactory press release. Include your manuscript tracking number (NSMB-A47227C) and our journal name, which they will need when they contact our press office.

About one week before your paper is published online, we shall be distributing a press release to news organizations worldwide, which may very well include details of your work. We are happy for your institution or funding agency to prepare its own press release, but it must mention the embargo date and Nature Structural & Molecular Biology. If you or your Press Office have any enquiries in the meantime, please contact press@nature.com.

Please note that *Nature Structural & Molecular Biology* is a Transformative Journal (TJ). Authors may publish their research with us through the traditional subscription access route or make their paper immediately open access through payment of an article-processing charge (APC). Authors will not be required to make a final decision about access to their article until it has been accepted. Find out more about Transformative Journals <https://www.springernature.com/gp/open-research/transformative-journals>

Authors may need to take specific actions to achieve <https://www.springernature.com/gp/open-research/funding/policy-compliance-faqs> compliance with funder and institutional open access mandates. If your research is supported by a funder that requires immediate open access (e.g. according to <https://www.springernature.com/gp/open-research/plan-s-compliance> Plan S principles) then you should select the gold OA route, and we will direct you to the compliant route where possible. For authors selecting the subscription publication route, the journal's standard licensing terms will need to be accepted, including <https://www.springernature.com/gp/open-research/policies/journal-policies> self-archiving policies. Those licensing terms will supersede any other terms that the author or any third party may assert apply to any version of the manuscript.

In approximately 10 business days you will receive an email with a link to choose the appropriate publishing options for your paper and our Author Services team will be in

touch regarding any additional information that may be required.

Sincerely,
Kat

Katarzyna Ciazynska
(she/her)
Associate Editor
Nature Structural & Molecular Biology
<https://orcid.org/0000-0002-9899-2428>
